

# Assessment of Air Pollution in Bangkok Metropolitan Region, Thailand

Pornpan Uttamang[1], Viney P Aneja[1], Adel Hanna[1, 2]

[1]Department of Marine, Earth, and Atmospheric Sciences, North Carolina State University, Raleigh, NC, 27695, USA
[2]Institute for the Environment, University of North Carolina at Chapel Hill, Chapel Hill, NC, 27517, USA

*Correspondence to*: Pornpan Uttamang (puttama@ncsu.edu)

**Abstract.** Analysis of gaseous criteria pollutants in Bangkok Metropolitan Region (BMR), Thailand,
during 2010-2014 reveals that the hourly concentrations of CO, $SO_2$ and $NO_2$ were mostly below the

National Ambient Air Quality Standards (NAAQs) of Thailand. However, the hourly concentrations of
$O_3$ exceeded the Thailand NAAQs. The maximum concentrations of $O_3$ ranged from 120-190 ppb. On
average, the number of hourly $O_3$ exceedances ranged from 1-60 hours a year depending on monitoring
station locations. The exceedances occurred during the summer and winter, dry seasons. Interconversion
between $O_3$, NO and $NO_2$ indicates crossover points between species occur when the concentration of

$NO_x$ ($[NO_x]$ = $[NO]+[NO_2]$) is ~60 ppb. However, when $[NO_x] < 60$ ppb, $O_3$ is the dominant species;
conversely, NO dominates when $[NO_x] > 60$ ppb. The calculated photochemical reaction rate (the reaction
between $NO_2$ with sunlight), during photostationary state ranges from 0.12 to 1.22 min$^{-1}$. Linear
regression analysis between the concentrations of $O_x$ ($[O_x]$ = $[O_3]+[NO_2]$) and $NO_x$ provides the role of
local and regional contributions to $O_x$. Both the local and regional $O_x$ contributions enhance the

concentration of $O_x$. Values of the local and regional $O_x$ contributions during non-episode were ~44-54
ppb and ~ $0.13[NO_x]$ to $0.33[NO_x]$, respectively. Those values were about double during $O_3$ episodes
($[O_3] > 100$ ppb). Ratio analysis suggests that the major contributors of primary pollutants over BMR are
mobile sources (CO/$NO_x$ = 19.8). The Air Quality Index (AQI) for BMR was predominantly between
good to moderate. Unhealthy $O_3$ categories were observed during episode conditions in the region.

# 1. Introduction

Bangkok (BKK), the capital city of Thailand, has the largest population and population density in
Thailand. Bangkok Metropolitan Region (BMR) refers to Bangkok and five adjacent provinces, including



Nakhon Pathom, Pathum Thani, Nonthaburi, Samut Prakan, and Samut Sakhon. These five provinces are linked to BKK in terms of traffic and industrial development (Zhang and Oanh, 2002). Since 1995, BKK has experienced exceedances in Thailand National Ambient Air Quality Standard (NAAQs) for particulate matter (PM) and ozone ($O_3$) (PCD, 2015). The largest number of $O_3$ exceedances ($[O_3]_{hourly} >$ 100 ppb) occurred in the year 2000 with 174 hours of exceedances (Oanh and Zhang, 2004). Furthermore, BMR is considered as a region with the worst air quality in Thailand (Watcharavitoon et al., 2013). The transportation and industrial sectors are considered to be the major sources of air pollutants in BKK (Watcharavitoon et al., 2013). The number of vehicles in Thailand has increased since 1989. During 2014, about 36 million new vehicles were registered and 29 % of these cars were registered in BKK (DLT, 2015). About 56 % and 28 % of the registered vehicles in BKK were gasoline and diesel engines. The remaining 16 % is Compressed Natural Gas (CNG). According to the database of the Department of Industrial Work (DIW), Thailand, the number of registered manufacturing plants in Nakhon Pathom, Pathum Thani, Nonthaburi, Samut Prakan, and Samut Sakhon are 3,282 (DIW, 2016), 3,756 (DIW, 2016a), 1,981 (DIW, 2016b), 7,357 (DIW, 2016c) and 6,035 (DIW, 2016d). A variety of manufacturing facilities are located on the outskirts of BKK, including, metal, auto parts, paper, plastic, food, chemical manufacturing and power plants.

In this study, gaseous criteria pollutants including carbon monoxide (CO), nitrogen oxide (NO), nitrogen dioxide ($NO_2$), sulfur dioxide ($SO_2$) and $O_3$ concentrations and trends in BMR during 2010-2014 are investigated. $O_3$ and its precursors (only NO and $NO_2$) are analyzed since they are the species that were measured at a majority of the monitoring sites. Moreover, BMR experiences primarily $O_3$ exceedances amongst all the other gaseous criteria pollutants. Interconversion between $O_3$ and its precursors and photochemical reaction rate during photostationary state are examined to assess $O_3$ formation over BMR. Local emission, regional contribution and possible emission sources of pollutants that associate with $O_3$ formation are identified.



## 2. Methodology

## 1.2 Study Area

BKK is located at latitude and longitude of 13°45' N and 100°85' E, over the low flat plain of Chao Praya River, elevation height ~2.3 m above mean sea level. Thailand has three official seasons–summer (around February-May), rainy (around May-October) and winter (around October-February) as per the Thai Meteorological Department (TMD) (TMD, 2015). During rainy season, this region is influenced by Southwest monsoon wind that travels from the Indian Ocean to Thailand. This marine air mass contains high moisture, resulting in the wet season in Thailand. During this season, Thailand is characterized by cloudy weather with high precipitation and high humidity. Around October-April, this region is influenced by Northeast monsoon wind that travels from the northeastern and the northern parts of Asia (China and Mongolia). This monsoon wind brings a cold and dry air mass, resulting in the dry season in Thailand. The dry season in Thailand can be classified into two minor local seasons–winter and summer. The local winter in Thailand is characterized by cool and dry weather, while the local summer is characterized by hot (~35 °C-40 °C) to extremely hot weather (> 40.0°C) due to the strong solar radiation. During the dry season, storms may occur especially during seasonal transitions (TMD, 2015). Due to its location in the coastal area of the Gulf of Thailand, land and sea breezes may play an important role on pollution dispersion over BMR. Phan and Manomaiphiboon (2012) showed that sea breezes from the Gulf of Thailand frequently occur during winter. Strong sea breezes that penetrated inland 22-55 km were found during the early to mid afternoon.

Hourly observations collected by Pollution Control Department (PCD), Thailand, from 15 monitoring sites located in BMR are analyzed in this study. It is assumed that the monitoring sites used were representative of BMR specific patterns and trends. The monitoring sites are categorized into three categories–Bangkok (BKK) sites, Roadside sites, and BKK suburb sites). Seven Bangkok sites including 3T, 5T, 10T, 11T, 12T, 15T and 61T sites, refer to the air quality monitoring sites that are located within BKK's residential, commercial, industrial and mixed areas. These monitoring sites are ~50-100 m away from the road. Two roadside sites including 52T and 54T sites, refer to the monitoring sites that are located in BKK within 2-5 m from the road (Zhang and Oanh, 2002). Six BKK suburb sites including 13T, 14T,




19T, 20T, 22T and 27T sites, refer to the monitoring sites that are located in provinces adjacent to BKK (Pathum Thani (site 20T), Nonthaburi (sites 13T and 22T), Samut Prakan (site 19T), and Samut Sakhon (sites 14T and 27T)). Figure 1 shows a map of BMR with the major monsoon winds over this region and the monitoring sites' location.

## 2.2 Data Collection and Data Analysis

The data sets in this study were provided by the PCD. Quality assurance and quality control on the data set were performed by PCD prior to receiving the data. Hourly observations of gaseous species and meteorological parameters including wind speed (WS), wind direction (WD), temperature (T) and relative humidity (RH) were automatically collected with auto calibration at the monitoring stations. Manual quality control was performed when unusual observations were found. The equipment and monitoring stations were calibrated every year.

Gaseous species were measured at 3 m above ground level (AGL). CO was measured using non-dispersive infrared detection (Thermo Scientific 48i), NO and $NO_2$ were measured using chemiluminescence detection (Thermo Scientific 42i), $SO_2$ was measured using ultraviolet (UV) fluorescence detection (Thermo Scientific 43i) and $O_3$ is measured by using UV absorption photometry detection (Thermo Scientific 49i). The meteorological parameters including wind speed (WS) and wind direction (WD) were measured at 10 m AGL by cup propeller and potentiometer wind vanes; temperature (T) and relative humidity (RH) were measured at 2 m AGL by thermistor and thin film capacitor, respectively (Watchravitoon et al., 2013). All the meteorological measurements were made by Met One or equivalent instruments.

Data analysis, statistical data analysis (t-test) and plots are performed using Excel 2016. Predominant wind directions over BMR are illustrated by wind rose diagrams which are performed using WRPLOT program (free software from Lake Environmental).



## 3. Result and Discussion

### 3.1 Status of Pollution in BMR during 2010-2014

The maximum and average concentration of gaseous criteria pollutants, during 2012-2014, from 15 monitoring sites, were analyzed and compared with the NAAQs of Thailand (NAAQs for hourly CO, $NO_2$, $SO_2$ and $O_3$ are 30 ppm, 170 ppb, 300 ppb and 100 ppb, respectively) as shown in Fig. (2(a)-(d)). During the 5 years of the study period, maximum hourly concentrations of CO, $NO_2$ and $SO_2$ were mostly lower than their hourly standard. Elevated CO and $NO_2$ concentrations were frequently found at roadside sites. The average concentrations of CO, were ~1.0±0.7 ppm over roadside sites and ~0.7±0.4 ppm over BKK sites and BKK suburb sites. The hourly maximum concentrations of CO ranged from ~3-8 ppm. The average concentrations of $NO_2$ were ~32.2±17.7 ppb, 21.1±13.6 ppb and 16.3±11.9 ppb over roadside sites, BKK sites and BKK suburb sites, respectively. The hourly maximum concentrations of $NO_2$ ranged from 62-180 ppb (an exceedance was found at 52T monitoring station, during 2013). High $SO_2$ concentrations were frequently found over BKK suburb sites. The average concentrations of $SO_2$ were ~3.8±3.9 ppb, 3.0 ± 2.1 ppb and 2.6±2.3 ppb over BKK suburb sites, BKK sites and roadside sites. The hourly maximum concentration of $SO_2$ ranged from 13-163 ppb.

Even though the hourly maximum concentrations of the other gaseous species were generally lower than their standards, the hourly maximum concentrations of $O_3$ were greater than its standard. The average concentrations of $O_3$ were ~22.0±19.8 ppb, 17.9±16.9 ppb and 13.3±12.7 ppb over BKK suburb sites, BKK sites and roadside sites, respectively. The hourly maximum concentration of $O_3$ ranged from 68-190 ppb. $O_3$ exceedances at BKK suburb sites were more frequently occurred than those at other sites. The average number of hourly $O_3$ exceedances during 2010-2014 for BKK suburb sites, BKK sites and roadside sites ranged from ~43±21 hours a year, ~16±9 hours a year, and ~9 hours a year (Fig. (2(e))). Moreover, the exceedances of $O_3$ concentration were commonly found during the dry season, especially in January (winter). During May, the transitional period between wet and dry seasons, the number of $O_3$ exceedances decreased and $O_3$ exceedance rarely occurred during wet season (Fig. (2(f))).





## 3.2 Diurnal Variation of the Gaseous Species

The primary precursors for tropospheric $O_3$, in the urban environment, are oxide of nitrogen ($NO_x$; refers to $NO + NO_2$) and non-methane volatile organic compounds (VOCs), methane or CO (The Royal Society, 2008, Monks et al., 2009; Cooper et al., 2014). $NO_x$ was measured continuously at all the monitoring sites. However, VOCs concentrations were measured periodically only at one monitoring station limiting its usefulness as part of this study.

Diurnal variations of $O_3$ and its precursors over BMR during 2010-2014 are shown in Fig. (3(a)-(c)). The diurnal variations of $O_3$ show a typical single-peak pattern (Aneja et al., 2001) with the concentrations increase after sunrise and reach the peak around 15:00 local time (LT). The concentrations begin to decline in the evening and reach the minimum concentrations around 7:00 LT in the next morning. The concentrations at the peaks of the diurnal variations of $O_3$ were ~40 ppb over BKK sites, ~30 ppb over roadside sites and ~45 ppb over BKK suburb sites. The diurnal variations of NO and $NO_2$, show double-peak patterns with the concentrations increase around 5:00 LT and reach the first-peak around 7:00-9:00 LT before they decline. The concentrations of NO and $NO_2$ start rising and reach the second-peak around 21:00-22:00 LT. The NO concentrations at the morning-peak over BKK sites, roadside sites and BKK suburb sites were ~40 ppb, 110 ppb and 30 ppb. In the afternoon-peak they were ~23 ppb, 73 ppb and 13 ppb. The $NO_2$ concentrations at the morning-peak over BKK and BKK suburb sites were ~23 ppb and 20 ppb and those at the afternoon-peak were ~28 ppb and 22 ppb. The $NO_2$ concentrations over roadside sites ranged from ~22-37 ppb and were near constant during the day. The diurnal variations of CO show double-peak patterns with the first- and the second-peak occur around 8:00 LT and 21:00 LT. The diurnal variations of CO are similar to those of NO. The concentrations increase around 4:00-5:00 LT and reach the first sharp peaks around 8:00 LT before they decline. The CO concentrations start rising and reach the second-peak at night. The CO concentrations at the morning-peak were ~1 ppm, 2 ppm and 1 ppm and those at the night-peak were ~1 ppm, 1.5 ppm and 1 ppm, over BKK, roadside and BKK suburb sites, respectively.

Diurnal patterns of NO, $NO_2$ and CO correspond to road traffic patterns and similar to those in other big cities (Tiwari et al., 2015). The study of Leong et al. (2002) on air pollution measurement in BKK showed




that, in BKK, morning rush hour occurred during 7:00-9:00 LT and evening rush hour occurred during 16:00-18:00 LT. During traffic rush hours, traffic volume was high with low vehicle speeds. While the first peak of the diurnal pattern of pollutants occurred during the morning traffic rush hour, the second peak occurred ~3-5 hours after the evening traffic rush hour. This is due to a combination of pollutants emissions and collapse of the planetary boundary layer during this time. The evening planetary boundary layer is characterized by weak turbulence and diffusion, allowing pollutants to accumulate in the layer (Arya, 1999; Jacobson, 2012).

The concentrations of $SO_2$ start increasing around 5:00 LT and reach maximum around 8:00 LT before the decline. The concentrations of $SO_2$ at the morning-peak were ~3 ppb over BKK sites and roadside sites, and ~6 ppb over BKK suburb sites. The concentrations of $SO_2$ increase again in the afternoon and reach a second-peak around 21:00 LT over roadside site.  Over BKK sites and BKK suburb sites, the concentrations of $SO_2$ are nearly constant after 19:00 LT. The concentrations of $SO_2$ at the second-peak over roadside sites were ~3 ppb and ~3-4 ppb over BKK sites and BKK suburb sites. The double-peak pattern of $SO_2$ over roadside sites indicates that $SO_2$ is influenced by emission primarily from vehicle exhaust using high sulfur content fuel, especially high sulfur diesel. The study of ambient air $SO_2$ patterns in European cities by Henschel et al. (2013) showed that diurnal patterns of $SO_2$ had a double-peak pattern which the morning peaks more likely related to emission during rush hour, evening peaks were possibly caused by traffic and meteorology–collapse of the planetary boundary layer. It is noteworthy that BKK has a large diesel engine fleet (an estimated 25 % of registered vehicles) (DLT, 2015a). The diesel fuel contains ~0.035 %wt Sulphur (DOEB, 2017). Given the timing of $SO_2$ peak (morning automotive rush hour), it is likely that $SO_2$ is emitted by automotive diesel engine exhaust.

### 3.3 Interconversion between $O_3$, NO and $NO_2$ and Photochemical Reaction

In this study, the photostationary state (PSS) is applied through all chemical reactions for $O_3$ formation during 10:00-16:00 LT. This time window is chosen due to the fully developed atmospheric boundary layer with well-mixed condition (Pochanart et al., 2001) in order to avoid accumulation due to surface inversion. To eliminate effects of the removal process by wet deposition, analysis and calculation are performed only during dry season.



The relationship among three chemical species (NO, $NO_2$ and $O_3$) under PSS is presented by Eq. (1) (Seinfeld and Pandis,1998)

$$[O_3]_{PSS} = \frac{j_1[NO_2]}{k_3[NO]} \qquad (1)$$

Where $[O_3]_{PSS}$ is the concentration of $O_3$, at PSS, $j_1$ and $k_3$ are reaction rate coefficient of photochemical reaction of $NO_2$ and reaction rate coefficient of chemical reaction between NO and $O_3$, respectively. According to Eq. (1), the concentration of $O_3$ depends on the ratio of $NO_2$ and NO. Therefore, other chemical reactions or processes that affect $NO_2$ and NO species will also affect $O_3$ concentrations in the atmosphere (Jacobson, 2012).

The ratio of $NO_2$ and NO are calculated only during dry season. During dry season, the values of the rations range from 0.54-4.33 in winter and from 0.87-4.33 in summer. T-test values for the ratios exhibit no significant difference with season (P-value > 0.05). While there is no significant difference with season, the t-test values exhibit a significant difference with locations of monitoring sites. The ratios of $NO_2$ and NO show significantly different between roadside sites and non-roadside sites (BKK sites and BKK suburb sites) with P-value < 0.05.

In this study, $j_1$ is calculated based on Eq. (1), since we cannot directly measure it. The values of $j_1$ range from 0.12-1.22 min$^{-1}$ in winter and from 0.13-0.90 min$^{-1}$ in summer (Table (1)). T-test values for $j_1$ exhibit no significant difference with season and location (P-value > 0.05). The values of $j_1$ from this study are similar to those values at an urban background site in Delhi, India (values of $j_1$ ranged from 0.4-1.8 min$^{-1}$ and the average was 0.8 min$^{-1}$) (Tiwari et al., 2015) and those values collected during a November daytime in the UK (values of $j_1$ was ~0.14 min$^{-1}$) (Clapp and Jenkin, 2001).

The values for $k_3$ (ppm$^{-1}$ min$^{-1}$) is calculated by Eq. (2) (Seinfeld and Pandis, 1998; Tiwari et al., 2015).

$$k_3 = 3.23 \times 10^3 \exp[-1430/T] \qquad (2)$$

During dry season, the values of $k_3$ range from 28.3-29.8 ppm$^{-1}$ min$^{-1}$ in winter and from 30.0-30.9 ppm$^{-1}$ min$^{-1}$ in summer. T-test values for $k_3$ exhibit a significant difference with season (P-value < 0.05) and no significant difference with locations of the monitoring sites (P-value > 0.05) (Table (1)). Since $k_3$ is a





function of temperature (T), therefore, the maximum values of $k_3$ (29.6 and 30.8 ppm$^{-1}$ min$^{-1}$ in winter and summer, respectively) occur during the afternoon (around 15:00 LT) when the temperature is highest. The maximum values of $k_3$ from this study conforms to the $k_3$ value (29.3 ppm$^{-1}$min$^{-1}$) that was found at an urban background site in Delhi, India, which the peak occurred at 15:00 LT (Tiwari et al., 2015).

Due to high value of $j_1$, high $O_3$ concentrations are expected to be found at 11T, 20T and 52T sites. However, high $O_3$ concentrations were found only at 20T and 52T sites, but low at 11T site. The low level of $O_3$ concentration at 11T site has an association with the titration of $O_3$ by NO, since high NO concentrations were observed at 11T site. In conclusion, the titration of $O_3$ by NO is perhaps one of the more important processes that control $O_3$ concentrations in urban areas.

To gain a better understanding of $O_3$ and its precursors over BMR, the concentrations of NO, $NO_2$ and $O_3$ are plotted against the concentrations of $NO_x$. Polynomial trend lines are added in order to investigate the interconversion among these species. Figure (4(a)-(c)) show relationship and crossover points between the species. The crossover points occur when the concentration of $NO_x$ is ~60 ppb. At this point, two regimes are identified–low $NO_x$ regime and high $NO_x$ regime. Under low $NO_x$ regime ([$NO_x$] < 60

ppb), $O_3$ is the dominant species among the others, and $NO_2$ concentrations are higher than NO for $NO_x$ species. On the other hand, under high $NO_x$ regime ([$NO_x$]> 60 ppb), NO and $NO_2$ increase and, the concentrations of $O_3$ rapidly decrease. Under the high $NO_x$ regime, the decrease of $O_3$ trend-lines may describe $O_3$ removal process through the titration of $O_3$ by NO.

**3.4 Local and Regional Contribution to $O_x$**

The $O_x$ concentration is the summation of $O_3$ and $NO_2$ concentration. Under the PSS condition, concentration of NO, $NO_2$ and $O_3$ approach an equilibrium and the concentration of $O_x$ may be considered constant (Keuken et al., 2009). Since the conversion between $O_3$ and $NO_2$ in the urban and suburban atmosphere is rapid, the use of $O_x$ to represent production of oxidants is more appropriate than only using $O_3$ (Lu et al, 2010). The local or $NO_x$-dependent contribution refers to $O_x$ concentration that is influenced

by concentration of the local pollutants. The regional contribution or $NO_x$-independent refers to the background concentration of $O_x$ that is not influenced by changes of the local pollutants (Clapp and Jenkin, 2001; Tiwari et al. 2015).



The effects of local and regional contributions to $O_x$ concentration are analyzed by plotting $O_x$ concentrations against $NO_x$ concentrations and fitting the plot with a linear regression (y = mx + c). The concentration of $NO_x$ and $O_x$ are referred by x and y, respectively. The slope of the linear regression (m) implies the local contribution, and the intercept with the y-axis (c) implies the regional (background) contribution (Aneja et al., 2000; Clapp and Jerkin, 2001; Notario et al., 2012). Table (2) shows the comparison between the fitted linear regressions from this study with other studies. The average background $O_x$ concentrations over BMR during non-episodes and episodes are ~48 ppb and 95 ppb, respectively. The local and regional contributions during the episode days, in general, were about double of those during the non-episode days. Therefore, $O_3$ formations during the episode days were influenced by both the local and regional contributions of $O_x$. It is noteworthy that the pattern of the local and regional contributions at roadside sites during non-episode period is composed of two $NO_x$ concentration regimes. The low $NO_x$ regime ($NO_x < 60$ppb) resembles the local and regional contributions during non-episode over BKK suburb sites. The high $NO_x$ regime ($NO_x > 60$ppb) may represent typical characteristic of air quality near roads.

The local contributions from the fitted linear regressions are compared with the local contribution that is calculated from delta $O_3$ method. A delta $O_3$ ($\Delta O_3$) analysis was performed to reflect on the intensity of $O_3$ production in BMR area (Lindsay and Chameides, 1988). We utilized hourly $O_3$ concentrations during 10:00-16:00 LT reflecting the role of photochemistry in $O_3$ formation. Thus, the difference between $O_3$ concentration measured at Samut Sakhon Provincial Administrative (site 27T) and Bangkok University Rangsit Campus (site 20T) during the predominant wind direction should reflect the difference in the amount of $O_3$ leaving and entering the city whenever the winds are out of the Southwest or Northeast direction (Fig. (6)). This analysis provides the net increment of photochemical $O_3$ added to an air mass over the course of the day as it advects over the city. For a more rigorous delta $O_3$ analysis, we need to consider the role of wind speed. Lindsay et al. (1989) analyzed high-$O_3$ events in Atlanta, GA, and showed that rural background $O_3$ during episode days ($[O_3] > 80$ ppb) in Atlanta Metropolitan Area were higher than its average and the concentration of $O_3$ increased from ~15-20 ppb when the air mass travelled across the city, enhancing the total $O_3$ concentration to 80-85 ppb. In our study, the concentrations of $O_3$ over the upwind and downwind monitoring sites are averaged during 10:00-16:00 LT in dry season when




backward trajectories from the National Oceanic and Atmospheric Administration (NOAA) HYSPLIT model reveal N-NE, S-SW wind directions with high $O_3$ concentrations ($[O_3] > 80$ ppb) at the monitoring sites. The $O_3$ concentrations at the downwind monitoring stations are expected to be greater than the $O_3$ concentrations at the upwind monitoring stations. However, a negative $\Delta O_3$ may be found. The negative $\Delta O_3$ suggests deposition of $O_3$ and/or $O_3$ was consumed as it passes over the city and/or there may have been a wind reversal so that air already polluted by the metropolitan area was brought back in to the city (Lindsay et al., 1989). The $\Delta O_3$ in BMR ranged from -53 to 86 ppb (average about 10.4 ppb.), and ranged from -66 to 96 ppb (average ~9.4 ppb.) when the predominant wind direction advecting into the city were from NE and SW, respectively. Thus, we find that there was ~10 ppb enhancement of the $O_3$ concentration during the air pollution high $O_3$ concentration in BMR ($[O_3] > 80$ ppb), which corroborates local $O_3$ production analysis based on linear regression.

## 3.5 Correlation of Air Pollutants

### 3.5.1 Local Sources Analysis

Characteristic of emission sources are often determined by the ratios between $CO/NO_x$ and $SO_2/NO_x$. In general, the major sources of $NO_x$ are point sources and mobile sources. However, $NO_x$ from point sources is more likely correlated with $SO_2$. $NO_x$ from mobile sources is more likely correlated with CO (Parrish et al., 1991). Therefore, the characteristics of mobile source are high $CO/NO_x$ ratios and low $SO_2/NO_x$ ratios. In contrast to mobile sources, the characteristic of point sources are low $CO/NO_x$ ratios and high $SO_2/NO_x$ ratios (Parrish et al., 1991; Rasheed et al., 2014).

Table (3) shows the comparison between the $CO/NO_x$ and $SO_2/NO_x$ ratios from this study and when compared with other studies. The ratio of $CO/NO_x$ is 19.8 and the ratio of $SO_2/NO_x$ is 0.1 over BMR. This suggests that the major contributors of primary pollutants over the BMR are mobile sources. However, this region may also be influenced by manufacturing facilities' point sources ($SO_2$ contributor) on the outskirts of the BKK. These point sources will impact the concentrations of $SO_2$, $NO_x$ and CO.



### 3.5.2 Effects of Pollutant Transport

In general, O$_3$ has a short (approximately hours) lifetime in polluted urban atmosphere. However, O$_3$ has a longer lifetime of several weeks in the free troposphere. This occurrence may allow O$_3$ to be transported over continental scales (Stevenson et al., 2006; Young et al., 2013; Monks et al., 2015). Figure 6 shows high O$_3$ concentrations ([O$_3$]$_{hourly}$ > 100 ppb) with the predominant wind directions over BMR during 2010 to 2014. The results show that O$_3$ exceedances are associated with the local wind directions which are related to locations of the monitoring sites. High O$_3$ concentrations are associated with the three predominant wind directions; westerly, northerly and southerly winds. Elevated O$_3$ concentrations associated with northerly winds were at 11T, 13T, 14T, 22T and 27T sites. At sites 3T, 5T, 10T, 12T, 15T, 19T, 54T and 61T, high O$_3$ concentrations are associated with southerly winds. At sites 52T and 20T high O$_3$ concentrations were predominantly observed with westerly wind directions. The results from this study are supported by an earlier study (Sahu et al, 2013) that showed pollution concentrations over BKK related with local wind direction.

### 3.6 Air Quality Index for O$_3$ Management

Enhanced ambient air pollution has an association with increased risk of adverse cardiovascular morbidity and mortality for humans. For example, increased levels of O$_3$ causes coughing, reduces lung function, enhances pulmonary inflammation and may increase the risk of death due to respiratory diseases (US.EPA, 2017c). While adverse health effects may occur in healthy people, enhanced ambient air pollution is a serious threat to sensitive groups (i.e. children, elders and people with respiratory system diseases). Increased lifetime exposure of tropospheric O$_3$ was a cause of decreased lung function in young adults (Targer et al., 2005). Buadong et al. (2009) studied the association between O$_3$ exposure and hospital visits for cardiovascular diseases (CVD) in the central of BKK, Thailand. The study showed a positive relationship between exposure to O$_3$ on the previous day with increasing number of hospital visits for CVD in elderly patients ($\geq$ 65 years). Fann et al (2011) studied the relationship between O$_3$ exposure with the national public health burden in the U.S. and found O$_3$ associated with premature death in metropolitan areas where these numbers were greater than other habitable environs. The study of the Global Burden of Disease, Injuries, and Risk Factor study 2013 (GBD 2013) for 188 countries by



Forouzanfar et al. (2015) reported the increased number of deaths during 1990 to 2013 (from 133 to 219 deaths in thousands) due to ambient $O_3$ pollution. World Health Organization (WHO) Regional Office for Europe, Economic Co-operation and Development (OECD) (2015) estimated the annual economic cost of premature deaths and those of morbidity from air pollution between US $1.431 trillion and $1.575 trillion across the countries of the WHO European Region.

For air pollutant species in the US, the AQI for each species is categorized into 6 categories (good, moderate, unhealthy for sensitive groups, unhealthy, very unhealthy, and hazardous). These categories are nonlinear and relate to human health (US.EPA, 2017, 2017a, 2017b). In Thailand, the NAAQs for the air pollutant species is pegged at an AQI value of 100. In the US AQI rating system the results were the following for Thailand: the majority of air quality over BMR were in the good AQI category (~93-99 %); followed by the moderate air quality category. However, unhealthy for sensitive group (88-632 hours), unhealthy (19-209 hours) and very unhealthy (2-59 hours) $O_3$ air quality categories were found during the study period. In general, BKK suburb sites have higher number of hours that were found in the unhealthy for sensitive group, unhealthy and very unhealthy categories than BKK and roadside sites. The average number of hours that were found in unhealthy for sensitive group, unhealthy and very unhealthy categories over BKK suburb sites were 425.8, 146.7 and 28.7 hours. Table (4) provides the ambient air quality over BMR during 2010 to 2014 based on the AQI of $O_3$.

This study provides measurements and analysis for the gaseous criteria pollutants. However, in order to provide a well-established air quality management policy, the integration of multidisciplinary analysis is needed. This will include scientific, socioeconomic and policy analysis (Aneja et al, 2001). The results from this study reveal evidence of violations for $O_3$ for air quality resulting in adverse health effects, human welfare, economics and environment over BMR. Source analysis suggests to control pollution emission from local sources that emissions from mobile sources should be the first priority. The complexity between $O_3$ and its precursors and the effects of pollution transport shows that decreasing only $NO_x$ emissions and/or local emissions may not be an effective policy to reduce $O_3$ since regional air pollution transport contributes to $O_3$ exceedances. To identify the proportional contribution between local and regional sources of $O_3$ concentrations during selected $O_3$ episode days, atmospheric modeling is needed to quantify various processes that contribute to the ambient concentration at specific locations.



This scientific analysis provides a frame work for the process of establishing an air quality policy while developing socioeconomic impacts.

## 4. Conclusion

Among measured gaseous criteria pollutants, $O_3$ is the only species whose concentrations frequently exceed the NAAQs of Thailand. The $O_3$ exceedances occur during the dry season (summer and winter) and most frequently occur over BKK sites and BKK suburb sites than roadside sites; which $O_3$ titration by NO played an important role to decrease $O_3$ concentrations. Interconversion between $O_3$, NO and $NO_2$ and photochemical reaction shows that $O_3$ has a non-linear relationship with its precursor with high concentrations of $O_3$ which occur when $NO_x$ concentration is less than 60 ppb. After this point, $O_3$ concentrations rapidly decrease, while $NO_x$ concentrations increase. Under high $NO_x$ regime, the concentration of $O_3$ is influenced by NO through the titration process. The result for the study shows that decreasing $NO_x$ emission will not directly decrease $O_3$ concentration over BMR. The regression curves reveal a background $O_x$ concentration of ~48 ppb (non-episode) and ~95 ppb (episode) over BMR. During an $O_3$ episode, both local and regional contributions play an important role in the increase of $O_x$ concentrations. The result reveals that, decreasing emission from only local sources may not improve air quality during $O_3$ episodes, since regional air pollution transport contributes to $O_3$ formation. Sources analysis suggests that to control pollution emission from local sources, the emissions from mobile roadside sources should be the first priority. Air Quality Index for $O_3$ reveal evidence of violations for $O_3$ for air quality resulting in adverse health effects, human welfare, economics and environment over BMR.

## Data Availability

Hourly observations in this study are provide by Pollution Control Department (PCD), Thailand.

Address: 92 Phahonyothin Rd, Khwaeng Samsen Nai, Khet Phaya Thai, Krung Thep Maha Nakhon 10400, Thailand.

Phone: +66 2 298 2000

Website: http://www.pcd.go.th/



**Competing Interest**

The authors declare that they have no conflict of interest.

**Acknowledgement**

We thank the Royal Thai Government for providing the Fellowship to Uttamang (ref. No.1018.2/4440).

5    We thank Professor Surat Bualert, Miss Naboon Riddhiraksa, the Pollution Control Department of the Ministry of Natural Resources and Environment, Bangkok, Thailand and Thai Meteorological Department of the Ministry of Information and Communication Technology for providing QA/QC air pollution and meteorology data. We also thank Ms. Elizabeth Adams for her assistance in the editorial review of the manuscript



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





# Figures

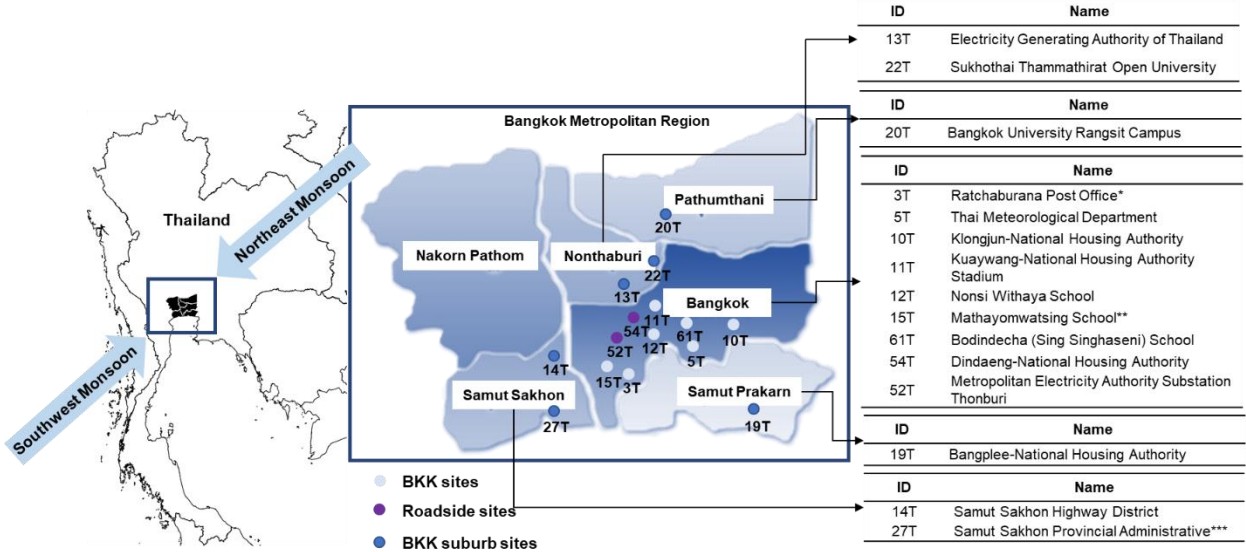

**Figure 1:** Map of BMR, including BKK and five adjacent provinces, with the two major monsoons winds. The locations of three categories of monitoring sites including BKK sites, roadside sites and BKK suburb sites are shown in light blue dots, purple dots and blue dots, respectively. (Note: * the station has been closed since 1 January 2014; ** the station has been closed since 1 August 2015; *** the station has been closed since 1 October 2013).

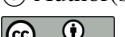



**Figure 2:** Maximum (vertical bars) and average (solid line) concentrations of (a) CO, (b) $SO_2$, (c) $NO_2$ and (d) $O_3$ from 15 monitoring stations, during 2010-2014, are compared with the 1-hour NAAQs (dotted line) of Thailand. Three different sites, the BKK sites, roadside sites and the BKK suburb sites are represented by light blue, purple and blue colors. The number of exceedances of hourly $O_3$ concentration are shown by (e) monitoring stations, and (f) months.



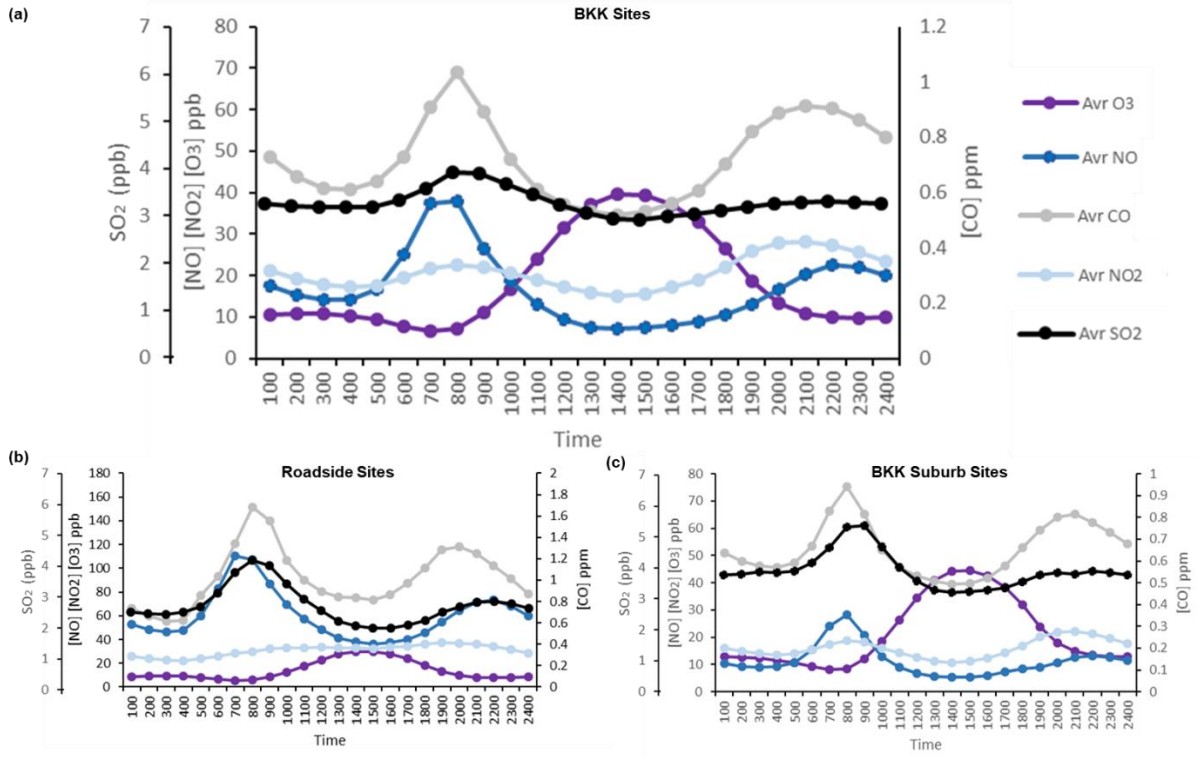

**Figure 3:** Diurnal variations of gaseous species including NO, NO₂, CO and SO₂ at (a) BKK site (b) roadside sites and (c) BKK suburb sites.

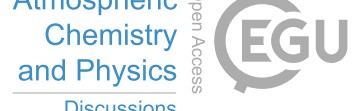



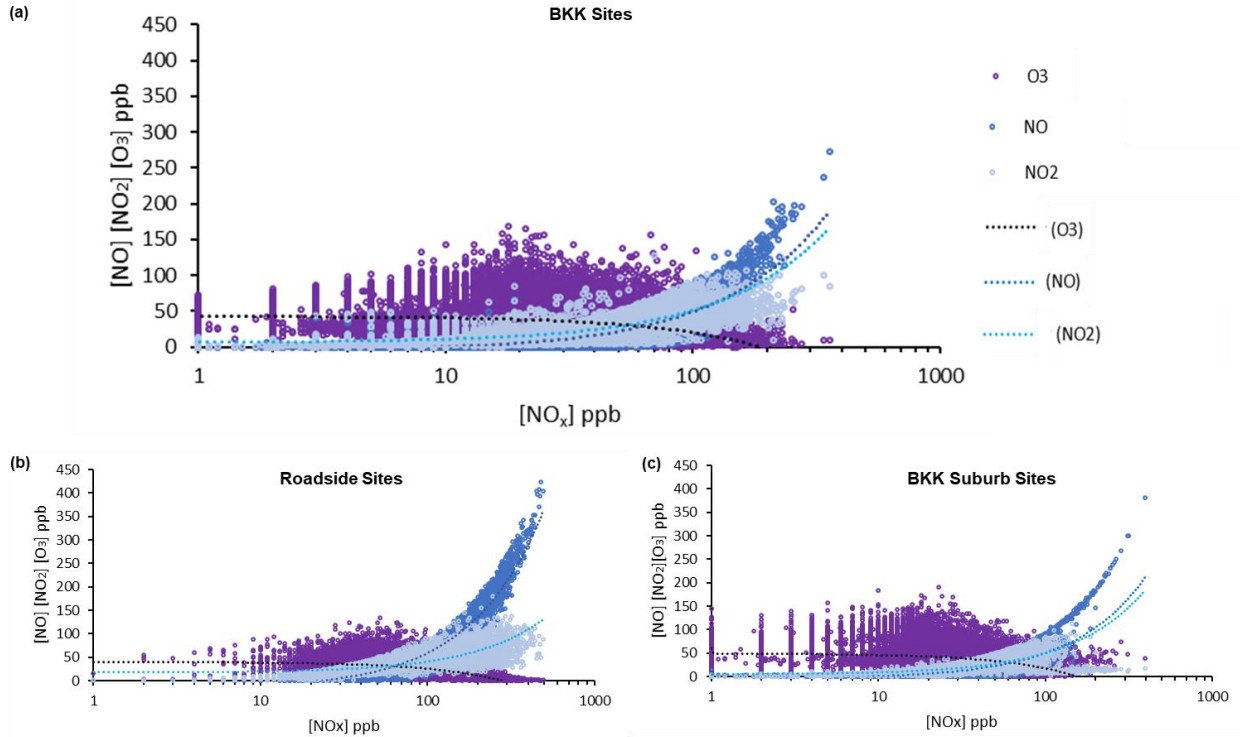

**Figure 4:** Interconversion between $O_3$, NO and $NO_2$ during 10:00-16:00 LT for 2010 to 2014 (a) BKK sites, (b) roadside sites and (c) BKK suburb sites. The polynomial regressions provide the crossover point for $NO_x$ (60 ppb).





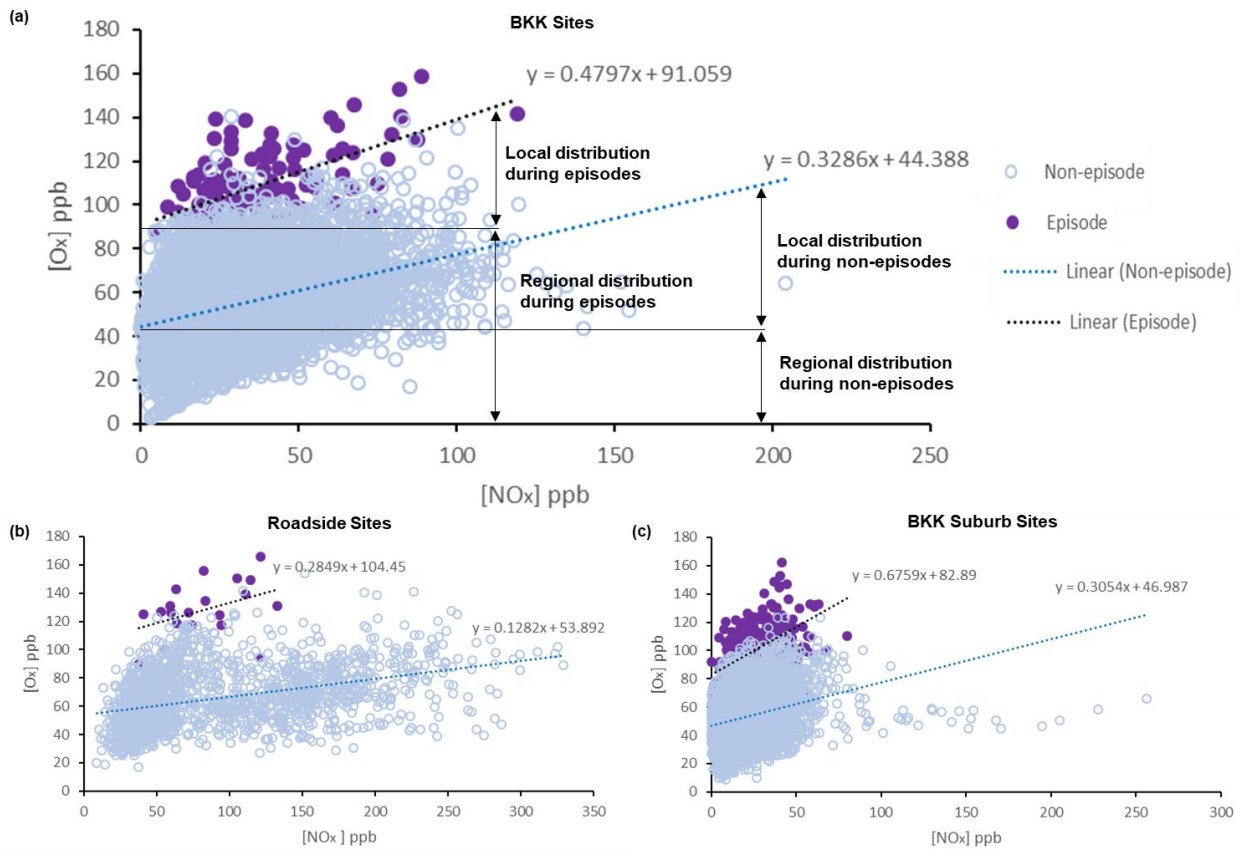

**Figure 5:** Effects of local and regional contributions on $O_x$ during non-episode and episode days over BMR (a) BKK sites (b) roadside sites and (c) BKK suburb sites during 2010-2014.



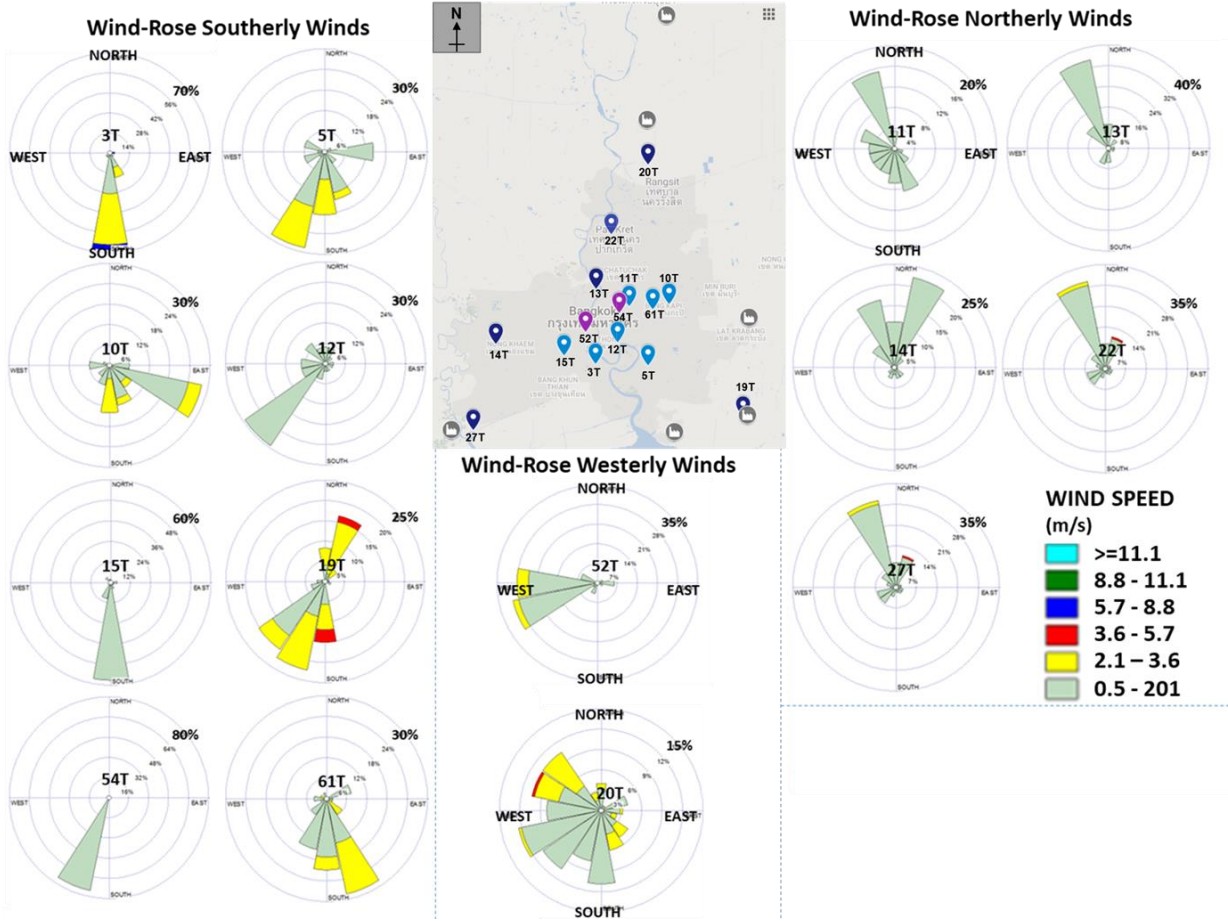

**Figure 6:** Relationship between high O₃ concentration ([O₃]ₕₒᵤᵣₗᵧ > 100 ppb) (monitoring site ID is listed at the center of the Wind Rose Plots) and wind directions over BMR during 2010 to 2014. The map illustrates several industrial areas located near the study area and the monitoring stations, blues, purples and dark blues identify BKK sites, roadside sites and BKK suburb sites; greys identify industrial areas





# Tables

**Table 1:** chemical rate coefficients ($k_3$, $j_1$) during winter and summer from the BKK sites, roadside and BKK suburb sites, 2010-2014

| Coefficient | Season | BKK sites | | | | | | | Roadside sites | | BKK suburb sites | | | | | |
|---|---|---|---|---|---|---|---|---|---|---|---|---|---|---|---|---|
| | | 3T | 5T | 10T | 11T | 12T | 15T | 61T | 52T | 54T | 13T | 14T | 19T | 20T | 22T | 27T |
| $k_3$ | Winter | 29.1 | 29.8 | 29.2 | 29.1 | 29.4 | 29.0 | 28.7 | 29.5 | 28.3 | 29.5 | 29.2 | 28.8 | 29.2 | 29.3 | 29.2 |
| ($ppm^{-1}$ $min^{-1}$) | Summer | 30.1 | 30.9 | 30.4 | 30.2 | 30.5 | 30.2 | 30.1 | 30.4 | 30.5 | 30.5 | 30.7 | 30.0 | 30.8 | 30.9 | 30.0 |
| $j_1$ | Winter | 0.12 | 0.50 | 0.32 | 0.76 | 0.95 | 0.39 | 0.50 | 0.79 | 0.51 | 0.42 | 0.39 | 0.37 | 1.22 | 0.34 | 0.53 |
| ($min^{-1}$) | Summer | 0.13 | 0.51 | 0.47 | 0.72 | 0.43 | 0.57 | 0.23 | 0.90 | 0.36 | 0.50 | 0.69 | 0.49 | 0.86 | 0.36 | 0.46 |



**Table 2:** the comparison of fitted linear regression from this study, including from BKK sites, roadside sites and BKK suburb sites with other studies.

|  | **Non-Episode** | **Episode** |
|---|---|---|
| **This study** |  |  |
| *-BKK sites* | $[O_x] = 0.33[NO_x]+44.39$ | $[O_x] = 0.48[NO_x]+91.10$ |
| *-Roadside sites* | $[O_x] = 0.13[NO_x]+53.89$ | $[O_x] = 0.29[NO_x]+104.45$ |
| *-BKK suburb sites* | $[O_x] = 0.31[NO_x]+47.0$ | $[O_x] = 0.68[NO_x]+82.89$ |
| UK* | $[O_x] = 0.112[NO_x]+55.5$ | $[O_x] = 0.097[NO_x]+38.2$ |
| Buenos Aires, Argentina** | $[O_x] = 0.099[NO_x]+22.0$ | |
| Delhi, India*** | $[O_x] = 0.54[NO_x]+28.89$ | |

*Clapp and Jenkin (2001)

**Mazzeo et al. (2005)

5   ***Tiwari et al. (2015)



**Table 3:** the comparison of $CO/NO_x$ and $SO_2/NO_x$ ratios from this study with other studies (modify from Rasheed et al., 2014)

| Region | Source | $CO/NO_x$ | $SO_2/NO_x$ |
|---|---|---|---|
| **This study** | | **19.8** | **0.1** |
| *- BKK sites* | | 18.25 | 0.09 |
| *- Roadside sites* | | 21.15 | 0.11 |
| *- BKK suburb sites* | | 19.20 | 0.09 |
| Eastern US | | 4.3 | 0.94 |
| | Mobile | 8.4 | 0.05 |
| | Point | 0.95 | 1.8 |
| Pennsylvania | | 2.6 | 1.7 |
| | Mobile | 7.8 | 0.05 |
| | Point | 0.8 | 2.3 |
| Western US | | 6.7 | 0.41 |
| | Mobile | 10.2 | 0.05 |
| | Point | 1.2 | 1.1 |
| Denver Metropolitan | | 7.3 | 0.19 |
| | Mobile | 10.5 | 0.05 |
| | Point | 0.18 | 0.44 |
| Raleigh, NC | | 16.3 | 0.73 |
| New Delhi, India | | 50 | 0.58 |
| Madrid City, Spain* | | 13.3 | 0.29 |
| Rouen City, France** | | 12-18 | |
| Islamabad, Pakistan | | | |
| *- Based on Emission Inventory, 2010* | Mobile | 4.94 | 0.34 |
| | Point | 0.63 | 7.0 |
| *- Based on Ambient Data* | | 10 | 0.01 |

* Fernandez-Jiménez et al., 2003

** Coppalle et al., 2001





**Table 4:** Number of hours that were found in different AQI categories of $O_3$ over the BMR during 2010 to 2014

| AQI | Hour | | | | | | | | | | | | | | |
| | BKK sites | | | | | | | Roadside sites | | BKK suburb sites | | | | | |
| | 3T | 5T | 10T | 11T | 12T | 15T | 61T | 52T | 54T | 13T | 14T | 19T | 20T | 22T | 27T |
| Good | 39018 | 32021 | 27959 | 40715 | 26606 | 33628 | 26442 | 32665 | 40231 | 31070 | 35429 | 33592 | 30793 | 34301 | 26873 |
| Moderate | 310 | 713 | 1023 | 556 | 367 | 479 | 1178 | 807 | 27 | 1620 | 944 | 1687 | 1340 | 1466 | 719 |
| Unhealthy for Sensitive Group | 88 | 139 | 225 | 109 | 82 | 108 | 295 | 151 | 0 | 454 | 288 | 515 | 632 | 448 | 218 |
| Unhealthy | 19 | 40 | 61 | 30 | 29 | 38 | 85 | 36 | 0 | 195 | 87 | 184 | 209 | 109 | 96 |
| Very Unhealthy | 0 | 6 | 12 | 0 | 0 | 10 | 26 | 0 | 0 | 59 | 2 | 51 | 28 | 23 | 9 |
| Hazardous | 0 | 0 | 0 | 0 | 0 | 0 | 0 | 0 | 0 | 0 | 0 | 0 | 0 | 0 | 0 |