# Peer review of "Assessment of Gaseous Criteria Pollutants in Bangkok"

_Atmospheric Chemistry and Physics, 2017_

## Referee Comment (RC1) · Anonymous Referee #1 · 9 Mar 2018

**General comment**

Pornpan Uttamang et al. have presented observations of CO, $NO_x$, $SO_2$ and $O_3$ from 15 monitoring sites at understudied Bangkok Metropolitan Region (BMR) for a five-year-long period from 2010-2014. Background pertaining to the air-quality in terms of PM and $O_3$ exceedance events in the BMR is provided. However, the authors do not mention the knowledge gap or scientific question that they want to address from this study. I have major concerns with the paper which include description of analytical methods and discussion about quality control (calibration and sampling protocols, filter criteria) of dataset used. The statistical analysis is also weak which mostly covers average/maximum over the entire study periods, without going into details of specific

seasons, inter-annual trends and pin-pointing the season-specific emission sources / formation processes and removal processes of the pollutants. The conclusions are drawn either from the regression lines having poor fit parameters or oversimplification of methods for source identification available in the peer-reviewed literature. The manuscript needs to address the major concerns (highlighted in specific comments) before it can be considered further. After performing the analysis suggested in the specific comments, corrections and restructuring the paper, the scientific outcome might be significantly different from the present version and should be considered as a new publication.

**Specific comments:**

**Title**

Authors might consider making the title of the paper more specific. Authors assess CO, $NO_x$, $SO_2$ and $O_3$ air pollution and not overall air pollution in general.

**Introduction**

The authors have included a description of auto-mobile fleet and manufacturing industries in the introduction which should rather be a part of the site description. The introduction is poorly structured. Authors should include a brief literature review of the previous works from BMR, outlook from these studies and what are the knowledge gaps they want to address from this paper.

The authors should also mention, why they have chosen to study CO, $NO_x$, $SO_2$ and $O_3$. At-least a line each about their importance regarding atmospheric chemistry and air quality should be present. The authors have referred to Zhang and Oanh, [2002]

for the site description. However the findings there should also be mentioned in the introduction, as Zhang and Oanh, [2002] have analyzed monthly and diel variation, $O_3$ exceedances, drivers for high ozone episodes and relationship of ozone production with $NO_x$/NMHC ratio. These are quite relevant for the present study. Similarly, the work of Pochanart et al., [2001] should be highlighted in the introduction. I found few other studies (mentioned below) which are relevant to the present work and should be highlighted in the introduction. There might be several more!

Jinsart, W., Tamura, K., Loetkamonwit, S., Thepanondh, S., Karita, K., and Yano, E.: Roadside Particulate Air Pollution in Bangkok, Journal of the Air & Waste Management Association, 52, 1102-1110, *10.1080/10473289.2002.10470845*, 2002.

Suthawaree, J., Tajima, Y., Khunchornyakong, A., Kato, S., Sharp, A., and Kajii, Y.: Identification of volatile organic compounds in suburban Bangkok, Thailand and their potential for ozone formation, Atmospheric Research, 104-105, 245-254, *10.1016/j.atmosres.2011.10.019*, 2012.

**Page 2, Line 23:** Authors state "possible emission sources of pollutants that associate with $O_3$ formation are identified ". However, such identification is not discussed in the manuscript. Authors have only used the ratio of CO/$NO_x$ and $SO_2$/$NO_x$ to identify whether the emission sources are mobile or point in nature. The method itself has an inherent limitation which is mentioned later in the specific comment for the section.

**Methodology**

The exact measurement period should be mentioned in this section. This paper discusses a five-year-long measurement period and shows data over 15 different measurement stations and authors should provide a time-line for data availability for each station.

**Page 3, line 22:** What is the basis of the assumption that monitoring sites used were

representative of BMR specific patterns and trends?

**Data Collection and Data Analysis:** I have major concerns with this section. Authors did not provide any sampling details. The trace gas analysers for CO, $NO_x$, $SO_2$ and $O_3$ are known to have drifts with time. Authors mention that equipment and monitoring stations are calibrated every year. This is not enough. There should be frequent zero drift check for CO (at-least daily) and for $NO_x$, $SO_2$ and $O_3$ (at-least once a week). The linearity of the detection should also be checked with calibration experiments performed at-least once a month. The authors did not provide any information about the drift in the sensitivity of instruments over the period of 5 years. Detection limits of the trace gas analysers and uncertainties of the measurements should also be provided.

**Page 4, line 6:** Authors mention that quality assurance and quality control on the dataset were performed by PCD prior to receiving the data. What are these quality controls?

**Page 4, line 9:** What are the manual quality controls? What are the criteria for choosing unusual observations?

**Result and Discussion**

**Section 3.1:**

Authors have only provided maximum and average over the entire five-year period. Since they have continuous one hour time resolution dataset from 15 monitoring stations for a five year long period, authors should also include inter-annual variability and seasonal statistics at-least for different monitoring station types. Given the advantage of also having wind speed/ wind direction data, authors should consider comparing various airmass fetch regions for some monitoring sites. For ozone, it makes more

sense to separate daytime and night-time before reporting the average concentrations. The authors discuss extensively about 1-hour exceedance of ozone concentrations, but there is no description of how are these exceedance events calculated. One cannot compare the hourly average concentrations directly with the NAAQS. What about the ozone exceedance from 8-h standard? Bangkok air quality standard provides criteria for both 1-hour and 8-hour average ozone. 8-h average is intended to provide a better protection from long term ozone exposure.

**Section 3.2 Diurnal Variation of the Gaseous Species:**

Regional meteorology has strong influence on primary emission processes, production of secondary pollutant e.g. ozone and ambient concentrations of pollutants. I would recommend season wise analysis of diel variation of gaseous species. For example, the authors can refer to the work of Gaur et al. [2014] and Kumar et al. [2016]. This would also enable to identify the periods when ozone production is maximum during the year. Authors should also analyse, how does rate of formation of ozone from sunrise until it attains the peak daytime values changes at different sites and in different seasons. Authors could refer to the work of Naja and Lal [2002].

Gaur, A., Tripathi, S. N., Kanawade, V. P., Tare, V., and Shukla, S. P.: Four-year measurements of trace gases ($SO_2$, $NO_x$, CO, and $O_3$) at an urban location, Kanpur, in Northern India, Journal of Atmospheric Chemistry, 1-19, *10.1007/s10874-014-9295-8*, 2014.

Kumar, V., Sarkar, C., and Sinha, V.: Influence of post-harvest crop residue fires on surface ozone mixing ratios in the N.W. IGP analyzed using 2 years of continuous in situ trace gas measurements, J. Geophys. Res., 121, 3619–3633 *10.1002/2015JD024308*, 2016.

Naja, M., and Lal, S.: Surface ozone and precursor gases at Gadanki (13.5°N, 79.2°E),

a tropical rural site in India, Journal of Geophysical Research: Atmospheres, 107, *10.1029/2001jd000357*, 2002.

Authors should provide an explanation for why a second peak is not observed in the diel profiles of $SO_2$ at all sites. In line 20 of page 7, authors speculate that $SO_2$ is emitted by automotive diesel engine exhaust. If we observe the diel profile of NO from the BKK sites, a bimodal profile is observed which is attributed to traffic emissions. Moreover, even if we assume that manufacturing facilities point sources are the $SO_2$ contributors as mentioned in line 23 of page 11, their emission strength would not vary over the time scale of a day and a bimodal profile driven by boundary layer meteorology should be observed.

Similarly, authors should also provide an explanation for the relatively flatter diel profile of $NO_2$ at roadside sites.

**3.3 Interconversion between $O_3$, NO and $NO_2$ and Photochemical Reaction:**

I have major concerns again with this section. In line 23, authors mention "the photo-stationary state (PSS) is applied through all chemical reactions for $O_3$ formation during 10:00-16:00 LT". However, later in the section they assume photostationary state only between $O_3$, NO and $NO_2$. In polluted environments, $RO_2$ and $HO_2$ also oxidize NO to $NO_2$ and hence disturb the PSS of NO, $NO_2$ and $O_3$ [Mannschreck at al., 2004]. Hence the $j_1$ values calculated by only considering $O_3$, NO and $NO_2$ in the PSS would not be accurate.

Mannschreck, K., Gilge, S., Plass-Duelmer, C., Fricke, W., and Berresheim, H.: Assessment of the applicability of NO-NO2-O3 photostationary state to long-term measurements at the Hohenpeissenberg GAW Station, Germany, Atmos. Chem. Phys., 4, 1265-1277, 10.5194/acp-4-1265-2004, 2004.

Moreover, $j_1$ values are strongly dependent on incoming solar radiation and mentioning

an average over 10:00 L.T. until 16:00 L.T. will be oversimplification. In the moderately polluted environment, The photostationary state between $O_3$, NO and $NO_2$ is achieved within 60 s to 300 s during daytime [Trebs et al., 2012]. Authors should perform a calculation of $j_1$ at similar timescales.

Trebs, I., Mayol-Bracero, O. L., Pauliquevis, T., Kuhn, U., Sander, R., Ganzeveld, L., Meixner, F. X., Kesselmeier, J., Artaxo, P., and Andreae, M. O.: Impact of the Manaus urban plume on trace gas mixing ratios near the surface in the Amazon Basin: Implications for the $NO$-$NO_2$-$O_3$ photostationary state and peroxy radical levels, Journal of Geophysical Research: Atmospheres, 117, *10.1029/2011JD016386*, 2012.

I cannot understand, why the authors emphasize the calculated $k_3$ values. It depends on a single parameter which is temperature! Do the authors want to show that their temperature measurements are reasonable or their calculation is accurate?

Next, the authors are using $O_3$ measurements to estimate the $j_1$ values and again using $j_1$ to explain high $O_3$ concentration at some sites. This is cyclic.

Polynomial trend lines are used to investigate the interconversion between $O_3$, NO and $NO_2$. However, as seen from Figure 4, The fit is very poor for $O_3$ in all the three cases. So inference drawn using these fits would not be conclusive.

**Section 3.4**

What are the criteria for differentiation between episodes and non-episodes?

For the linear regression presented in this section, one can observe significant scatter around the fitted line. In some cases, (for example roadside sites, non-episode), one can clearly observe two different regions in the plots and a single linear fit over entire dataset cannot be justified.

For the delta $O_3$ analysis, how were the back trajectories calculated? How many trajectories per day and how many days backward trajectories at what height were calculated? Authors should also provide the number of days/hours when N-NE and S-SE wind directions respectively were observed. How was the agreement between local wind directions and the wind directions derived from NOAA HYSPLIT model?

Given the large scatter around average of $\sim$ 10 ppb delta $O_3$, the conclusion of local production is rather week for days with $O_3$ concentrations $>$ 80 ppb. The sentence structuring is poor and was difficult to follow. This also needs improvement. The conclusion regarding crossover points is drawn from polynomial regressions which have very poor fit parameters (and not even mentioned in the paper). The high $NO_x$ and low $NO_x$ regime should be calculated based on the ratio of $NO_x$ OH reactivity and VOC OH reactivity or using model calculated indicators (e.g. $CH_2O/NO_y$, $H_2O_2/HNO_3$ and $O_3/(NO_y–NO_x)$) as described by Kumar et al., [2011]. Classification based on cross over points are an oversimplification of the polynomial fits.

Kumar, R., Naja, M., Pfister, G. G., Barth, M. C., Wiedinmyer, C., and Brasseur, G. P.: Simulations over South Asia using the Weather Research and Forecasting model with Chemistry (WRF-Chem): chemistry evaluation and initial results, Geosci. Model Dev., 5, 619-648, *10.5194/gmd-5-619-2012*, 2012.

**Section 3.5**

**Page 11, Line 16:** A good correlation implies good correlation coefficient (r) for a linear regression and not necessarily a large value of slope. Authors' logic of having a high $CO/NO_x$ ratio (slope of fit) because of a better correlation between the two species emitted from point sources is difficult to follow.

The authors state that high $CO/NO_x$ and low $SO_2/NO_x$ ratio is characteristic of mobile sources. What are the values they referring to? Is there a threshold? What is the correlation coefficient of the liner regression between CO and $NO_x$? Such correlation plots

should at-least be provided in supplement. Since the authors have a great advantage of having the data from multiple receptor locations, they should use some statistical source apportionment models (for example, Positive Matrix Factorization (PMF) or the work by Garg and Sinha [2017])

Garg, S., and Sinha, B.: Determining the contribution of long-range transport, regional and local source areas, to PM10 mass loading in Hessen, Germany using a novel multi-receptor based statistical approach, Atmospheric Environment, 167, 566-575, *10.1016/j.atmosenv.2017.08.029*, 2017.

The authors have referred to the work of Parrish et al. [1991] for local source identification using $CO/NO_x$ ratio. However, longer-lived $NO_y$ should be used in place of $NO_x$. This method can be used for estimating the background concentration of a short-lived species by performing a lognormal regression with a long-lived species. Simply using the ratio of CO and $NO_x$ to conclude the dominance of mobile source over point sources or vice versa by performing a linear regression over entire dataset of a group of specific monitoring station type will be a wrong over-interpretation of these ratios. This is also evident from the $SO_2/NO_x$ ratios reported in Table 3. The $SO_2/NO_x$ values are very similar for all the types of sites and even higher for roadside sites as compared for suburban and BKK sites. Based on authors assertion, it should be minimum for roadside sites among the three categories.

**Section 3.5.2**

Why are wind rose plotted for separate wind directions? It is very confusing. Authors should show a wind rose showing the fraction of wind coming from all the directions for $O_3$ concentrations higher than 100 ppb. Higher fraction of wind from a particular direction would automatically point out major contribution from a particular wind direction. A polar plot with wind speed as radius axis, wind direction as angle and markers coloured according to observed $O_3$ concentration could also be an alternative plot. How does

the wind rose look like for periods with $O_3$ concentration less than 100 ppb? If it is different from the ones for higher concentration, this would make the conclusion of higher ozone production from a particular wind direction stronger.

**Section 3.6**

Authors need to describe the calculation of air quality index. If they have used the simple hourly average $O_3$ concentration for calculation of AQI, then it is wrong. For calculation of hourly air quality index, $O_3$ concentration for a given hour should be taken as the average for the previous 4 hours, current hour and next 3 hours. However, it is recommended to consider 8 hour AQI as mentioned previously in the review.

**Technical comments:**

**Page 1 Line 27 – page 2 line 2; Page 2, lines 11-16** These are better suited for site description.

**Page 2, line 17:** NO is Nitric oxide and not nitrogen oxide. Nitrogen oxides refer to the family of oxides of nitrogen.

**Page 2, Line 20:** What is the basis of the statement "Moreover, BMR experiences primarily $O_3$ exceedances amongst all the other gaseous criteria pollutants."

**Page 4, Line 4:** Figure 1 should be mentioned earlier in the section.

**Page 4, Line 20:** What are the "equivalent instruments"?

**Page 5, Line 3:** Is the measurement period 2012-2014 or 2010-2014?

**Page 5, Line 7:** What is the hourly "standard"?

**Page 6, line 5:** Authors mention " VOCs concentrations were measured periodically

only at one monitoring station limiting 5 its usefulness as part of this study ". However, Zhang et al. [2002] have reported $CH_4$ and NMVOC data from 10 out of 13 monitoring stations from BMR. Did the stations stopped monitoring $CH_4$ and NMVOCs?

**Page 6, Line 12 and Figure 3:** What is the explanation for a rather flat diel profile of $NO_2$ at roadside sites. Roadside sites are influenced maximum by traffic emissions, and one would expect a bimodal shape of diel profile.

**Page 8,** The rate constants and photolysis frequencies should be expressed in $cm^3molecule^{-1}s^{-1}$ and $s^{-1}$ respectively.

**Page 9, line 9,** The titration of $O_3$ with NO will not effectively reduce the $O_3$ concentrations. Such a titration process with produce $NO_2$ which will again photolyze in the daytime and produce $O_3$.

**Page 12, line 2.** Please check the lifetime of $O_3$. It should be few days (if not few weeks) in urban atmosphere.

**Page 12, Line 15- Page 12, Line 5:** Such description is better suited for introduction.

**Page 13 Line 18 to page 19 line 2:** Such discussion is well suited for outlook after proper restructuring.

**Figure 1:** I would recommend showing airmass back trajectories rather than showing wind directions with two indicator arrows.

**Figure 2:** Ambient variability should also be shown along with average values. Authors should also show the concentrations of NO, in addition to CO, $SO_2$, $O_3$, and $NO_2$. The colour for year 2010 and 2011 look same in panel "e".

**Figure 3**: Quality of figure should be improved (overall presentation, axis labels and legends). Ambient variability (as interquartile range or 1 $\sigma$ standard deviation) should also be shown in addition to the average values. This should be done for other figures also in the paper.

**Figure 6:** The minimum wind speed bin should be 0.5 – 2.0 (not 201). Please use the same radius scale for the wind rose plots.

**Table 1:** Please refer to the comment for section 3.3.

**Table 3:** Authors should also include the $SO_2$/$NO_x$ ratio reported from various cities in India for mobile sources as reported by Mallik and Lal, 2014.

Mallik, C., and Lal, S.: Seasonal characteristics of $SO_2$, $NO_2$, and CO emissions in and around the Indo-Gangetic Plain, Environmental Monitoring and Assessment, 186, 1295-1310, *10.1007/s10661-013-3458-y*, 2014.
* * *

---

## Referee Comment (RC2) · D. Mikel (Referee) · 3 May 2018

I have finished my review of this proposed paper. Overall, the article is well written and examines the interaction of Ozone with NOx regime. The analysis was well done. However, some of the text needs either clarification or at least additional discussion. Below please find my specific comments.

1. In line 12, the statement is made, "On average, the number of hourly O3 exceedences ranged from 1 - 60 hours a year." This line is confusing. The overall average should be a value, not a range. If you wish to express it as a range, then do it by year, such as 2010 that average was XX hours, 2011, the average was XX hours. This range of 1-60 hours makes no sense.

[Figure]

2. When you express a range (this applies throughout, do not mix the units and values. In Section 2, Methodology line 12 you state the temperature is ($\sim$35C - 40C). This appears to read that it ranges from 35 degrees to - (minus) 40 degrees. Do this instead: (35 - 40 C).

3. Same section line 21, it states, " It is assumed that the monitoring sites used were representative of BMR specific patterns and trends." I think it goes without stating this that the professionals at the PCD would have done this and this does not need to be stated, but you would hope you would infer this. Remove this statement.

4. Same section, line 27 -29, you list the sites (19T, 20T, etc...) which mean absolutely nothing to the reader then you state in line 29 that the figure shows these. The statement that mentions the figure should be the first line to the paragraph, not the last line. Move this line to the front so the reader can go get the figure look at it while you read the information.

5. Section 2.2, line 2 you mention wind speed and direction. Is this average or vector data? Please state. This is important when calculating direction from which winds are blowing.

6. Same section, line 10, it is mentioned that equipment and monitoring station are calibrated every year. This is vague and could cast a shadow on validity of data. does this mean that this is done only once per year? Pollution instruments and met, or only met instruments. I am sure the PCD does calibrations more often than once annually. Please clarify this statement.

7. In Section 3.3 line 24, you use the term "atmospheric boundary layer." Is this the same as planetary boundary layer that was used previously? If it is the same term, then be consistent. If it isn't then please explain what this term means on how it differs from the PBL.

8. Page 8, line 11, Please explain why the ratios of NO2 and NO show significant

difference. You make the statement but you don't say why. this is an important claim that you make in this paper/

9. Page 9, line 9, you state, "In conclusion, the titration of O3 and NO is perhaps one of the most important processes..." Please elaborate about why this is so important.

Page 11, line 5, you state, "However, a negative delta O3 may be negative. However, it appears that the data doesn't support this in the paragraph. Why is this statement made?

---

## Author Comment (AC1) · 5 Jun 2018

General Comment: Pornpan Uttamang et al. have presented observations of CO, NOx, SO2 and O3 from 15 monitoring sites at understudied Bangkok Metropolitan Region (BMR) for a fiveyear- long period from 2010-2014. Background pertaining to the air-quality in terms of PM and O3 exceedance events in the BMR is provided. However, the authors do not mention the knowledge gap or scientific question that they want to address from this study. I have major concerns with the paper which include description of analytical methods and discussion about quality control (calibration and sampling protocols, filter criteria) of dataset used. The statistical analysis is also weak which mostly covers average/maximum over the entire study periods, without going into details of specific seasons, inter-annual trends and pin-pointing the season-specific

emission sources /formation processes and removal processes of the pollutants. The conclusions are drawn either from the regression lines having poor fit parameters or oversimplification of methods for source identification available in the peer-reviewed literature. The manuscript needs to address the major concerns (highlighted in specific comments) before it can be considered further. After performing the analysis suggested in the specific comments, corrections and restructuring the paper, the scientific outcome might be significantly different from the present version and should be considered as a new publication.

Authors' response: Thank you. The last paragraph of the Introduction (line 17 – 24) succinctly provides both the knowledge gap and scientific question being addressed. We have now made the statistical analysis more robust based on reviewer' suggestions.

Specific Comments: Title: Authors might consider making the title of the paper more specific. Authors assess CO, NOx, SO2 and O3 air pollution and not overall air pollution in general. Authors' response: Thank you. We have now modified the title to "Assessment of Gaseous Pollutants in Bangkok Metropolitan Region, Thailand"

Introduction: The authors have included a description of auto-mobile fleet and manufacturing industries in the introduction which should rather be a part of the site description. The introduction is poorly structured. Authors should include a brief literature review of the previous works from BMR, outlook from these studies and what are the knowledge gaps they want to address from this paper. The authors should also mention, why they have chosen to study CO, NOx, SO2 and O3. At-least a line each about their importance regarding atmospheric chemistry and air quality should be present. The authors have referred to Zhang and Oanh, [2002] for the site description. However, the findings there should also be mentioned in the introduction, as Zhang and Oanh, [2002] have analyzed monthly and diel variation, O3 exceedances, drivers for high ozone episodes and relationship of ozone production with NOx/NMHC ratio. These are quite relevant for the present study. Similarly, the work of Pochanart et al., [2001] should be highlighted in the introduction. I found few other studies (mentioned below) which are relevant to the present work and should be highlighted in the introduction. There might be several more! Jinsart, W., Tamura, K., Loetkamonwit, S., Thepanondh, S., Karita, K., and Yano, E.: Roadside Particulate Air Pollution in Bangkok, Journal of the Air & Waste Management Association, 52, 1102-1110, 10.1080/10473289.2002.10470845, 2002. Suthawaree, J., Tajima, Y., Khunchornyakong, A., Kato, S., Sharp, A., and Kajii, Y.: Identification of volatile organic compounds in suburban Bangkok, Thailand and their potential for ozone formation, Atmospheric Research, 104-105, 245-254, 10.1016/j.atmosres.2011.10.019, 2012

Authors' response: Thank you. We have now modified the Introduction to include the discussion in the references provided by the reviewer.

"1. Introduction Over the last three decades, Thailand's rapid industrialization and urbanization has led to an increase in global economic prowess (World Bank, 2018a). A majority of the country's development has occurred within and around Bangkok (BKK) (13ïĆř45' N and 100ïĆř85' E), the capital city of Thailand and Bangkok Metropolitan Region (BMR). The BKK is comprised of the five adjacent provinces of BKK (World Bank, 2018a and 2018b). The increase in emissions is due to accelerated growth combined with high photochemical activity, strong solar insolation, high temperatures and high humidity (Kumar et al., 2012). BMR, with these conditions, has begun to experience air quality degradation, in particular, enhanced secondary pollutants. Since 1995, BKK has experienced exceedances in Thailand NAAQs for particulate matter (PM) and ozone (O3) (PCD, 2015). The correlation between BMR air pollution and public health has been observed in several published studies. Ruchirawat et al. (2007) reported the children who lived in BKK are exposed to high levels of carcinogenic air pollutants which may cause an elevated cancer risk. Buadong et al. (2009) reported the exposure to elevated PM and O3 during the previous day, in elderly patients ($\geq$ 65 years), is associated with increasing the number of daily hospital visits for cardiovascular diseases. Jinsart et al. (2002, 2012) reported polices and drivers in BKK tended to expose higher

level of PM concentrations compared with the general environment, in which the concentrations of PM were already high. Role of atmospheric processes in elevated O3 in Thailand were reported in several studies. Long-range transport played an important role in increasing O3 concentration in Thailand. Generally, long-range transports from the Asia continental contained higher O3 concentrations compared with long-range transports from the Indian ocean (Pochanart et al., 2001). In BMR, local emission and regional transport were the major contributors to high O3 concentrations and seasonal fluctuations of O3, respectively (Zhang and Oahn, 2002). In suburban areas of BKK, volatile organic compounds (VOCs) tended to be a potential factor to enhance O3 concentration (Suthawaree et al., 2012). The availability and analysis of multi-year measurements of such gaseous pollutants in the BMR will improve our understanding of how they contribute to the air quality of this area. As a major metropolitan area, BMR is dominated by mobile emissions sources, which contributes to the emissions of CO and NOx, precursors of ozone formation. The emissions from industrial activities also dominates the BMR metropolitan area and contributes to the emissions of sulfur dioxide (SO2) and the formation of particulate matter. In this study, diurnal variations, seasonal variations and inter-annual trends of gaseous pollutants including carbon monoxide (CO), nitric oxide (NO), nitrogen dioxide (NO2), SO2 and O3 during 2010 to 2014, in BMR, have been analysed. Chemical and physical processes associated with high O3 concentrations have been investigated. Since the concentrations of nitrogen oxide (NOx) was measured at most of the monitoring station, therefore, O3 precursors in this study is referred to NOx. The photochemical reaction was investigated during the photostationary state. The effects of local emission and regional contributions of Ox are presented. The severity of air pollution concentrations in BMR on human health are assessed by performing Air Quality Index (AQI)."

Page 2, Line 23: Authors state "possible emission sources of pollutants that associate with O3 formation are identified". However, such identification is not discussed in the manuscript. Authors have only used the ratio of CO/NOx and SO2/NOx to identify whether the emission sources are mobile or point in nature. The method itself has an

inherent limitation which is mentioned later in the specific comment for the section.

Authors' response: Thank you. We have now modified the Introduction. Please refer to our discussion in the comment "Section 3.5 Page 11, Line 16" below.

Methodology: The exact measurement period should be mentioned in this section. This paper discusses a five-year-long measurement period and shows data over 15 different measurement stations and authors should provide a time-line for data availability for each station. Authors' response: Thank you. We have now provided the measurement period in the modified manuscript and provided a time-line for data availability for each station as part of the supplement material.

Page 3, line 22: What is the basis of the assumption that monitoring sites used were representative of BMR specific patterns and trends?

Authors' response: Thank you. Based on the reviewer, we have modified the manuscript by removing the sentence.

Data Collection and Data Analysis: I have major concerns with this section. Authors did not provide any sampling details. The trace gas analysers for CO, NOx, SO2 and O3 are known to have drifts with time. Authors mention that equipment and monitoring stations are calibrated every year. This is not enough. There should be frequent zero drift check for CO (at-least daily) and for NOx, SO2 and O3 (at-least once a week). The linearity of the detection should also be checked with calibration experiments performed at-least once a month. The authors did not provide any information about the drift in the sensitivity of instruments over the period of 5 years. Detection limits of the trace gas analysers and uncertainties of the measurements should also be provided.

Authors' response: Thank you. As indicated in the manuscript, the data were collected, and after QA/QC, were provided by the Pollution Control Department (PCD), Thailand. Data loggers are calibrated/ checked at least every 15 days. Air inlets are cleaned at least every 15 days. Equipment is single-point calibrated and multi-point calibrated at

least every 15 days and at least every 3 months. Monitoring stations and equipment are audited by external auditors every year. We have modified our manuscript to make a clarification.

"Quality assurance and quality control on the data set were performed by PCD prior to receiving the data. Hourly observations of the gaseous pollutants and meteorological parameters were automatically collected with auto calibration at the monitoring stations. Manual quality control was performed when unusual observations were found. External audit of the equipment and monitoring stations were done every year. "

Page 4, line 6: Authors mention that quality assurance and quality control on the dataset were performed by PCD prior to receiving the data. What are these quality controls? Authors' response: Thank you. QA/AC protocols are published in the PCD, Thailand government document.

Page 4, line 9: What are the manual quality controls? What are the criteria for choosing unusual observations? Authors' response: Thank you. QA/AC protocols are published in the PCD, Thailand government document. However, we did not provide any additional guidelines for data collection.

Result and Discussion Section 3.1: Authors have only provided maximum and average over the entire five-year period. Since they have continuous one hour time resolution dataset from 15 monitoring stations for a five year long period, authors should also include inter-annual variability and seasonal statistics at-least for different monitoring station types. Given the advantage of also having wind speed/ wind direction data, authors should consider comparing various airmass fetch regions for some monitoring sites. For ozone, it makes more sense to separate daytime and night-time before reporting the average concentrations. The authors discuss extensively about 1-hour exceedance of ozone concentrations, but there is no description of how are these exceedance events calculated. One cannot compare the hourly average concentrations directly with the NAAQS. What about the ozone exceedance from 8-h standard? Bangkok

air quality standard provides criteria for both 1-hour and 8-hour average ozone. 8-h average is intended to provide a better protection from long term ozone exposure.

Authors' response: Thank you. We have now provided inter-annual plots and seasonal variation plots for gaseous criteria pollutants from the monitoring stations. In the plots, we have averaged the data for each monitoring type station (from BKK sites, roadside sites and BKK suburb sites). This figure is now included in the supplementary material. However, the discussion associated with this information is now included in the manuscript. Moreover, we have now included the average concentrations of O3 during daytime (6:00 AM to 6:00 PM) and during nighttime (6:00 PM to 6:00 AM) in the modified manuscript.

"3.1 Status of Pollution in BMR during 2010 to 2014 Figure 2 a) to e) shows the maximum and average concentration of gaseous pollutants, during 2010 to 2014 from the 15 monitoring stations. These concentrations are compared with the hourly NAAQs of Thailand (NAAQs of Thailand for hourly CO, NO2, SO2 and O3 are 30 ppm, 170 ppb, 300 ppb and 100 ppb, respectively (PCD, 2018)). Since, NO is not a criteria pollutant, only the maximum and average concentrations are presented. During the study period, the maximum concentrations of CO, NO2 and SO2 were mostly in their hourly standards (an exceedance of NO2 was found at 52T monitoring station, during 2013). However, the maximum concentrations of O3 exceeded its standard. Elevated CO, NO, and NO2 concentrations were frequently observed at roadside sites than other sites. The average concentrations of CO, NO, and NO2, at roadside sites, were $\sim 1.0 \pm 0.1$ ppm, $\sim 60.5 \pm 42.7$ ppb, and $\sim 30.9 \pm 8.1$ ppb, respectively. Elevated SO2 were commonly observed at BKK suburb sites than other sites. The average concentrations of SO2 at BKK suburb sites were $\sim 4.0 \pm 2.3$ ppb. The average concentrations of O3 during daytime (6:00 to 18:00 LT) over BKK sites, roadside sites and BKK suburb sites were $\sim 24.4 \pm 13.5$ ppb, $\sim 18.2 \pm 12.3$ ppb and $\sim 27.7 \pm 14.7$ ppb, and those values during night-time (18:00 to 6:00 LT) were $\sim 11.3 \pm 3.3$ ppb, $\sim 9.1 \pm 4.9$ ppb and $\sim 14.2 \pm 5.4$ ppb, respectively. The 24-hour average O3 concentrations were highest at BKK suburb sites (∼22.0±19.8 ppb) and following by BKK sites (17.9±16.9 ppb) and roadside sites (13.3±12.7 ppb). The maximum and average of gaseous pollutants the three monitoring types are provided in Table I, supplement material. The seasonal variations of the gaseous pollutants reveal that, in general, elevated concentrations were observed during dry season and those decreased during wet season (Figure II, supplement material). Inter-annual variations of the gaseous pollutants reveal that, while the concentrations of CO, NO2 and SO2 decreased or remained constant, the concentration of O3 tended to increase during the study period (Figure III, supplement material). An O3 exceedances was recorded when an hourly concentration of O3 was greater than 100 ppb (hourly O3 standard). Figure 2 f) to g) illustrate the number of hourly O3 exceedances, which they are shown by locations and by seasons, respectively. The hourly O3 exceedances at BKK suburb sites were more frequently observed than those at the other sites. The average number of hourly O3 exceedances was ∼16 hours year-1 at BKK sites, ∼9 hours year-1 at roadside sites and ∼43 hours year-1 at BKK suburb sites. The hourly O3 exceedances were commonly observed during dry season than during the transitional period between the seasons (May) and rarely observed during wet season. "

With regards to segregating wind direction data we performed a more robust back trajectory analysis. Moreover, we provided wind-rose plots for each of the monitoring stations and discussed it in the manuscript.

The National Ambient Air Quality Standards of Thailand provides hourly and 8-hour average standards of O3 (0.10 ppm and 0.07 ppm, respectively). In this study, we compared the hourly concentrations of O3 with the hourly O3 standard in order to examine number of O3 exceedances. To study the effects of O3 on human health, we applied Air Quality Index (AQI) of O3 instead of using the O3 exceedance from 8-hour standard, which we believe that, using AQI of O3 will provide more advantages than using the O3 exceedance from 8-hour standard. Since AQI of O3 is categorized into six categories, with four of the six categories providing the information of the severity

of high O3 concentrations on human health, from sensitive groups to healthy people; therefore, applying AQI for O3 will provide better information for air quality management.

Section 3.2 Diurnal Variation of the Gaseous Species: Regional meteorology has strong influence on primary emission processes, production of secondary pollutant e.g. ozone and ambient concentrations of pollutants. I would recommend season wise analysis of diel variation of gaseous species. For example, the authors can refer to the work of Gaur et al. [2014] and Kumar et al. [2016]. This would also enable to identify the periods when ozone production is maximum during the year. Authors should also analyse, how does rate of formation of ozone from sunrise until it attains the peak daytime values changes at different sites and in different seasons. Authors could refer to the work of Naja and Lal [2002]. Gaur, A., Tripathi, S. N., Kanawade, V. P., Tare, V., and Shukla, S. P.: Four-year measurements of trace gases (SO2, NOx, CO, and O3) at an urban location, Kanpur, in Northern India, Journal of Atmospheric Chemistry, 1-19, 10.1007/s10874-014-9295-8, 2014. Kumar, V., Sarkar, C., and Sinha, V.: Influence of post-harvest crop residue fires on surface ozone mixing ratios in the N.W. IGP analyzed using 2 years of continuous in situ trace gas measurements, J. Geophys. Res., 121, 3619–3633 10.1002/2015JD024308, 2016. Naja, M., and Lal, S.: Surface ozone and precursor gases at Gadanki (13.5_N, 79.2_E), a tropical rural site in India, Journal of Geophysical Research: Atmospheres, 107, 10.1029/2001jd000357, 2002. Authors should provide an explanation for why a second peak is not observed in the diel profiles of SO2 at all sites. In line 20 of page 7, authors speculate that SO2 is emitted by automotive diesel engine exhaust. If we observe the diel profile of NO from the BKK sites, a bimodal profile is observed which is attributed to traffic emissions. Moreover, even if we assume that manufacturing facilities point sources are the SO2 contributors as mentioned in line 23 of page 11, their emission strength would not vary over the time scale of a day and a bimodal profile driven by boundary layer meteorology should be observed. Similarly, authors should also provide an explanation for the relatively flatter diel profile of NO2 at roadside sites.

Authors' response: Thank you. We have now provided season wise analysis of diel variation of gaseous species for the three monitoring station types in the supplementary material and a discussion is provided in the manuscript.

[revised manuscript text omitted]

to 9:00 LT). The second peak of those occurred ∼3 to 5 hours after the evening traffic rush hour (16:00 to 18:00 LT) (Leong et al., 2002), due to a combination of pollutants emissions and collapse of the planetary boundary layer (weak turbulence and diffusion) during this time. The diurnal variations of SO2 show a bimodal pattern with the first- and the second-peak of SO2 occurred ∼8:00 LT and 21:00 LT, respectively. The concentrations of SO2 at the first- and the second-peak were ∼3 ppb (both peaks) at BKK sites, ∼3 ppb (both peaks) at roadside sites, and ∼6 ppb ∼3 ppb at BKK suburb sites. At the roadside sites, the peaks are more obvious than the other sites. The result indicates that at this monitoring station type, SO2 is primarily influenced by emissions from vehicle exhaust using high sulfur content fuel (Henschel et al. 2013). It is noteworthy that BKK has a large diesel engine fleet (an estimated 25 % of registered vehicles) (DLT, 2015). The diesel fuel contains ∼0.035 %wt Sulphur (DOEB, 2017). Season wise of the diurnal variations are provided in Figure IV, supplement material. Figure 4 a) to c) shows diurnal variations of rate of change of O3 concentration (Δ[O3]/dt) during dry season (local summer and local winter) and wet season at the three monitoring station types (the data has been averaged for each monitoring station type to capture the rate of change of O3 concentration characteristics). The diurnal variations of Δ[O3]/dt is a combination of O3 chemistry and meteorology. In general, Δ[O3]/dt during wet season were lower than those during dry season. However, during local winter, the rates of change O3 concentration were the highest. The Δ[O3]/dt at the three monitoring station types, during 10:00 to 11:00 LT, were 4.5 to 7.0 ppb hr-1 during wet season, 6.7 to 7.5 ppb hr-1 during local summer, and 5.7 to 9.2 ppb hr-1 during local winter. The Δ[O3]/dt became negative during 14:00 to 15:00 LT. As expected, the rate of change of O3 concentration was nearly constant during nighttime. Rapid changes in the mixing height and solar insolation during morning increases Δ[O3]/dt. After sunset, the formation of O3 is inhibited and the planetary boundary layer becomes more stable resulting in O3 reduction through chemical reactions (for example, the oxidation of O3 by NOx) and physical processes (for example, dry deposition to the earth surface) (Naja and Lal, 2002). " We have also provided the rate of change of O3 concentration (Naja

and Lal, 2002) during the three seasons and in the three monitoring station types with explanation in the modified manuscript.

"Figure 4 a) to c) shows diurnal variations of rate of change of O3 concentration ($\Delta$[O3]/dt) during dry season (local summer and local winter) and wet season at the three monitoring station types (the data has been averaged for each monitoring station type to capture the rate of change of O3 concentration characteristics). The diurnal variations of $\Delta$[O3]/dt is a combination of O3 chemistry and meteorology. In general, $\Delta$[O3]/dt during wet season were lower than those during dry season. However, during local winter, the rates of change O3 concentration were the highest. The $\Delta$[O3]/dt at the three monitoring station types, during 10:00 to 11:00 LT, were 4.5 to 7.0 ppb hr-1 during wet season, 6.7 to 7.5 ppb hr-1 during local summer, and 5.7 to 9.2 ppb hr-1 during local winter. The $\Delta$[O3]/dt became negative during 14:00 to 15:00 LT. As expected, the rate of change of O3 concentration was nearly constant during nighttime. Rapid changes in the mixing height and solar insolation during morning increases $\Delta$[O3]/dt. After sunset, the formation of O3 is inhibited and the planetary boundary layer becomes more stable resulting in O3 reduction through chemical reactions (for example, the oxidation of O3 by NOx) and physical processes (for example, dry deposition to the earth surface) (Naja and Lal, 2002)"

For the diurnal variations of SO2, in the manuscript, we explained that "...The concentrations of SO2 increase again in the afternoon and reach a second-peak around 21:00 LT over roadside sites. Over BKK sites and BKK suburb sites, the concentrations of SO2 are nearly constant after 19:00 LT..." which the second peak of SO2 were observed over three monitoring station types, but the magnitude of the concentrations of SO2 over BKK sites and BKK suburb sites were small. For the diurnal variation of NO2, at the roadside sites also showed a bimodal distribution, but flatter than those at other sites. However, we have now included the clarification and explanation in the manuscript.

"The diurnal variations of SO2 show a bimodal pattern with the first- and the secondpeak of SO2 occurred ~8:00 LT and 21:00 LT, respectively. The concentrations of SO2 at the first- and the second-peak were ~3 ppb (both peaks) at BKK sites, ~3 ppb (both peaks) at roadside sites, and ~6 ppb ~3 ppb at BKK suburb sites. At the roadside sites, the peaks are more obvious than the other sites. The result indicates that at this monitoring station type, SO2 is primarily influenced by emissions from vehicle exhaust using high sulfur content fuel (Henschel et al. 2013). It is noteworthy that BKK has a large diesel engine fleet (an estimated 25 % of registered vehicles) (DLT, 2015). The diesel fuel contains ~0.035 %wt Sulphur (DOEB, 2017). Season wise of the diurnal variations are provided in Figure IV, supplement material. "

"The concentrations of NO2 at the first- and the second-peak were ~23 ppb and ~28 ppb at BKK sites, ~33 ppb and ~37 ppb at roadside sites, and ~20 ppb and ~22 ppb at BKK suburb sites. Even the diurnal variations of NOx show a bimodal pattern, at roadside sites, the pattern was flatter than at other sites. The flatter pattern of NOx at roadside sites reveals that this monitoring station type was affected by high concentration of NOx all day. "

Section 3.3 Interconversion between O3, NO and NO2 and Photochemical Reaction: I have major concerns again with this section. In line 23, authors mention "the photo-stationary state (PSS) is applied through all chemical reactions for O3 formation during 10:00-16:00 LT". However, later in the section they assume photostationary state only between O3, NO and NO2. In polluted environments, RO2 and HO2 also oxidize NO to NO2 and hence disturb the PSS of NO, NO2 and O3 [Mannschreck at al., 2004]. Hence the j1 values calculated by only considering O3, NO and NO2 in the PSS would not be accurate. Mannschreck, K., Gilge, S., Plass-Duelmer, C., Fricke, W., and Berresheim, H.: Assessment of the applicability of NO-NO2-O3 photostationary state to long-term measurements at the Hohenpeissenberg GAW Station, Germany, Atmos. Chem. Phys., 4, 1265-1277, 10.5194/acp-4 1265-2004, 2004. Moreover, j1 values are strongly dependent on incoming solar radiation and mentioning an average over 10:00 L.T. until 16:00 L.T. will be oversimplification. In the moderately polluted environment,

the photostationary state between O3, NO and NO2 is achieved within 60 s to 300 s during daytime [Trebs et al., 2012]. Authors should perform a calculation of j1 at similar timescales. Trebs, I., Mayol-Bracero, O. L., Pauliquevis, T., Kuhn, U., Sander, R., Ganzeveld, L., Meixner, F. X., Kesselmeier, J., Artaxo, P., and Andreae, M. O.: Impact of the Manaus urban plume on trace gas mixing ratios near the surface in the Amazon Basin: Implications for the NO-NO2-O3 photostationary state and peroxy radical levels, Journal of Geophysical Research: Atmospheres, 117, 10.1029/2011JD016386, 2012. I cannot understand, why the authors emphasize the calculated k3 values. It depends on a single parameter which is temperature! Do the authors want to show that their temperature measurements are reasonable or their calculation is accurate? Next, the authors are using O3 measurements to estimate the j1 values and again using j1 to explain high O3 concentration at some sites. This is cyclic. Polynomial trend lines are used to investigate the interconversion between O3, NO and NO2. However, as seen from Figure 4, The fit is very poor for O3 in all the three cases. So inference drawn using these fits would not be conclusive.

Authors' response: Thank you. In our study, we evaluated the relationship between O3 with the gaseous criteria pollutants for the NAAQs of Thailand. The assumption of the photostationary state (PSS) (ÏŢ = 1), therefore, was applied through the chemical reactions of O3 and NOx only.

Mannschreck et al., 2004, reviewed the PSS parameter (ÏŢ) as:

Where j was the photolysis rate of NO2, and k was the rate of the chemical reaction of NO and O3. In the Mannschreck et al., 2004, when ÏŢ was equal to 1, then other chemical reactions converting NO to NO2 and local emissions of either compound were negligible. However, these cases were rare and were limited to very polluted conditions. On the other hand, peroxy radicals (RO2) played an important role to contribute to additional NO and NO2, under clean or moderately polluted conditions. In the study of Mannschreck et al., (2004), the measurement was performed in a rural site, generally, the site was affected by relatively clean air masses (yearly average of NOx was

below 3.5 ppb). The site was surrounded by forests (70%, mostly coniferous) and agri-cultural pastures (30%). The distance to the nearest urban and major industrial areas was about 80 km. Furthermore, the study mentioned that "for high NOx concentrations the levels of peroxy radicals should approach zero, since the sink for RO2 increases with increasing NO and since OH as a precursor for RO2 as well as RO species are re-moved via reaction with NO2". Therefore, we believe that the assumption of PSS holds for our study region (e.g. average of hourly concentration of NOx at BKK sites, roadside sites, and BKK suburb sites were ∼30 ppb, ∼88 ppb and ∼21 ppb, respectively. These NOx values are far in excess of rural/semi-rural values in the study of Mannschreck et al., (2004)). With regards to the calculation of j1, it is strongly dependent on incoming solar radiation and on other variables (i.e. the following equation):

j_(q,p)=∫ _⊕∞âŰŠãĂŰãĂŰ4πIãĂŮ_(p,λ) b_(a,g,q,λ,T) Y_(q,p,λ,T) dλãĂŮ

Where ãĂŰ4πIãĂŮ_(p,λ) = Actinic flux b_(a,g,q,λ,T) = Average absorption cross sec-tion Y_(q,p,λ,T) = Average quantum yield

However, these variables were not measured in our study at the monitoring stations. With regards to the calculation of k3, our intention is not to emphasize the k3 calculation to show that our temperature measurements were reasonable, but rather to calculate j1. However, we have now modified our manuscript and removed using j1 values to explain O3 concentration. With regards to the polynomial trend lines, we have now modified the plots by including histogram of the concentrations of O3, NO and NO2 to present data distribution of these species. Generally, most of the records are in low to middle concentration bins. (Fig. 5)

Section 3.4: What are the criteria for differentiation between episodes and non-episodes? For the linear regression presented in this section, one can observe sig-nificant scatter around the fitted line. In some cases, (for example roadside sites, non-episode), one can clearly observe two different regions in the plots and a single linear fit over entire dataset cannot be justified. For the delta O3 analysis, how were the back

trajectories calculated? How many trajectories per day and how many days backward trajectories at what height were calculated? Authors should also provide the number of days/hours when N-NE and S-SE wind directions respectively were observed. How was the agreement between local wind directions and the wind directions derived from NOAA HYSPLIT model? Given the large scatter around average of 10 ppb delta O3, the conclusion of local production is rather week for days with O3 concentrations > 80 ppb. The sentence structuring is poor and was difficult to follow. This also needs improvement. The conclusion regarding crossover points is drawn from polynomial regressions which have very poor fit parameters (and not even mentioned in the paper). The high NOx and low NOx regime should be calculated based on the ratio of NOx OH reactivity and VOC OH reactivity or using model calculated indicators (e.g. CH2O/NOy, H2O2/HNO3 and O3/(NOy–NOx)) as described by Kumar et al., [2011]. Classification based on cross over points are an oversimplification of the polynomial fits. Kumar, R., Naja, M., Pfister, G. G., Barth, M. C., Wiedinmyer, C., and Brasseur, G. P.: Simulations over South Asia using the Weather Research and Forecasting model with Chemistry (WRF-Chem): chemistry evaluation and initial results, Geosci. Model Dev., 5, 619-648, 10.5194/gmd-5-619-2012, 2012.

Authors' response: Thank you. We had explained that an O3 episode was identified when hourly O3 concentrations were greater than 100 ppb (the O3 NAAQs for Thailand).

With regards to the linear regression, we presented "the estimation" of local and regional contributions of Ox. Furthermore, we also compared the result from our study to results from other studies (using similar linear regression method).

For the two different observed NOx regions at roadside sites, we provided in our discussion the following "It is noteworthy that the pattern of the local and regional contributions at roadside sites during non-episode period is composed of two NOx concentration regimes. The low NOx regime (NOx < 60 ppb) resembles the local and regional contributions during non-episode over BKK suburb sites. The high NOx regime (NOx

> 60 ppb) may represent typical characteristic of air quality near roads".

To estimate the local and regional contribution by plotting Ox against NOx were reported in several published studied, for example Clapp and Jenkin (2001), Aneja et al., (2001), Mazzeo et al., (2005), Tang et al., (2009), Notario et al., (2012), Rasheed et al., (2014), Tiwari et al., (2015). These studies provide similar plots to our study. All these references are cited in the manuscript.

Clapp, L. J. and Jenkin, M. E.: Analysis of the relationship between ambient levels of O3, NO2 and NO as a function of NOx in the UK, Atmospheric Environment, 35(36), 6391- 6405, doi:10.1016/S1352-2310(01)00378-8, 2001. Aneja, V. P., Agarwal, A., Roelle, P. A., Phillips, S. B., Tong, Q., Watkins, N., and Yablonsky, R.: Measurements and Analysis of Criteria Pollutants in New Delhi, India, Environment International, 27, 35-42, doi:10.1016/s0160-4120(01)00051-4, 2001. Mazzeoa, N. A., Venegasa, L. E. and Chorenc, H: Analysis of NO, NO2, O3 and NOx concentrations measured at a green area of Buenos Aires City during wintertime, Atmospheric Environment, 39, 3055–3068, doi:10.1016/j.atmosenv.2005.01.029, 2005. Tang, G., Li, X., Wang, Y., Xin, J., and Ren, X.: Surface ozone trend details and interpretations in Beijing, 2001–2006, Atmos. Chem. Phys., 9, 8813–8823, 2009. Notario, A., Bravo, I., Adame, J. A., Díaz-de-Mera, Y., Aranda, A., Rodríguez, A., and Rodríguez, D.: Analysis of NO, NO2, NOx, O3 and oxidant (Ox = O3+NO2) levels measured in a metropolitan area in the southwest Iberian Peninsula, Atmospheric Research, 104-105, 217-226, doi:10.1016/j.atmosres.2011.10.008, 2012. Rasheed, A., Aneja, V. P., Aiyyer, A., and Rafique, U.: Measurements and analysis of air quality in Islamabad, Pakistan, Earth's Future, 2, 303-314, doi:10.1002/2013EF000174, 2014. Tiwari, S., Dahiya, A., and Kumar, N.: Investigation into relationships among NO, NO2, NOx, O3, and CO at an urban background site in Delhi, India, Atmospheric Research, 157, 119-126, doi:10.1016/j.atmosres.2015.01.008, 2015. With regards to the delta O3 analysis, back trajectory was determined when hourly concentration of O3 > 80 ppb was observed either at 27T or 20T sites. By performing the backward trajectories using the NOAA

HYSPLIT model, we identified the wind directions that related with high O3 concentrations at both the monitoring stations. We calculated delta O3 when air masses were observed from NE to SW or vice versa direction (about 200 records during the study period). In general, it should be noted that there is good agreement between the local station wind direction measurement (wind-roses analysis) and the back-trajectory analysis.

With regards to the low and high NOx regimes in our manuscript, the low and high NOx regimes refer to the concentrations of NOx that are either lower or higher than the cross over point i.e. [NOx] = 60 ppb (previous studies (Clapp and Jenkin, 2001; Notario et al., 2012; Tiwari et al., 2015) have also suggested similar NOx regimes). Furthermore, we do not have CH2O, NOy, H2O2 and HNO3 observations. Therefore, our analysis was limited only to NOx species. The paper mentioned by the reviewer (Kumar et al., 2012) reported the ratio of CH2O/NOy, H2O2/HNO3 and O3/(NOy–NOx) based on the modeling analysis performed by WRF-Chem model. This modeling analysis is not within the scope of our study. Clapp, L. J. and Jenkin, M. E.: Analysis of the relationship between ambient levels of O3, NO2 and NO as a function of NOx in the UK, Atmospheric Environment, 35(36), 6391- 6405, doi:10.1016/S1352-2310(01)00378-8, 2001. Notario, A., Bravo, I., Adame, J. A., Díaz-de-Mera, Y., Aranda, A., Rodríguez, A., and Rodríguez, D.: Analysis of NO, NO2, NOx, O3 and oxidant (Ox = O3+NO2) levels measured in a metropolitan area in the southwest Iberian Peninsula, Atmospheric Research, 104-105, 217-226, doi:10.1016/j.atmosres.2011.10.008, 2012. Tiwari, S., Dahiya, A., and Kumar, N.: Investigation into relationships among NO, NO2, NOx, O3, and CO at an urban background site in Delhi, India, Atmospheric Research, 157, 119-126, doi:10.1016/j.atmosres.2015.01.008, 2015.

Section 3.5 Page 11, Line 16: A good correlation implies good correlation coefficient (r) for a linear regression and not necessarily a large value of slope. Authors' logic of having a high CO/NOx ratio (slope of fit) because of a better correlation between the two species emitted from point sources is difficult to follow. The authors state

that high CO/NOx and low SO2/NOx ratio is characteristic of mobile sources. What are the values they referring to? Is there a threshold? What is the correlation coefficient of the liner regression between CO and NOx? Such correlation plots should at-least be provided in supplement. Since the authors have a great advantage of having the data from multiple receptor locations, they should use some statistical source apportionment models (for example, Positive Matrix Factorization (PMF) or the work by Garg and Sinha [2017]) Garg, S., and Sinha, B.: Determining the contribution of long-range transport, regional and local source areas, to PM10 mass loading in Hessen, Germany using a novel multi-receptor based statistical approach, Atmospheric Environment, 167, 566-575, 10.1016/j.atmosenv.2017.08.029, 2017. The authors have referred to the work of Parrish et al. [1991] for local source identification using CO/NOx ratio. However, longer-lived NOy should be used in place of NOx. This method can be used for estimating the background concentration of a short-lived species by performing a lognormal regression with a long-lived species. Simply using the ratio of CO and NOx to conclude the dominance of mobile source over point sources or vice versa by performing a linear regression over entire dataset of a group of specific monitoring station type will be a wrong over-interpretation of these ratios. This is also evident from the SO2/NOx ratios reported in Table 3. The SO2/NOx values are very similar for all the types of sites and even higher for roadside sites as compared for suburban and BKK sites. Based on authors assertion, it should be minimum for roadside sites among the three categories.

Authors' response: Thank you. The threshold or value to classify the difference between emissions from mobile sources and point sources has not been quantified definitively, however, the relative magnitude of the ratios provides an insight on source characteristics. We have compared our results with other published studies from different urban locations in US., Europe, and Asia before making our conclusion. It should also be noted that Positive Matrix Factorization (PMF) or the work by Garg and Sinha [2017] is in general applied to particulate matter. However, we have now provided correlation plots in the supplementary material.

With regards to the ration analysis using NOy species, the NOy data was not corrected as part of this study.

Section 3.5.2: Why are wind rose plotted for separate wind directions? It is very confusing. Authors should show a wind rose showing the fraction of wind coming from all the directions for O3 concentrations higher than 100 ppb. Higher fraction of wind from a particular direction would automatically point out major contribution from a particular wind direction. A polar plot with wind speed as radius axis, wind direction as angle and markers coloured according to observed O3 concentration could also be an alternative plot. How does the wind rose look like for periods with O3 concentration less than 100 ppb? If it is different from the ones for higher concentration, this would make the conclusion of higher ozone production from a particular wind direction stronger.

Authors' response: Thank you. The wind rose plots were created from wind speeds and wind directions (blowing from), during O3 episodes ([O3]hourly > 100 ppb). The wind rose plots were analyzed and, then, classified the into 3 groups, according to the predominant local wind directions, including northerly, westerly and southerly winds. These wind roses were not plotted based on wind directions alone. However, we have now provided new plots of wind speed, wind direction (blowing from) versus the concentrations of O3, during O3 episodes and non-episodes, in the modified manuscript. Generally, high O3 concentrations relate with low wind speed (lower than 4 ms-1) and relate with the predominant wind directions, especially, at 3T, 10T, 19T, 20T, 22T, 52T and 61T monitoring stations. Elevated O3 concentrations associated with northerly winds were at 22T monitoring station. At 3T, 10T, 19T, 20T and 61T monitoring stations, high concentrations of O3 associated with southerly winds. At 52T monitoring station, high concentrations of O3 associated with westerly winds. Moreover, the limited back trajectory analysis (based on NOAA HYSPLIT model) corroborates these findings and are now discussed in the manuscript.

"3.5.2 Effects of Pollutant Transport In general, O3 has a short lifetime in polluted urban atmosphere (approximately hours). However, O3 has a longer lifetime of several

weeks in the free troposphere. This occurrence may allow O3 to be transported over continental scales (Stevenson et al., 2006; Young et al., 2013; Monks et al., 2015). Figure 8 shows O3 concentrations, during episodes and non-episodes, with predominant wind directions and wind speeds. The results show that O3 exceedances are associated with low wind speed and predominant wind directions. In general, elevated O3 concentrations were observed with wind speed lower than 4 ms-1 with northerly winds (22T station), southerly winds (3T, 10T, 19T, 20T and 61T stations) and westerly winds (52T station). It is noteworthy that the southerly winds, generally, bring cleaner marine air mass to the land. However, under a stagnant condition (low wind speed), elevated O3 concentrations were observed (Sahu et al., 2013a, 2013b)."

Section 3.6: Authors need to describe the calculation of air quality index. If they have used the simple hourly average O3 concentration for calculate of AQI, then it is wrong. For calculation of hourly air quality index, O3 concentration for a give hour should be taken as the average for the previous 4 hours, current hour and next 3 hours. However, it is recommended to consider 8-hour AQI as mentioned previously in the review.

Authors' response: Thank you. To calculate AQI for O3, we calculate midpoints of 8-hour average of O3 concentration from the average of hourly O3 concentration of the previous four hours, at the given hour and the following three hours (this analysis is similar to the reviewer's suggestion). To get a valid calculation, at least 6 of 8 records (75%), are needed. Then we compared the calculated midpoints with the AQI table. However, we have now included this information in the supplement material.

US.EPA (2017), Air Quality Index (AQI) Basics, Available from: https://airnow.gov /index.cfm?action=aqibasics.aqi, (Accessed April 2017). US.EPA (2017), Daily and Hourly AQI – Ozone, Available from: https://forum.airnowtech.org/t/daily-and-hourly-aqi-ozone/170, (Accessed April 2017).

Technical comments: Page 1 Line 27 – page 2 line 2; Page 2, lines 11-16: These are better suited for site description.

Authors' response: Thank you. We have moved the information to "Section 1.2 Study Area".

"2.1 Study Area Figure 1 shows a map of BMR, the location of the monitoring stations in this study and major monsoon winds over the region. BMR refers to BKK and the five adjacent provinces, including Nakhon Pathom, Pathum Thani, Nonthaburi, Samut Prakan, and Samut Sakhon. These provinces are closely linked to BKK in terms of traffic and industrial development (Zhang and Oanh, 2002). Thailand has three official seasons–local summer (February to May), rainy (May to October) and local winter (October to February) as per the Thai Meteorological Department (TMD) (TMD, 2015). During the rainy season, this region's weather is influenced by Southwest monsoon wind that travels from the Indian Ocean to Thailand. This marine air mass contains high moisture, resulting in the wet season in Thailand. During this season, Thailand is characterized by cloudy weather with high precipitation and high humidity. From October to April, this region is influenced by Northeast monsoon wind that travels from the north-eastern and the northern parts of Asia (China and Mongolia). This monsoon wind brings a cold and dry air mass, which leads to the dry season (local summer and local winter) in Thailand. The local winter in Thailand is characterized by cool and dry weather, while the local summer is characterized by hot (35 to 40 âĐČ) to extremely hot weather (> 40 âĐČ) due to strong solar radiation. During the dry season, storms may occur during the seasonal transition (TMD, 2015). Transportation and industrial sectors are considered to be the major sources of air pollutants in the study area (Watcharavitoon et al., 2013). For example, in 2014, ∼36 million new vehicles were registered in Thailand and 29 % of these cars were registered in BKK (DLT, 2015). About 56 % and 28 % of the registered vehicles in BKK were gasoline and diesel engines. The remaining 16 % were Compressed Natural Gas (CNG) (DLT, 2017). In fact, the outskirts of BKK are populated with a variety of metal, auto parts, paper, plastic, food and chemical manufacturing facilities and power plants (DIW, 2016, 2016a, 2016b, 2016c, 2016d). "

Page 2, line 17: NO is Nitric oxide and not nitrogen oxide. Nitrogen oxides refer to the family of oxides of nitrogen.

Authors' response: Thank you. This typographical error is now corrected.

Page 2, Line 20: What is the basis of the statement "Moreover, BMR experiences primarily O3 exceedances amongst all the other gaseous criteria pollutants."

Authors' response: Thank you. The current study provides the basis for this statement.

Page 4, Line 4: Figure 1 should be mentioned earlier in the section.

Authors' response: Thank you. We have now mentioned the figure earlier in the section.

Page 4, Line 20: What are the "equivalent instruments"?

Authors' response: Thank you. The US EPA provides on its website (https://archive.epa.gov/ emap/archive-emap/web/html/qa_terms.html) clarity to equivalent method. Often it may also be referred to as Alternate method which is any body of procedures and techniques of sample collection and/or analysis for a characteristic of interest which is not a reference or approved equivalent method but which has been demonstrated in specific cases to produce results comparable to those obtained from a reference method.

Page 5, Line 3: Is the measurement period 2012-2014 or 2010-2014?

Authors' response: Thank you. This was a typographical error, which has now corrected.

Page 5, Line 7: What is the hourly "standard"?

Authors' response: Thank you. The National Ambient Air Quality Standards of Thailand provides hourly and 8-hour average standards of CO (30 ppm and 9 ppm, respectively), hourly and annually average standards of NO2 (0.17 ppm and 0.03 ppm, respectively),

hourly, 24-hour and annually average standards of SO2 (0.3 ppm, 0.12 ppm and 0.04 ppm, respectively), and hourly and 8-hour average standards of O3 (0.10 ppm and 0.07 ppm, respectively).

Page 6, line 5: Authors mention "VOCs concentrations were measured periodically only at one monitoring station limiting 5 its usefulness as part of this study". However, Zhang et al. [2002] have reported CH4 and NMVOC data from 10 out of 13 monitoring stations from BMR. Did the stations stopped monitoring CH4 and NMVOCs?

Authors' response: Thank you. We provided the limitation in the manuscript "While NOx was measured continuously at all the monitoring site, VOCs were measured periodically only at one monitoring station limiting its usefulness as part of this study".

Page 6, Line 12 and Figure 3: What is the explanation for a rather flat diel profile of NO2 at roadside sites. Roadside sites are influenced maximum by traffic emissions, and one would expect a bimodal shape of diel profile.

Authors' response: Thank you. We have now included the explanation in the manuscript.

Page 8: The rate constants and photolysis frequencies should be expressed in cm3 molecule-1 s-1 and s-1 respectively.

Authors' response: Thank you. We calculated j1 and k3 in the unit of min-1 and ppm-1 min-1, since we wanted to compare our values with other published studies that they reported their values in min-1 and ppm-1 min-1 (Clapp and Jenkin, 2001), (Tiwari et al., 2015).

Tiwari, S., Dahiya, A., and Kumar, N.: Investigation into relationships among NO, NO2, NOx, O3, and CO at an urban background site in Delhi, India, Atmospheric Research, 157, 119-126, doi:10.1016/j.atmosres.2015.01.008, 2015. Clapp, L. J. and Jenkin, M. E.: Analysis of the relationship between ambient levels of O3, NO2 and NO as a function of NOx in the UK, Atmospheric Environment, 35(36), 6391- 6405,

doi:10.1016/S1352-2310(01)00378-8, 2001. However, we have now provided j1 and k3 in the unit of s-1 and cm3 molecule-1 s-1 in the supplement material.

Page 9, line 9: The titration of O3 with NO will not effectively reduce the O3 concentrations. Such a titration process with produce NO2 which will again photolyze in the daytime and produce O3.

Authors' response: Thank you, however, we believe that the titration of O3 by fresh NO emitted from vehicles probably causes the lower O3 concentration observed at the roadside sites. Several studies reported the similar results, for example, Chan et al., (1998) studied surface ozone pattern in Hong Kong and reported that "In fact, this O3 sink is a common feature observed in many countries in the Northern Hemisphere, such as in Great Britain and Canada. In these two countries, the urban stations in central London (Bower et al. 1989; UKPORG 1990) and Alberta (Angle and Sandhu 1988) show lower O3 concentrations than their counterparts in the rural areas. This can be explained by the fact that the fresh precursor emissions from traffic and other sources cause direct chemical scavenging of O3." And "Indeed, Bell et al. (1970, 1977) has shown that even under light wind conditions, pollutants generated from local sources will be dispersed within 2–3 h. Thus, the titration effect of the fresh O3 precursors, especially NO, emitted from the metropolitan area of Hong Kong leads to the lower O3 levels in the urban stations in our study." Ghim and Chang (2002) studied ground-level ozone distribution in Korea and reported that "many studies reveal that background ozone concentrations in the Northern Hemisphere are around 3 5-40 ppb [Akimoto et al., 1996; Husar, 1998]. However, even in summer, monthly mean ozone levels in Korea are lower than this background level. . ..This could be primarily due to local effects of titration of O3 by fresh NOx emissions, since most ozone monitoring stations are located in or near major cities [Fuentes and Dann, 1994]". Munir et al., (2014) studied the diurnal variations of O3 in the UK and reported that "the lowest ozone concentrations are exhibited by Marylebone monitoring site which is located approximately 1 m from the edge of Marylebone road. This road has six lanes and has a flow of 80,000

vehicles per day. Most probably titration of ozone by fresh NO emitted by road transport keeps ozone concentrations low at this site." Chan, L. Y., Chan, C. Y. and Qin, Y.: Surface Ozone Pattern in Hong Kong, Journal of Applied Meteorology, 37, 1153-1165, 1998. Ghim, Y. S., and Chang, Y-. S.: Ground-level ozone distribution in Korea, Journal of Geographical Research, 105(7), 8877-8890, 2000. Munir, S., Chen, H., and Ropkins, K: Characterising the temporal variations of ground-level ozone and its relationship with traffic-related air pollutants in the United Kingdom: a quantile regression approach, Int. J. Sus. Dev. Plann, 9(1), 29-41, 2014.

Page 12, line 2: Please check the lifetime of O3. It should be few days (if not few weeks) in urban atmosphere.

Authors' response: Thank you. The lifetime of O3 that was provided in our manuscript was the lifetime in "a polluted urban atmosphere" where the lifetime of O3 is relatively short in this atmospheric condition. Monks et al., (2015) that reported ". . ..ozone has a relatively short atmospheric lifetime, typically hours, in polluted urban regions where concentrations of its precursors are high, its lifetime in the free troposphere is of the order of several weeks (Stevenson et al., 2006; Young et al., 2013). . ." Monks, P. S., Archibald, A. T., Colette, A., Cooper, O., Coyle, M., Derwent, R., Fowler, D., Granier, C., Law, K. S., Mills, G. E., Stevenson, D. S., Tarasova, O., Thouret, V., Schneidemesser, E., Sommariva, R., Wild, O., Williams, M. L.: Tropospheric ozone and its precursors from the urban to the global scale from air quality to short-lived climate forcer, Atmospheric Chemistry and Physics, 15(15), 8889-8973, doi:10.5194/acp-15-8889-2015, 2015.

Page 12, Line 15- Page 12, Line 5: Such description is better suited for introduction.

Authors' response: Thank you. We have now modified the manuscript by moving this description to the introduction section

Page 13 Line 18 to page 19 line 2: Such discussion is well suited for outlook after proper restructuring.

Authors' response: Thank you. Based on the reviewer, we have now modified our manuscript.

Figure 1: I would recommend showing airmass back trajectories rather than showing wind directions with two indicator arrows.

Authors' response: Thank you. Based on the reviewer, we have now modified the figure and included the airmass back trajectories based on NOAA HYSPLIT model. (Fig.1)

Figure 2: Ambient variability should also be shown along with average values. Authors should also show the concentrations of NO, in addition to CO, SO2, O3, and NO2. The colour for year 2010 and 2011 look same in panel "e".

Authors' response: Thank you. We did not provide a plot of NO in the manuscript, since this species is not a criteria pollutant. However, we have added a plot of the concentrations of NO and modified the figure in the modified manuscript. (Fig. 2)

Figure 3: Quality of figure should be improved (overall presentation, axis labels and legends). Ambient variability (as interquartile range or 1 _ standard deviation) should also be shown in addition to the average values. This should be done for other figures also in the paper.

Authors' response: Thank you. We have now modified the figure by adding standard deviations, and improved axis labels, and legends.

Figure 6: The minimum wind speed bin should be 0.5 – 2.0 (not 201). Please use the same radius scale for the wind rose plots.

Authors' response: Thank you. We have provided a new figure in the modified manuscript.

Table 1: Please refer to the comment for section 3.3.

Authors' response: Thank you. We have now modified our manuscript based on the reviewer's suggestion.

Table 3: Authors should also include the SO2/NOx ratio reported from various cities in India for mobile sources as reported by Mallik and Lal, 2014. Mallik, C., and Lal, S.: Seasonal characteristics of SO2, NO2, and CO emissions in and around the Indo-Gangetic Plain, Environmental Monitoring and Assessment, 186, 1295-1310, 10.1007/s10661-013-3458-y, 2014.

Authors' response: Thank you. We have now included the SO2/NOx ratios from the study of Mallik and Lal, 2014 in the manuscript.

"Guwahati and Nagpur, India*** SO2/NOx > 0.3 Kolkata, and Durgapur, India*** SO2/NOx $\leq$ 0.13"

Please also note the supplement to this comment:
https://www.atmos-chem-phys-discuss.net/acp-2017-1063/acp-2017-1063-AC1-supplement.pdf
* * *
[Figure]

[Figure]

| ID | Name |
|---|---|
| 13T | Electricity Generating Authority of Thailand |
| 22T | Sukhothai Thammathirat Open University |

| ID | Name |
|---|---|
| 20T | Bangkok University Rangsit Campus |

| ID | Name |
|---|---|
| 3T | Ratchaburana Post Office |
| 5T | Thai Meteorological Department |
| 10T | Klongjun-National Housing Authority |
| 11T | Kuaywang-National Housing Authority Stadium |
| 12T | Nonsi Withaya School |
| 15T | Mathayomwatsing School |
| 61T | Bodindecha (Sing Singhaseni) School |
| 54T | Dindaeng-National Housing Authority |
| 52T | Metropolitan Electricity Authority Substation Thonburi |

| ID | Name |
|---|---|
| 19T | Bangplee-National Housing Authority |

| ID | Name |
|---|---|
| 14T | Samut Sakhon Highway District |
| 27T | Samut Sakhon Provincial Administrative * |

**Fig. 1.** Map of BMR, monitoring station locations and two major monsoons winds (from NOAA HYSPLIT back trajectory model). Three monitoring station types, including BKK sites, roadside sites and BKK suburb site

[Figure]

**Fig. 2.** Maximum (vertical bars) and average (solid line) concentrations of a) CO, b) SO2, c) NO2 d) O3 and e) NO from the 15 monitoring stations, during 2010 to 2014, are compared with the hourly NAAQs (dotte

**Fig. 3.** Diurnal variations of gaseous species including O3, NO, NO2, CO and SO2 at a) BKK site b) roadside sites and c) BKK suburb sites.

[Figure]

**Fig. 4.** Diurnal variations of rate of change of O3 concentration ($\Delta[O3]/dt$) during a) local summer b) wet season and c) local winter.

[Figure]

**Fig. 5.** relationship, crossover point and concentration distribution of NO, NO2 and O3 at a) BKK sites b) roadside sites and c) BKK suburb sites.

[Figure]

**Fig. 6.** Effects of local and regional contributions on Ox during non-episode and episode days
at a) BKK sites, b) roadside sites and c) BKK suburb sites

[Figure]

**Fig. 7.** Backward trajectories from HYSPLIT model reveal a) NE wind direction (Jan 13, 2010) and b) SW wind direction (Jan 1, 2010)

[Figure]

**Fig. 8.** Relationship between the concentrations of O3, wind speeds and wind directions during a) O3 episodes ([O3]hourly > 100 ppb) and b) during non O3 episodes ([O3]hourly $\leq$ 100 ppb), over BMR during 2010 t

**Supplement:**

**Supplement Material**

**Data availability of the study**

**Fig. I:** Data availability from the 15 monitoring stations during 2010 to 2014.

**Maximum and average concentrations of gaseous pollutants**

**Table I:** Maximum and average concentration of gaseous pollutants from the three monitoring station types during 2010 to 2014.

| Monitoring station type | Maximum concentration** (ppb) | | | | | Average concentration*** (ppb) | | | | |
|---|---|---|---|---|---|---|---|---|---|---|
| | CO* | NO | NO$_2$ | SO$_2$ | O$_3$ | CO* | NO | NO$_2$ | SO$_2$ | O$_3$ |
| BKK sites | 5.7±0.9 | 419.2±236.0 | 120.5±14.8 | 29.8±5.3 | 153.7±10.8 | 0.7±0.2 | 16.3±7.8 | 20.2±5.7 | 3.3±1.0 | 18.6±2.3 |
| Roadside sites | 8.0±0.4 | 683.0±396.0 | 166.0±19.8 | 26.0±5.7 | 130.5±14.8 | 1.0±0.1 | 60.5±42.7 | 30.9±8.1 | 2.6±1.0 | 13.9±8.6 |
| BKK suburb sites | 4.5±1.2 | 297.5±70.6 | 115.8±15.8 | 72.2±58.3 | 163.0±18.5 | 0.7±0.1 | 11.4±3.8 | 16.1±2.6 | 4.0±2.3 | 21.4±3.3 |

* in ppm

** average maximum concentration ± 1 standard deviation (SD)

*** average concentration ± 1 SD

**Seasonal variations of gaseous pollutants**

**Fig. II:** Seasonal variations of a) CO b) $SO_2$ c) $NO_2$ and d) $O_3$ from the three monitoring station types during 2010 to 2014.

**Inter-annual variations of gaseous pollutants**

**Fig. III:** Inter-annual variations of a) CO b) $SO_2$ c) $NO_2$ and d) $O_3$ from the three monitoring station types during 2010 to 2014.

**Season wise of the diurnal variations of gaseous pollutants**

**Fig. IV:** Season wise of the diurnal variations of CO, $SO_2$, $NO_2$ and $O_3$ during a) local summer b) wet season and c) local winter at the three monitoring station types.

**Chemical rate coefficients**

**Table I:** chemical rate coefficients during dry season at BKK sites, roadside and BKK suburb sites, 2010 to 2014.

| Rate coefficient | Unit | BKK sites | Roadside sites | BKK suburb sites |
|:---:|:---|:---:|:---:|:---:|
| $j_1$ | $min^{-1}$ | 29.7±0.7 | 29.7±1.0 | 29.8±0.7 |
| | $s^{-1}$ | 0.004±0.002 | 0.007±0.0001 | 0.006±0.003 |
| $k_3$ | $ppm^{-1}\,min^{-1}$ | 0.47±0.2 | 0.64±0.3 | 0.55±0.3 |
| | $cm^3\,molecule^{-1}\,s^{-1}$ | $2.02e^{-14}±2.1e^{-16}$ | $2.03e^{-14}±1.2e^{-18}$ | $2.03e^{-14}±1.4e^{-16}$ |

**Correlation plots and correlation (r) between CO/NOₓ and SO₂/NOₓ and concentration distribution of species**

**Fig. V:** Correlation plots and correlation (r) between CO/NO$_x$ and SO$_2$/NO$_x$ and concentration distribution of species at a) BKK sites b) roadside sites and c) BKK suburb sites.

**AQI O₃ Calculation**

To calculate AQI for O$_3$, a midpoint of 8-hour average of O$_3$ concentration is needed. The midpoint of a specific hour is calculated from the average of hourly O$_3$ concentration of the previous four hours, at the given hour and the following three hours (Fig. VI a). To get a valid calculation, at least 6 of 8 records (75%), are needed. Calculated AQI values are compared with the AQI table (Fig. VI b) (US.EPA, 2017a, 2017b)

| Hours | 1 | 2 | 3 | 4 | 5 | 6 | 7 | 8 | 9 |
|-------|---|---|---|---|---|---|---|---|---|
| [O$_3$]$_{hourly}$ | | | | | | * | | | |
| | | The previous 4 hours | | | | AQI | The following 3 hours | | |

**Fig VI a:** The calculation of a midpoint of AQI for O$_3$ (modified from US.EPA, 2017a)

**Fig. VI b:** The AQI (US.EPA, 2017b)

---

## Author Comment (AC2) · 5 Jun 2018

We wish to thank the reviewer for the careful and thoughtful review of our manuscript. We appreciate the reviewer's comments "Overall, the article is well written and examines the interaction of Ozone with NOx regime. The analysis was well done." All the comments and suggestions are now incorporated in the manuscript.

Line 12: the statement is made, "On average, the number of hourly O3 exceedences ranged from 1 - 60 hours a year." This line is confusing. The overall average should be a value, not a range. If you wish to express it as a range, then do it by year, such as 2010 that average was XX hours, 2011, the average was XX hours. This range of 1-60 hours makes no sense.

Authors' response: Thank you. We have now incorporated the change in the modified manuscript.

"Abstract. Analysis of gaseous criteria pollutants in Bangkok Metropolitan Region (BMR), Thailand, during 2010 to 2014 reveals that while the hourly concentrations of CO, SO2 and NO2 were mostly in the National Ambient Air Quality Standards (NAAQs) of Thailand, the hourly concentrations of O3 frequently exceeded the standard. The results reveal that the problem of high O3 concentration continuously persisted in this area. Interconversion between O3, NO and NO2 indicates crossover points between the species occur when the concentration of NOx (= NO + NO2) is ~60 ppb. Under low NOx regime ([NOx] < 60 ppb), O3 is the dominant species, while, under high NOx regime ([NOx] > 60 ppb), NO dominates. Linear regression analysis between the concentrations of Ox (= O3 + NO2) and NOx provides the role of local and regional contributions to Ox. During O3 episodes ([O3]hourly > 100 ppb), the values of the local and regional contributions were nearly double of those during non-episodes. Ratio analysis suggests that the major contributors of primary pollutants over BMR are mobile sources. The Air Quality Index (AQI) for BMR was predominantly between good to moderate, however, unhealthy O3 categories were observed during episode conditions in the region."

Section 2, Methodology line 12: When you express a range (this applies throughout, do not mix the units and values. In Section 2, Methodology line 12 you state the temperature is (âĹij35C - 40C). This appears to read that it ranges from 35 degrees to - (minus) 40 degrees. Do this instead: (35 - 40 C).

Authors' response: Thank you. We have now incorporated the change in the modified manuscript.

"2.1 Study Area Figure 1 shows a map of BMR, the location of the monitoring stations in this study and major monsoon winds over the region. BMR refers to BKK and the five adjacent provinces, including Nakhon Pathom, Pathum Thani, Nonthaburi, Samut

Prakan, and Samut Sakhon. These provinces are closely linked to BKK in terms of traffic and industrial development (Zhang and Oanh, 2002). Thailand has three official seasons–local summer (February to May), rainy (May to October) and local winter (October to February) as per the Thai Meteorological Department (TMD) (TMD, 2015). During the rainy season, this region's weather is influenced by Southwest monsoon wind that travels from the Indian Ocean to Thailand. This marine air mass contains high moisture, resulting in the wet season in Thailand. During this season, Thailand is characterized by cloudy weather with high precipitation and high humidity. From October to April, this region is influenced by Northeast monsoon wind that travels from the north-eastern and the northern parts of Asia (China and Mongolia). This monsoon wind brings a cold and dry air mass, which leads to the dry season (local summer and local winter) in Thailand. The local winter in Thailand is characterized by cool and dry weather, while the local summer is characterized by hot (35 to 40 âĎĊ) to extremely hot weather (> 40 âĎĊ) due to strong solar radiation. During the dry season, storms may occur during the seasonal transition (TMD, 2015). Transportation and industrial sectors are considered to be the major sources of air pollutants in the study area (Watcharavitoon et al., 2013). For example, in 2014, ~36 million new vehicles were registered in Thailand and 29 % of these cars were registered in BKK (DLT, 2015). About 56 % and 28 % of the registered vehicles in BKK were gasoline and diesel engines. The remaining 16 % were Compressed Natural Gas (CNG) (DLT, 2017). In fact, the outskirts of BKK are populated with a variety of metal, auto parts, paper, plastic, food and chemical manufacturing facilities and power plants (DIW, 2016, 2016a, 2016b, 2016c, 2016d). "

Section 2, Methodology line 21: it states, " It is assumed that the monitoring sites used were representative of BMR specific patterns and trends." I think it goes without stating this that the professionals at the PCD would have done this and this does not need to be stated, but you would hope you would infer this. Remove this statement.

Authors' response: Thank you. We have now incorporated the change in the modified

manuscript and the statement has been removed.

Section 2, Methodology, line 27 -29: you list the sites (19T, 20T, etc...) which mean absolutely nothing to the reader then you state in line 29 that the figure shows these. The statement that mentions the figure should be the first line to the paragraph, not the last line. Move this line to the front so the reader can go get the figure look at it while you read the information.

Authors' response: Thank you. We have now modified the manuscript by removing site lists and referring the figure earlier in the section.

Section 2.2, line 2: you mention wind speed and direction. Is this average or vector data? Please state. This is important when calculating direction from which winds are blowing.

Authors' response: Thank you. The wind speed and wind direction are hourly averages.

Section 2.2, line 10: it is mentioned that equipment and monitoring station are calibrated every year. This is vague and could cast a shadow on validity of data. does this mean that this is done only once per year? Pollution instruments and met, or only met instruments. I am sure the PCD does calibrations more often than once annually. Please clarify this statement.

Authors' response: Thank you. As indicated in the manuscript, the data were collected, and after QA/QC, were provided to us by the Pollution Control Department (PCD), Thailand. Data loggers are calibrated/ checked at least every 15 days. Air inlets are cleaned at least every 15 days. Equipment is single-point calibrated and multi-point calibrated at least every 15 days and at least every 3 months. Monitoring stations and equipment are audited by external auditors every year. We have modified our manuscript to make a clarification.

".2 Data Collection and Data Analysis Over the four-year period, January 1, 2010 to

December 31, 2014, hourly observations from 15 Pollution Control Department (PCD) monitoring stations were analysed. The monitoring stations are categorized into three categories: BKK sites, roadside sites, and BKK suburb sites. BKK sites refer to the monitoring stations that are located within BKK's residential, commercial, industrial and mixed areas. They are within ∼50 to 100 m away from the road. Roadside sites refer to the monitoring stations that are located in BKK within 2 to 5 m from the road (Zhang and Oanh, 2002). BKK suburb sites refer to the monitoring stations that are located in provinces adjacent to BKK (Figure 1). Quality assurance and quality control on the data set were performed by PCD prior to receiving the data. Hourly observations of the gaseous pollutants and meteorological parameters were automatically collected with auto calibration at the monitoring stations. Manual quality control was performed when unusual observations were found. External audit of the equipment and monitoring stations were done every year. Data availability is provided in Figure I, supplement material. Gaseous species were measured at 3m above ground level (AGL). CO was measured using non-dispersive infrared detection (Thermo Scientific 48i). NO and $NO_2$ were measured using chemiluminescence detection (Thermo Scientific 42i). $SO_2$ was measured using ultraviolet (UV) fluorescence detection (Thermo Scientific 43i) and $O_3$ is measured by using UV absorption photometry detection (Thermo Scientific 49i). The meteorological parameters including wind speed (WS) and wind direction (WD) were measured at 10 m AGL by cup propeller and potentiometer wind vanes. Temperature (T) and relative humidity (RH) were measured at 2 m AGL by thermistor and thin film capacitor, respectively (Watchravitoon et al., 2013). All the meteorological measurements were made by Met One or equivalent method. Data analysis, statistical analysis and plots are performed using Excel 2016. Predominant wind directions related to $O_3$ concentrations are performed using Openair package (tool for the analysis of air pollution data) on RStudio program. "

Section 3.3 line 24: you use the term "atmospheric boundary layer." Is this the same as planetary boundary layer that was used previously? If it is the same term, then be consistent. If it isn't then please explain what this term means on how it differs from the

PBL.

Authors' response: Thank you. We have now corrected the manuscript by using "planetary boundary layer" instead of "atmospheric boundary layer" to provide consistency in the manuscript.

Page 8, line 11: Please explain why the ratios of NO2 and NO show significant difference. You make the statement but you don't say why. this is an important claim that you make in this paper.

Authors' response: Thank you. As suggested by the reviewer, we have now removed this from our manuscript.

Page 9, line 9: you state, "In conclusion, the titration of O3 and NO is perhaps one of the most important processes..." Please elaborate about why this is so important.

Authors' response: Thank you. The titration of O3 by NO is perhaps one of the most important processes to reduce O3 concentration at roadside sites, due to this monitoring station type is more affected by fresh NO emitted from vehicles than the other monitoring station types. Several studies reported the effect of the titration of O3 by NO, for example, Chan et al., (1998) studied surface ozone pattern in Hong Kong and reported that "In fact, this O3 sink is a common feature observed in many countries in the Northern Hemisphere, such as in Great Britain and Canada. In these two countries, the urban stations in central London (Bower et al. 1989; UKPORG 1990) and Alberta (Angle and Sandhu 1988) show lower O3 concentrations than their counterparts in the rural areas. This can be explained by the fact that the fresh precursor emissions from traffic and other sources cause direct chemical scavenging of O3." And "Indeed, Bell et al. (1970, 1977) has shown that even under light wind conditions, pollutants generated from local sources will be dispersed within 2–3 h. Thus, the titration effect of the fresh O3 precursors, especially NO, emitted from the metropolitan area of Hong Kong leads to the lower O3 levels in the urban stations in our study." Ghim and Chang (2002) studied ground-level ozone distribution in Korea and reported that "many studies reveal

that background ozone concentrations in the Northern Hemisphere are around 3 5-40 ppb [Akimoto et al., 1996; Husar, 1998]. However, even in summer, monthly mean ozone levels in Korea are lower than this background level. . ..This could be primarily due to local effects of titration of O3 by fresh NOx emissions, since most ozone monitoring stations are located in or near major cities [Fuentes and Dann, 1994]". Munir et al., (2014) studied the diurnal variations of O3 in the UK and reported that "the lowest ozone concentrations are exhibited by Marylebone monitoring site which is located approximately 1 m from the edge of Marylebone road. This road has six lanes and has a flow of 80,000 vehicles per day. Most probably titration of ozone by fresh NO emitted by road transport keeps ozone concentrations low at this site." Chan, L. Y., Chan, C. Y. and Qin, Y.: Surface Ozone Pattern in Hong Kong, Journal of Applied Meteorology, 37, 1153-1165, 1998. Ghim, Y. S., and Chang, Y-. S.: Ground-level ozone distribution in Korea, Journal of Geographical Research, 105(7), 8877-8890, 2000. Munir, S., Chen, H., and Ropkins, K: Characterising the temporal variations of ground-level ozone and its relationship with traffic-related air pollutants in the United Kingdom: a quantile regression approach, Int. J. Sus. Dev. Plann, 9(1), 29-41, 2014.

However, we have removed this from the manuscript.

Page 11, line 5: you state, "However, a negative delta O3 may be negative. However, it appears that the data doesn't support this in the paragraph. Why is this statement made?

Authors' response: Thank you. We put this statement to clarify that a negative delta O3 was possibly to be observed due to O3 deposition and/or O3 consummation. Our analysis, negative values of delta O3 were observed several times, however, the average of those was positive.
* * *

---

## Author Response (AR1)

**Response to Reviewers Comments**

**Assessment of Gaseous Pollutants in Bangkok Metropolitan Region, Thailand**

Pornpan Uttamang, Viney P Aneja, Adel Hanna

**Ref:** acp-2017-1063

We wish to thank the reviewers for the careful and thoughtful review of our manuscript. We appreciate reviewer 2' s comments "Overall, the article is well written and examines the interaction of Ozone with $NO_x$ regime. The analysis was well done." All the comments and suggestions are now incorporated in the manuscript.

**Reviewer #1**

**General Comment:**

Pornpan Uttamang et al. have presented observations of CO, $NO_x$, $SO_2$ and $O_3$ from 15 monitoring sites at understudied Bangkok Metropolitan Region (BMR) for a fiveyear- long period from 2010-2014. Background pertaining to the air-quality in terms of PM and $O_3$ exceedance events in the BMR is provided. However, the authors do not mention the knowledge gap or scientific question that they want to address from this study. I have major concerns with the paper which include description of analytical methods and discussion about quality control (calibration and sampling protocols, filter criteria) of dataset used. The statistical analysis is also weak which mostly covers average/maximum over the entire study periods, without going into details of specific seasons, inter-annual trends and pin-pointing the season-specific emission sources /formation processes and removal processes of the pollutants. The conclusions are drawn either from the regression lines having poor fit parameters or oversimplification of methods for source identification available in the peer-reviewed literature. The manuscript needs to address the major concerns (highlighted in specific comments) before it can be considered further. After performing the analysis suggested in the specific comments, corrections and restructuring the paper, the scientific outcome might be significantly different from the present version and should be considered as a new publication.

**Authors' response:** Thank you. The last paragraph of the Introduction (line 17 – 24) succinctly provides both the knowledge gap and scientific question being addressed. We have now made the statistical analysis more robust based on reviewer' suggestions.

**Specific Comments:**

**Title:** Authors might consider making the title of the paper more specific. Authors assess CO, $NO_x$, $SO_2$ and $O_3$ air pollution and not overall air pollution in general.

**Authors' response:** Thank you. We have now modified the title to "Assessment of Gaseous Pollutants in Bangkok Metropolitan Region, Thailand"

**Introduction:** The authors have included a description of auto-mobile fleet and manufacturing industries in the introduction which should rather be a part of the site description. The introduction is poorly structured. Authors should include a brief literature review of the previous works from BMR, outlook from these studies and what are the knowledge gaps they want to address from this paper.

The authors should also mention, why they have chosen to study CO, $NO_x$, $SO_2$ and $O_3$. At-least a line each about their importance regarding atmospheric chemistry and air quality should be present. The authors have referred to Zhang and Oanh, [2002] for the site description. However, the findings there should also be mentioned in the introduction, as Zhang and Oanh, [2002] have analyzed monthly and diel variation, $O_3$ exceedances, drivers for high ozone episodes and relationship of ozone production with NOx/NMHC ratio. These are quite relevant for the present study. Similarly, the work of Pochanart et al., [2001] should be highlighted in the introduction. I found few other studies (mentioned below) which are relevant to the present work and should be highlighted in the introduction. There might be several more!

Jinsart, W., Tamura, K., Loetkamonwit, S., Thepanondh, S., Karita, K., and Yano, E.: Roadside Particulate Air Pollution in Bangkok, Journal of the Air & Waste Management Association, 52, 1102-1110, 10.1080/10473289.2002.10470845, 2002.

Suthawaree, J., Tajima, Y., Khunchornyakong, A., Kato, S., Sharp, A., and Kajii, Y.: Identification of volatile organic compounds in suburban Bangkok, Thailand and their potential for ozone formation, Atmospheric Research, 104-105, 245-254, 10.1016/j.atmosres.2011.10.019, 2012

**Authors' response:** Thank you. We have now modified the Introduction to include the discussion in the references provided by the reviewer**.**

**Page 2, Line 23:** Authors state "possible emission sources of pollutants that associate with $O_3$ formation are identified". However, such identification is not discussed in the manuscript. Authors have only used the ratio of CO/$NO_x$ and $SO_2$/$NO_x$ to identify whether the emission sources are mobile or point in nature. The method itself has an inherent limitation which is mentioned later in the specific comment for the section.

**Authors' response:** Thank you. We have now modified the Introduction. Please refer to our discussion in the comment "Section 3.5 Page 11, Line 16" below.

**Methodology:** The exact measurement period should be mentioned in this section. This paper discusses a five-year-long measurement period and shows data over 15 different measurement stations and authors should provide a time-line for data availability for each station.

**Authors' response:** Thank you. We have now provided the measurement period in the modified manuscript and provided a time-line for data availability for each station as part of the supplement material.

**Page 3, line 22:** What is the basis of the assumption that monitoring sites used were representative of BMR specific patterns and trends?

**Authors' response:** Thank you. Based on the reviewer, we have modified the manuscript by removing the sentence.

**Data Collection and Data Analysis:** I have major concerns with this section. Authors did not provide any sampling details. The trace gas analysers for CO, $NO_x$, $SO_2$ and $O_3$ are known to have drifts with time. Authors mention that equipment and monitoring stations are calibrated every year. This is not enough. There should be frequent zero drift check for CO (at-least daily) and for NOx, $SO_2$ and $O_3$ (at-least once a week). The linearity of the detection should also be checked with calibration experiments performed at-least once a month. The authors did not provide any information about the drift in the sensitivity of instruments over the period of 5 years. Detection limits of the trace gas analysers and uncertainties of the measurements should also be provided.

**Authors' response:** Thank you. As indicated in the manuscript, the data were collected, and after QA/QC, were provided by the Pollution Control Department (PCD), Thailand. Data loggers are calibrated/ checked at least every 15 days. Air inlets are cleaned at least every 15 days. Equipment is single-point calibrated and multi-point calibrated at least every 15 days and at least every 3 months. Monitoring stations and equipment are audited by external auditors every year. We have modified our manuscript to make a clarification.

**Page 4, line 6:** Authors mention that quality assurance and quality control on the dataset were performed by PCD prior to receiving the data. What are these quality controls?

**Authors' response:** Thank you. QA/AC protocols are published in the PCD, Thailand government document.

**Page 4, line 9:** What are the manual quality controls? What are the criteria for choosing unusual observations?

**Authors' response:** Thank you. QA/AC protocols are published in the PCD, Thailand government document. However, we did not provide any additional guidelines for data collection.

**Result and Discussion**

**Section 3.1:** Authors have only provided maximum and average over the entire five-year period. Since they have continuous one hour time resolution dataset from 15 monitoring stations for a five year long period, authors should also include inter-annual variability and seasonal statistics at-least for different monitoring station types. Given the advantage of also having wind speed/ wind direction data, authors should consider comparing various airmass fetch regions for some monitoring sites. For ozone, it makes more sense to separate daytime and night-time before reporting the average concentrations. The authors discuss extensively about 1-hour exceedance of ozone concentrations, but there is no description of how are these exceedance events calculated.

One cannot compare the hourly average concentrations directly with the NAAQS. What about the ozone exceedance from 8-h standard? Bangkok air quality standard provides criteria for both 1-hour and 8-hour average ozone. 8-h average is intended to provide a better protection from long term ozone exposure.

**Authors' response:** Thank you. We have now provided inter-annual plots and seasonal variation plots for gaseous criteria pollutants from the monitoring stations. In the plots, we have averaged the data for each monitoring type station (from BKK sites, roadside sites and BKK suburb sites). This figure is now included in the supplementary material. However, the discussion associated with this information is now included in the manuscript. Moreover, we have now included the average concentrations of $O_3$ during daytime (6:00 AM to 6:00 PM) and during nighttime (6:00 PM to 6:00 AM) in the modified manuscript.

With regards to segregating wind direction data we performed a more robust back trajectory analysis. Moreover, we provided wind-rose plots for each of the monitoring stations and discussed it in the manuscript.

The National Ambient Air Quality Standards of Thailand provides hourly and 8-hour average standards of $O_3$ (0.10 ppm and 0.07 ppm, respectively). In this study, we compared the hourly concentrations of $O_3$ with the hourly $O_3$ standard in order to examine number of $O_3$ exceedances. To study the effects of $O_3$ on human health, we applied Air Quality Index (AQI) of $O_3$ instead of using the $O_3$ exceedance from 8-hour standard, which we believe that, using AQI of $O_3$ will provide more advantages than using the $O_3$ exceedance from 8-hour standard. Since AQI of $O_3$ is categorized into six categories, with four of the six categories providing the information of the severity of high $O_3$ concentrations on human health, from sensitive groups to healthy people; therefore, applying AQI for $O_3$ will provide better information for air quality management.

**Section 3.2 Diurnal Variation of the Gaseous Species:** Regional meteorology has strong influence on primary emission processes, production of secondary pollutant e.g. ozone and ambient concentrations of pollutants. I would recommend season wise analysis of diel variation of gaseous species. For example, the authors can refer to the work of Gaur et al. [2014] and Kumar et al. [2016]. This would also enable to identify the periods when ozone production is maximum during the year. Authors should also analyse, how does rate of formation of ozone from sunrise until it attains the peak daytime values changes at different sites and in different seasons. Authors could refer to the work of Naja and Lal [2002].

Gaur, A., Tripathi, S. N., Kanawade, V. P., Tare, V., and Shukla, S. P.: Four-year measurements of trace gases (SO2, NOx, CO, and O3) at an urban location, Kanpur, in Northern India, Journal of Atmospheric Chemistry, 1-19, 10.1007/s10874-014-9295-8, 2014.

Kumar, V., Sarkar, C., and Sinha, V.: Influence of post-harvest crop residue fires on surface ozone mixing ratios in the N.W. IGP analyzed using 2 years of continuous in situ trace gas measurements, J. Geophys. Res., 121, 3619–3633 10.1002/2015JD024308, 2016.

Naja, M., and Lal, S.: Surface ozone and precursor gases at Gadanki (13.5_N, 79.2_E), a tropical rural site in India, Journal of Geophysical Research: Atmospheres, 107, 10.1029/2001jd000357, 2002.

Authors should provide an explanation for why a second peak is not observed in the diel profiles of $SO_2$ at all sites. In line 20 of page 7, authors speculate that $SO_2$ is emitted by automotive diesel engine exhaust. If we observe the diel profile of NO from the BKK sites, a bimodal profile is observed which is attributed to traffic emissions. Moreover, even if we assume that manufacturing facilities point sources are the $SO_2$ contributors as mentioned in line 23 of page 11, their emission strength would not vary over the time scale of a day and a bimodal profile driven by boundary layer meteorology should be observed.

Similarly, authors should also provide an explanation for the relatively flatter diel profile of $NO_2$ at roadside sites.

**Authors' response:** Thank you. We have now provided season wise analysis of diel variation of gaseous species for the three monitoring station types in the supplementary material and a discussion is provided in the manuscript.

We have also provided the rate of change of $O_3$ concentration (Naja and Lal, 2002) during the three seasons and in the three monitoring station types with explanation in the modified manuscript.

For the diurnal variations of $SO_2$, in the manuscript, we explained that "…The concentrations of $SO_2$ increase again in the afternoon and reach a second-peak around 21:00 LT over roadside sites. Over BKK sites and BKK suburb sites, the concentrations of $SO_2$ are nearly constant after 19:00 LT…" which the second peak of $SO_2$ were observed over three monitoring station types, but the magnitude of the concentrations of $SO_2$ over BKK sites and BKK suburb sites were small. For the diurnal variation of $NO_2$, at the roadside sites also showed a bimodal distribution, but flatter than those at other sites. However, we have now included the clarification and explanation in the manuscript.

**Section 3.3 Interconversion between $O_3$, NO and $NO_2$ and Photochemical Reaction:** I have major concerns again with this section. In line 23, authors mention "the photostationary state (PSS) is applied through all chemical reactions for $O_3$ formation during 10:00-16:00 LT". However, later in the section they assume photostationary state only between $O_3$, NO and $NO_2$. In polluted environments, $RO_2$ and $HO_2$ also oxidize NO to $NO_2$ and hence disturb the PSS of NO, $NO_2$ and $O_3$ [Mannschreck at al., 2004]. Hence the $j_1$ values calculated by only considering $O_3$, NO and $NO_2$ in the PSS would not be accurate.

Mannschreck, K., Gilge, S., Plass-Duelmer, C., Fricke, W., and Berresheim, H.: Assessment of the applicability of NO-NO2-O3 photostationary state to long-term measurements at the Hohenpeissenberg GAW Station, Germany, Atmos. Chem. Phys., 4, 1265-1277, 10.5194/acp-4 1265-2004, 2004.

Moreover, $j_1$ values are strongly dependent on incoming solar radiation and mentioning an average over 10:00 L.T. until 16:00 L.T. will be oversimplification. In the moderately polluted environment, the photostationary state between $O_3$, NO and $NO_2$ is achieved within 60 s to 300 s during daytime [Trebs et al., 2012]. Authors should perform a calculation of $j_1$ at similar timescales.

Trebs, I., Mayol-Bracero, O. L., Pauliquevis, T., Kuhn, U., Sander, R., Ganzeveld, L., Meixner, F. X., Kesselmeier, J., Artaxo, P., and Andreae, M. O.: Impact of the Manaus urban plume on trace gas mixing ratios near the surface in the Amazon Basin: Implications for the NO-NO2-O3 photostationary state and peroxy radical levels, Journal of Geophysical Research: Atmospheres, 117, 10.1029/2011JD016386, 2012.

I cannot understand, why the authors emphasize the calculated $k_3$ values. It depends on a single parameter which is temperature! Do the authors want to show that their temperature measurements are reasonable or their calculation is accurate? Next, the authors are using $O_3$ measurements to estimate the $j_1$ values and again using $j_1$ to explain high $O_3$ concentration at some sites. This is cyclic. Polynomial trend lines are used to investigate the interconversion between $O_3$, NO and

NO2. However, as seen from Figure 4, The fit is very poor for $O_3$ in all the three cases. So inference drawn using these fits would not be conclusive.

**Authors' response:** Thank you. In our study, we evaluated the relationship between $O_3$ with the gaseous criteria pollutants for the NAAQs of Thailand. The assumption of the photostationary state (PSS) ($\phi = 1$), therefore, was applied through the chemical reactions of $O_3$ and $NO_x$ only.

Mannschreck et al., 2004, reviewed the PSS parameter ($\phi$) as:

$$\phi = \frac{j[NO_2]}{k[NO][O_3]}$$

Where $j$ was the photolysis rate of NO2, and $k$ was the rate of the chemical reaction of NO and $O_3$. In the Mannschreck et al., 2004, when $\phi$ was equal to 1, then other chemical reactions converting NO to NO2 and local emissions of either compound were negligible. However, these cases were rare and were limited to **very polluted** conditions. On the other hand, peroxy radicals (RO2) played an important role to contribute to additional NO and NO2, under clean or moderately polluted conditions. In the study of Mannschreck et al., (2004), the measurement was performed in a rural site, generally, the site was affected by relatively clean air masses (yearly average of $NO_x$ was below 3.5 ppb). The site was surrounded by forests (70%, mostly coniferous) and agricultural pastures (30%). The distance to the nearest urban and major industrial areas was about 80 km. Furthermore, the study mentioned that *"for high $NO_x$ concentrations the levels of peroxy radicals should approach zero, since the sink for RO2 increases with increasing NO and since OH as a precursor for RO2 as well as RO species are removed via reaction with NO2"*. Therefore, we believe that the assumption of PSS holds for our study region (e.g. average of hourly concentration of $NO_x$ at BKK sites, roadside sites, and BKK suburb sites were ~30 ppb, ~88 ppb and ~21 ppb, respectively. These $NO_x$ values are far in excess of rural/semi-rural values in the study of Mannschreck et al., (2004)).

With regards to the calculation of $j_1$, it is strongly dependent on incoming solar radiation and on other variables (i.e. the following equation):

$$j_{q,p} = \int_0^\infty 4\pi I_{p,\lambda} b_{a,g,q,\lambda,T} Y_{q,p,\lambda,T} d\lambda$$

Where

$$4\pi I_{p,\lambda} = \text{Actinic flux}$$

$$b_{a,g,q,\lambda,T} = \text{Average absorption cross section}$$

$$Y_{q,p,\lambda,T} = \text{Average quantum yield}$$

However, these variables were not measured in our study at the monitoring stations.

With regards to the calculation of $k_3$, our intention is not to emphasize the $k_3$ calculation to show that our temperature measurements were reasonable, but rather to calculate $j_1$. However, we have now modified our manuscript and removed using $j_1$ values to explain $O_3$ concentration.

With regards to the polynomial trend lines, we have now modified the plots by including histogram of the concentrations of $O_3$, NO and $NO_2$ to present data distribution of these species. Generally, most of the records are in low to middle concentration bins.

**Section 3.4:** What are the criteria for differentiation between episodes and non-episodes?

For the linear regression presented in this section, one can observe significant scatter around the fitted line. In some cases, (for example roadside sites, non-episode), one can clearly observe two different regions in the plots and a single linear fit over entire dataset cannot be justified.

For the delta $O_3$ analysis, how were the back trajectories calculated? How many trajectories per day and how many days backward trajectories at what height were calculated? Authors should also provide the number of days/hours when N-NE and S-SE wind directions respectively were observed. How was the agreement between local wind directions and the wind directions derived from NOAA HYSPLIT model? Given the large scatter around average of 10 ppb delta $O_3$, the conclusion of local production is rather week for days with $O_3$ concentrations > 80 ppb. The sentence structuring is poor and was difficult to follow. This also needs improvement. The conclusion regarding crossover points is drawn from polynomial regressions which have very poor fit parameters (and not even mentioned in the paper). The high NOx and low NOx regime should be calculated based on the ratio of $NO_x$ OH reactivity and VOC OH reactivity or using model calculated indicators (e.g. $CH_2O/NO_y$, $H_2O_2/HNO_3$ and $O_3/(NO_y–NO_x)$) as described by Kumar et al., [2011]. Classification based on cross over points are an oversimplification of the polynomial fits.

Kumar, R., Naja, M., Pfister, G. G., Barth, M. C., Wiedinmyer, C., and Brasseur, G. P.:

Simulations over South Asia using the Weather Research and Forecasting model with

Chemistry (WRF-Chem): chemistry evaluation and initial results, Geosci. Model Dev.,

5, 619-648, 10.5194/gmd-5-619-2012, 2012.

**Authors' response:** Thank you. We had explained that an $O_3$ episode was identified when hourly $O_3$ concentrations were greater than 100 ppb (the $O_3$ NAAQs for Thailand).

With regards to the linear regression, we presented "*the estimation*" of local and regional contributions of $O_x$. Furthermore, we also compared the result from our study to results from other studies (using similar linear regression method).

For the two different observed $NO_x$ regions at roadside sites, we provided in our discussion the following "It is noteworthy that the pattern of the local and regional contributions at roadside sites during non-episode period is composed of two $NO_x$ concentration regimes. The low $NO_x$ regime ($NO_x < 60$ ppb) resembles the local and regional contributions during non-episode over BKK suburb sites. The high $NO_x$ regime ($NO_x > 60$ ppb) may represent typical characteristic of air quality near roads".

To estimate the local and regional contribution by plotting $O_x$ against $NO_x$ were reported in several published studied, for example Clapp and Jenkin (2001), Aneja et al., (2001), Mazzeo et al., (2005), Tang et al., (2009), Notario et al., (2012), Rasheed et al., (2014), Tiwari et al., (2015). These studies provide similar plots to our study. All these references are cited in the manuscript.

Clapp, L. J. and Jenkin, M. E.: Analysis of the relationship between ambient levels of $O_3$, $NO_2$ and NO as a function of $NO_x$ in the UK, Atmospheric Environment, 35(36), 6391- 6405, doi:10.1016/S1352-2310(01)00378-8, 2001.

Aneja, V. P., Agarwal, A., Roelle, P. A., Phillips, S. B., Tong, Q., Watkins, N., and Yablonsky, R.: Measurements and Analysis of Criteria Pollutants in New Delhi, India, Environment International, 27, 35-42, doi:10.1016/s0160-4120(01)00051-4, 2001.

Mazzeoa, N. A., Venegasa, L. E. and Chorenc, H: Analysis of NO, $NO_2$, $O_3$ and $NO_x$ concentrations measured at a green area of Buenos Aires City during wintertime, Atmospheric Environment, 39, 3055–3068, doi:10.1016/j.atmosenv.2005.01.029, 2005.

Tang, G., Li, X., Wang, Y., Xin, J., and Ren, X.: Surface ozone trend details and interpretations in Beijing, 2001–2006, Atmos. Chem. Phys., 9, 8813–8823, 2009.

Notario, A., Bravo, I., Adame, J. A., Díaz-de-Mera, Y., Aranda, A., Rodríguez, A., and Rodríguez, D.: Analysis of NO, $NO_2$, $NO_x$, $O_3$ and oxidant (Ox = $O_3$+$NO_2$) levels measured in a metropolitan area in the southwest Iberian Peninsula, Atmospheric Research, 104-105, 217-226, doi:10.1016/j.atmosres.2011.10.008, 2012.

Rasheed, A., Aneja, V. P., Aiyyer, A., and Rafique, U.: Measurements and analysis of air quality in Islamabad, Pakistan, Earth's Future, 2, 303-314, doi:10.1002/2013EF000174, 2014.

Tiwari, S., Dahiya, A., and Kumar, N.: Investigation into relationships among NO, $NO_2$, $NO_x$, $O_3$, and CO at an urban background site in Delhi, India, Atmospheric Research, 157, 119-126, doi:10.1016/j.atmosres.2015.01.008, 2015.

With regards to the delta $O_3$ analysis, back trajectory was determined when hourly concentration of $O_3 > 80$ ppb was observed either at 27T or 20T sites. By performing the backward trajectories using the NOAA HYSPLIT model, we identified the wind directions that related with high $O_3$ concentrations at both the monitoring stations. We calculated delta $O_3$ when air masses were observed from NE to SW or vice versa direction (about 200 records during the study period). In general, it should be noted that there is good agreement between the local station wind direction measurement (wind-roses analysis) and the back-trajectory analysis.

With regards to the low and high $NO_x$ regimes in our manuscript, the low and high $NO_x$ regimes refer to the concentrations of $NO_x$ that are either lower or higher than the cross over point i.e. $[NO_x] = 60$ ppb (previous studies (Clapp and Jenkin, 2001; Notario et al., 2012; Tiwari et al., 2015) have also suggested similar $NO_x$ regimes). Furthermore, we do not have $CH_2O$, $NO_y$, $H_2O_2$ and $HNO_3$ observations. Therefore, our analysis was limited only to $NO_x$ species. The paper mentioned by the reviewer (Kumar et al., 2012) reported the ratio of $CH_2O/NO_y$, $H_2O_2/HNO_3$ and $O_3/(NO_y-NO_x)$ based on the modeling analysis performed by WRF-Chem model. This modeling analysis is not within the scope of our study.

Clapp, L. J. and Jenkin, M. E.: Analysis of the relationship between ambient levels of $O_3$, $NO_2$ and NO as a function of $NO_x$ in the UK, Atmospheric Environment, 35(36), 6391- 6405, doi:10.1016/S1352-2310(01)00378-8, 2001.

Notario, A., Bravo, I., Adame, J. A., Díaz-de-Mera, Y., Aranda, A., Rodríguez, A., and Rodríguez, D.: Analysis of NO, $NO_2$, $NO_x$, $O_3$ and oxidant ($Ox = O_3+NO_2$) levels measured in a metropolitan area in the southwest Iberian Peninsula, Atmospheric Research, 104-105, 217-226, doi:10.1016/j.atmosres.2011.10.008, 2012.

Tiwari, S., Dahiya, A., and Kumar, N.: Investigation into relationships among NO, $NO_2$, $NO_x$, $O_3$, and CO at an urban background site in Delhi, India, Atmospheric Research, 157, 119-126, doi:10.1016/j.atmosres.2015.01.008, 2015.

**Section 3.5 Page 11, Line 16:** A good correlation implies good correlation coefficient (r) for a linear regression and not necessarily a large value of slope. Authors' logic of having a high CO/NOx ratio (slope of fit) because of a better correlation between the two species emitted from point sources is difficult to follow. The authors state that high CO/NOx and low SO2/NOx ratio is characteristic of mobile sources. What are the values they referring to? Is there a threshold? What is the correlation coefficient of the liner regression between CO and NOx? Such correlation plots should at-least be provided in supplement. Since the authors have a great advantage of having the data from multiple receptor locations, they should use some statistical source apportionment models (for example, Positive Matrix Factorization (PMF) or the work by Garg and Sinha [2017])

Garg, S., and Sinha, B.: Determining the contribution of long-range transport, regional and local source areas, to PM10 mass loading in Hessen, Germany using a novel multi-receptor based statistical approach, Atmospheric Environment, 167, 566-575, 10.1016/j.atmosenv.2017.08.029, 2017.

The authors have referred to the work of Parrish et al. [1991] for local source identification using CO/$NO_x$ ratio. However, longer-lived $NO_y$ should be used in place of $NO_x$. This method can be used for estimating the background concentration of a short-lived species by performing a lognormal regression with a long-lived species. Simply using the ratio of CO and NOx to conclude the dominance of mobile source over point sources or vice versa by performing a linear regression over entire dataset of a group of specific monitoring station type will be a wrong over-interpretation of these ratios. This is also evident from the $SO_2/NO_x$ ratios reported in Table 3. The SO2/NOx values are very similar for all the types of sites and even higher for roadside sites as compared for suburban and BKK sites. Based on authors assertion, it should be minimum for roadside sites among the three categories.

**Authors' response:** Thank you. The threshold or value to classify the difference between emissions from mobile sources and point sources has not been quantified definitively, however, the relative magnitude of the ratios provides an insight on source characteristics. We have compared our results with other published studies from different urban locations in US., Europe, and Asia before making our conclusion. It should also be noted that Positive Matrix Factorization (PMF) or the work by Garg and Sinha [2017] is in general applied to particulate matter. However, we have now provided correlation plots in the supplementary material.

With regards to the ration analysis using NOy species, the $NO_y$ data was not corrected as part of this study.

**Section 3.5.2:** Why are wind rose plotted for separate wind directions? It is very confusing. Authors should show a wind rose showing the fraction of wind coming from all the directions for $O_3$ concentrations higher than 100 ppb. Higher fraction of wind from a particular direction would automatically point out major contribution from a particular wind direction. A polar plot with wind speed as radius axis, wind direction as angle and markers coloured according to observed $O_3$ concentration could also be an alternative plot. How does the wind rose look like for periods with $O_3$ concentration less than 100 ppb? If it is different from the ones for higher concentration, this would make the conclusion of higher ozone production from a particular wind direction stronger.

**Authors' response:** Thank you. The wind rose plots were created from wind speeds and wind directions (blowing from), during $O_3$ episodes ($[O_3]_{hourly} > 100$ ppb). The wind rose plots were analyzed and, then, classified the into 3 groups, according to the predominant local wind directions, including northerly, westerly and southerly winds. These wind roses were not plotted based on wind directions alone. However, we have now provided new plots of wind speed, wind direction (blowing from) versus the concentrations of $O_3$, during $O_3$ episodes and non-episodes, in the modified manuscript. Generally, high $O_3$ concentrations relate with low wind speed (lower than 4 ms$^{-1}$) and relate with the predominant wind directions, especially, at 3T, 10T, 19T, 20T, 22T, 52T and 61T monitoring stations. Elevated $O_3$ concentrations associated with northerly winds were at 22T monitoring station. At 3T, 10T, 19T, 20T and 61T monitoring stations, high concentrations of $O_3$ associated with southerly winds. At 52T monitoring station, high concentrations of $O_3$ associated with westerly winds. Moreover, the limited back trajectory analysis (based on NOAA HYSPLIT model) corroborates these findings and are now discussed in the manuscript.

**Section 3.6:** Authors need to describe the calculation of air quality index. If they have used the simple hourly average $O_3$ concentration for calculate of AQI, then it is wrong. For calculation of hourly air quality index, $O_3$ concentration for a give hour should be taken as the average for the previous 4 hours, current hour and next 3 hours. However, it is recommended to consider 8-hour AQI as mentioned previously in the review.

**Authors' response:** Thank you. To calculate AQI for $O_3$, we calculate midpoints of 8-hour average of $O_3$ concentration from the average of hourly $O_3$ concentration of the previous four hours, at the given hour and the following three hours (this analysis is similar to the reviewer's suggestion). To get a valid calculation, at least 6 of 8 records (75%), are needed. Then we compared the calculated midpoints with the AQI table. However, we have now included this information in the supplement material.

US.EPA (2017), Air Quality Index (AQI) Basics, Available from: https://airnow.gov
/index.cfm?action=aqibasics.aqi, (Accessed April 2017).

US.EPA (2017), Daily and Hourly AQI – Ozone, Available from:
https://forum.airnowtech.org/t/daily-and-hourly-aqi-ozone/170, (Accessed April 2017).

**Technical comments:**

**Page 1 Line 27 – page 2 line 2; Page 2, lines 11-16:** These are better suited for site description.

**Authors' response:** Thank you. We have moved the information to "Section 1.2 Study Area".

**Page 2, line 17:** NO is Nitric oxide and not nitrogen oxide. Nitrogen oxides refer to the family of oxides of nitrogen.

**Authors' response:** Thank you. This typographical error is now corrected.

**Page 2, Line 20:** What is the basis of the statement "Moreover, BMR experiences primarily O3 exceedances amongst all the other gaseous criteria pollutants."

**Authors' response:** Thank you. The current study provides the basis for this statement.

**Page 4, Line 4:** Figure 1 should be mentioned earlier in the section.

**Authors' response:** Thank you. We have now mentioned the figure earlier in the section.

**Page 4, Line 20:** What are the "equivalent instruments"?

**Authors' response:** Thank you. The US EPA provides on its website (https://archive.epa.gov/

emap/archive-emap/web/html/qa_terms.html) clarity to equivalent method. Often it may also be referred to as Alternate method which is any body of procedures and techniques of sample collection and/or analysis for a characteristic of interest which is not a reference or approved equivalent method but which has been demonstrated in specific cases to produce results comparable to those obtained from a reference method.

**Page 5, Line 3:** Is the measurement period 2012-2014 or 2010-2014?

**Authors' response:** Thank you. This was a typographical error, which has now corrected.

**Page 5, Line 7:** What is the hourly "standard"?

**Authors' response:** Thank you. The National Ambient Air Quality Standards of Thailand provides hourly and 8-hour average standards of CO (30 ppm and 9 ppm, respectively), hourly and annually average standards of $NO_2$ (0.17 ppm and 0.03 ppm, respectively), hourly, 24-hour and annually average standards of $SO_2$ (0.3 ppm, 0.12 ppm and 0.04 ppm, respectively), and hourly and 8-hour average standards of $O_3$ (0.10 ppm and 0.07 ppm, respectively).

**Page 6, line 5:** Authors mention "VOCs concentrations were measured periodically only at one monitoring station limiting 5 its usefulness as part of this study". However, Zhang et al. [2002] have reported $CH_4$ and NMVOC data from 10 out of 13 monitoring stations from BMR. Did the stations stopped monitoring $CH_4$ and NMVOCs?

**Authors' response:** Thank you. We provided the limitation in the manuscript "While $NO_x$ was measured continuously at all the monitoring site, VOCs were measured periodically only at one monitoring station limiting its usefulness as part of this study".

**Page 6, Line 12 and Figure 3:** What is the explanation for a rather flat diel profile of $NO_2$ at roadside sites. Roadside sites are influenced maximum by traffic emissions, and one would expect a bimodal shape of diel profile.

**Authors' response:** Thank you. We have now included the explanation in the manuscript.

**Page 8:** The rate constants and photolysis frequencies should be expressed in $cm^3$ molecule$^{-1}$ s$^{-1}$ and s$^{-1}$ respectively.

**Authors' response:** Thank you. We calculated $j_1$ and $k_3$ in the unit of min$^{-1}$ and ppm$^{-1}$ min$^{-1}$, since we wanted to compare our values with other published studies that they reported their values in min$^{-1}$ and ppm$^{-1}$ min$^{-1}$ (Clapp and Jenkin, 2001), (Tiwari et al., 2015).

Tiwari, S., Dahiya, A., and Kumar, N.: Investigation into relationships among NO, $NO_2$, $NO_x$, $O_3$, and CO at an urban background site in Delhi, India, Atmospheric Research, 157, 119-126, doi:10.1016/j.atmosres.2015.01.008, 2015.

Clapp, L. J. and Jenkin, M. E.: Analysis of the relationship between ambient levels of $O_3$, $NO_2$ and NO as a function of $NO_x$ in the UK, Atmospheric Environment, 35(36), 6391- 6405, doi:10.1016/S1352-2310(01)00378-8, 2001.

However, we have now provided *j1* and $k_3$ in the unit of s$^{-1}$ and $cm^3$ molecule$^{-1}$ s$^{-1}$ in the supplement material.

**Page 9, line 9:** The titration of $O_3$ with NO will not effectively reduce the $O_3$ concentrations.

Such a titration process with produce $NO_2$ which will again photolyze in the daytime and produce $O_3$.

**Authors' response:** Thank you, however, we believe that the titration of $O_3$ by fresh NO emitted from vehicles probably causes the lower $O_3$ concentration observed at the roadside sites. Several studies reported the similar results, for example, Chan et al., (1998) studied surface ozone pattern in Hong Kong and reported that "*In fact, this $O_3$ sink is a common feature observed in many countries in the Northern Hemisphere, such as in Great Britain and Canada. In these two countries, the urban stations in central London (Bower et al. 1989; UKPORG 1990) and Alberta (Angle and Sandhu 1988) show lower $O_3$ concentrations than their counterparts in the rural areas. This can be explained by the fact that the fresh precursor emissions from traffic and other sources cause direct chemical scavenging of $O_3$.*" And "*Indeed, Bell et al. (1970, 1977) has shown that even under light wind conditions, pollutants generated from local sources will be dispersed within 2–3 h. Thus, the titration effect of the fresh $O_3$ precursors, especially NO, emitted from the metropolitan area of Hong Kong leads to the lower $O_3$ levels in the urban stations in our study.*" Ghim and Chang (2002) studied ground-level ozone distribution in Korea and reported that "*many studies reveal that background ozone concentrations in the Northern Hemisphere are around 3 5-40 ppb [Akimoto et al., 1996; Husar, 1998]. However, even in summer, monthly mean ozone levels in Korea are lower than this background level....This could be primarily due to local effects of titration of $O_3$ by fresh NOx emissions, since most ozone monitoring stations are located in or near major cities [Fuentes and Dann, 1994]*". Munir et al., (2014) studied the diurnal variations of $O_3$ in the UK and reported that "*the lowest ozone concentrations are exhibited by Marylebone monitoring site which is located approximately 1 m from the edge of Marylebone road. This road has six lanes and has a flow of 80,000 vehicles per day. Most probably titration of ozone by fresh NO emitted by road transport keeps ozone concentrations low at this site.*"

Chan, L. Y., Chan, C. Y. and Qin, Y.: Surface Ozone Pattern in Hong Kong, Journal of Applied Meteorology, 37, 1153-1165, 1998.

Ghim, Y. S., and Chang, Y-. S.: Ground-level ozone distribution in Korea, Journal of Geographical Research, 105(7), 8877-8890, 2000.

Munir, S., Chen, H., and Ropkins, K: Characterising the temporal variations of ground-level ozone and its relationship with traffic-related air pollutants in the United Kingdom: a quantile regression approach, Int. J. Sus. Dev. Plann, 9(1), 29-41, 2014.

**Page 12, line 2:** Please check the lifetime of $O_3$. It should be few days (if not few weeks) in urban atmosphere.

**Authors' response:** Thank you. The lifetime of $O_3$ that was provided in our manuscript was the lifetime in "a polluted urban atmosphere" where the lifetime of $O_3$ is relatively short in this atmospheric condition. Monks et al., (2015) that reported "*....ozone has a relatively short atmospheric lifetime, typically hours, in polluted urban regions where concentrations of its precursors are high, its lifetime in the free troposphere is of the order of several weeks (Stevenson et al., 2006; Young et al., 2013)...*"

Monks, P. S., Archibald, A. T., Colette, A., Cooper, O., Coyle, M., Derwent, R., Fowler, D., Granier, C., Law, K. S., Mills, G. E., Stevenson, D. S., Tarasova, O., Thouret, V., Schneidemesser, E., Sommariva, R., Wild, O., Williams, M. L.: Tropospheric ozone and its precursors from the urban to the global scale from air quality to short-lived climate forcer, Atmospheric Chemistry and Physics, 15(15), 8889-8973, doi:10.5194/acp-15-8889-2015, 2015.

**Page 12, Line 15- Page 12, Line 5:** Such description is better suited for introduction.

**Authors' response:** Thank you. We have now modified the manuscript by moving this description to the introduction section

**Page 13 Line 18 to page 19 line 2:** Such discussion is well suited for outlook after proper restructuring.

**Authors' response:** Thank you. Based on the reviewer, we have now modified our manuscript.

**Figure 1:** I would recommend showing airmass back trajectories rather than showing wind directions with two indicator arrows.

**Authors' response:** Thank you. Based on the reviewer, we have now modified the figure and included the airmass back trajectories based on NOAA HYSPLIT model.

**Figure 2:** Ambient variability should also be shown along with average values. Authors should also show the concentrations of NO, in addition to CO, $SO_2$, $O_3$, and $NO_2$. The colour for year 2010 and 2011 look same in panel "e".

**Authors' response:** Thank you. We did not provide a plot of NO in the manuscript, since this species is not a criteria pollutant. However, we have added a plot of the concentrations of NO and modified the figure in the modified manuscript.

**Figure 3:** Quality of figure should be improved (overall presentation, axis labels and legends). Ambient variability (as interquartile range or 1 _ standard deviation) should also be shown in addition to the average values. This should be done for other figures also in the paper.

**Authors' response:** Thank you. We have now modified the figure by adding standard deviations, and improved axis labels, and legends.

**Figure 6:** The minimum wind speed bin should be 0.5 – 2.0 (not 201). Please use the same radius scale for the wind rose plots.

**Authors' response:** Thank you. We have provided a new figure in the modified manuscript.

**Table 1:** Please refer to the comment for section 3.3.

**Authors' response:** Thank you. We have now modified our manuscript based on the reviewer's suggestion.

**Table 3:** Authors should also include the $SO_2/NOx$ ratio reported from various cities in India for mobile sources as reported by Mallik and Lal, 2014.

Mallik, C., and Lal, S.: Seasonal characteristics of $SO_2$, $NO_2$, and CO emissions in and around the Indo-Gangetic Plain, Environmental Monitoring and Assessment, 186, 1295-1310, 10.1007/s10661-013-3458-y, 2014.

**Authors' response:** Thank you. We have now included the $SO_2/NO_x$ ratios from the study of Mallik and Lal, 2014 in the manuscript.

**Reviewer #2**

We wish to thank the reviewer for the careful and thoughtful review of our manuscript. We appreciate the reviewer' s comments "Overall, the article is well written and examines the interaction of Ozone with $NO_x$ regime. The analysis was well done."

All the comments and suggestions are now incorporated in the manuscript.

**Line 12:** the statement is made, "On average, the number of hourly $O_3$ exceedences ranged from 1 - 60 hours a year." This line is confusing. The overall average should be a value, not a range. If you wish to express it as a range, then do it by year, such as 2010 that average was XX hours, 2011, the average was XX hours. This range of 1-60 hours makes no sense.

**Authors' response:** Thank you. We have now incorporated the change in the modified manuscript.

**Section 2, Methodology line 12:** When you express a range (this applies throughout, do not mix the units and values. In Section 2, Methodology line 12 you state the temperature is ($\sim$35C - 40C). This appears to read that it ranges from 35 degrees to - (minus) 40 degrees. Do this instead: (35 - 40 C).

**Authors' response:** Thank you. We have now incorporated the change in the modified manuscript.

**Section 2, Methodology line 21:** it states, " It is assumed that the monitoring sites used were representative of BMR specific patterns and trends." I think it goes without stating this that the professionals at the PCD would have done this and this does not need to be stated, but you would hope you would infer this. Remove this statement.

**Authors' response:** Thank you. We have now incorporated the change in the modified manuscript and the statement has been removed.

**Section 2, Methodology, line 27 -29:** you list the sites (19T, 20T, etc...) which mean absolutely nothing to the reader then you state in line 29 that the figure shows these. The statement that mentions the figure should be the first line to the paragraph, not the last line. Move this line to the front so the reader can go get the figure look at it while you read the information.

**Authors' response:** Thank you. We have now modified the manuscript by removing site lists and referring the figure earlier in the section.

**Section 2.2, line 2:** you mention wind speed and direction. Is this average or vector data? Please state. This is important when calculating direction from which winds are blowing.

**Authors' response:** Thank you. The wind speed and wind direction are hourly averages.

**Section 2.2, line 10:** it is mentioned that equipment and monitoring station are calibrated every year. This is vague and could cast a shadow on validity of data. does this mean that this is done only once per year? Pollution instruments and met, or only met instruments. I am sure the PCD does calibrations more often than once annually. Please clarify this statement.

**Authors' response:** Thank you. As indicated in the manuscript, the data were collected, and after QA/QC, were provided to us by the Pollution Control Department (PCD), Thailand. Data loggers are calibrated/ checked at least every 15 days. Air inlets are cleaned at least every 15 days. Equipment is single-point calibrated and multi-point calibrated at least every 15 days and at least every 3 months. Monitoring stations and equipment are audited by external auditors every year. We have modified our manuscript to make a clarification.

**Section 3.3 line 24:** you use the term "atmospheric boundary layer." Is this the same as planetary boundary layer that was used previously? If it is the same term, then be consistent. If it isn't then please explain what this term means on how it differs from the PBL.

**Authors' response:** Thank you. We have now corrected the manuscript by using "planetary boundary layer" instead of "atmospheric boundary layer" to provide consistency in the manuscript.

**Page 8, line 11:** Please explain why the ratios of $NO_2$ and NO show significant difference. You make the statement but you don't say why. this is an important claim that you make in this paper.

**Authors' response:** Thank you. As suggested by the reviewer, we have now removed this from our manuscript.

**Page 9, line 9:** you state, "In conclusion, the titration of $O_3$ and NO is perhaps one of the most important processes..." Please elaborate about why this is so important.

**Authors' response:** Thank you. The titration of $O_3$ by NO is perhaps one of the most important processes to reduce $O_3$ concentration at roadside sites, due to this monitoring station type is more affected by fresh NO emitted from vehicles than the other monitoring station types. Several studies reported the effect of the titration of $O_3$ by NO, for example, Chan et al., (1998) studied surface ozone pattern in Hong Kong and reported that "*In fact, this $O_3$ sink is a common feature observed*

*in many countries in the Northern Hemisphere, such as in Great Britain and Canada. In these two countries, the urban stations in central London (Bower et al. 1989; UKPORG 1990) and Alberta (Angle and Sandhu 1988) show lower $O_3$ concentrations than their counterparts in the rural areas. This can be explained by the fact that the fresh precursor emissions from traffic and other sources cause direct chemical scavenging of $O_3$.* " And "*Indeed, Bell et al. (1970, 1977) has shown that even under light wind conditions, pollutants generated from local sources will be dispersed within 2–3 h. Thus, the titration effect of the fresh $O_3$ precursors, especially NO, emitted from the metropolitan area of Hong Kong leads to the lower $O_3$ levels in the urban stations in our study.*" Ghim and Chang (2002) studied ground-level ozone distribution in Korea and reported that "*many studies reveal that background ozone concentrations in the Northern Hemisphere are around 3 5-40 ppb [Akimoto et al., 1996; Husar, 1998]. However, even in summer, monthly mean ozone levels in Korea are lower than this background level….This could be primarily due to local effects of titration of $O_3$ by fresh NOx emissions, since most ozone monitoring stations are located in or near major cities [Fuentes and Dann, 1994]*". Munir et al., (2014) studied the diurnal variations of $O_3$ in the UK and reported that "*the lowest ozone concentrations are exhibited by Marylebone monitoring site which is located approximately 1 m from the edge of Marylebone road. This road has six lanes and has a flow of 80,000 vehicles per day. Most probably titration of ozone by fresh NO emitted by road transport keeps ozone concentrations low at this site.*"

Chan, L. Y., Chan, C. Y. and Qin, Y.: Surface Ozone Pattern in Hong Kong, Journal of Applied Meteorology, 37, 1153-1165, 1998.

Ghim, Y. S., and Chang, Y-. S.: Ground-level ozone distribution in Korea, Journal of Geographical Research, 105(7), 8877-8890, 2000.

Munir, S., Chen, H., and Ropkins, K: Characterising the temporal variations of ground-level ozone and its relationship with traffic-related air pollutants in the United Kingdom: a quantile regression approach, Int. J. Sus. Dev. Plann, 9(1), 29-41, 2014.

However, we have removed this from the manuscript.

**Page 11, line 5:** you state, "However, a negative delta $O_3$ may be negative. However, it appears that the data doesn't support this in the paragraph. Why is this statement made?

**Authors' response:** Thank you. We put this statement to clarify that a negative delta $O_3$ was possibly to be observed due to $O_3$ deposition and/or $O_3$ consummation. Our analysis, negative values of delta $O_3$ were observed several times, however, the average of those was positive.
* * *

[revised manuscript text omitted]

---

## Referee Report (RR1)

**General comment**

I appreciate the major revision undertaken by the authors. They have improved the quality of analysis. Thanks for including a timeline of the measurements, inter-annual plots and seasonal variation. However, some of the important crucial concerns are still present. I had to face hard time to read the interactive discussion for my first review, where several special characters and formulae were not typeset properly. Some examples are on page C15 and Page C16. There have been several careless mistakes in the supplement. While the main text mentions values of *j1* in the range of 0.12-1.22 min$^{-1}$, corresponding values provided in the supplement table 1 are $\sim 29$ min $^{-1}$. The figure captions and legends are difficult to follow and sometimes even not explained properly. Examples are Main text figure 3, Main text figure 5, supplement Figure 5, An important concern I want to raise for the editor is related to the journal scope which is focused on studies with general implications for atmospheric science rather than *investigations that are primarily of local or technical interest*. How does this article fit in the scope of ACP considering the investigation of air pollution of a region presented in this study?

Some other concerns are:

1. Gaseous pollutants in the title is still too broad a domain for a study reporting only $O_3$, CO, $NO_x$ and $SO_2$.

2. The details of calibrations are still not provided. Given the long measurement period reported in this study, it is very important to know how the instrument response drifted over time.

3. Several major conclusions are drawn from poor correlation. Examples are:

   - Section 3.3. I am not convinced by the PSS analysis performed by the authors in the revised manuscript. Apart from the method by Trebs et al. (as suggested in the first review), authors could have used NCAR TUV model for calculation of *j1*. Even in the polluted environment like in Delhi, deviation from PSS was observed at $NO_x$ values more than 10 ppb (Chate el al 2014). At such high $NO_x$ concentration, systematic deviation from PSS with Leighton ration less than 1 was observed. Value of Leighton ratio =1 is a very rare finding in ambient environment. Hence, I again question the validity of conclusion drawn on this assumption. I again ask the authors to calculate *j1* using TUV model or using solar radiation and check the Leighton ratio. In any case, given that *j1* only depends on actinic flux, quantum yield and absorption cross-section, how would the authors explain a variation

of more than an order of magnitude during the daytime hours of the same season (line 210 of the revised manuscript).

Chate, D. M., et al. (2014), Deviations from the O3NONO2 photo-stationary state in Delhi, India, Atmospheric Environment, 96(0), 353-358, doi:http://dx.doi.org/10.1016/j.atmosenv.2014.07.054.

- Cross over point and regime identification: First of all, legends are not provided in this figure 5. If I assume the purple points to be $O_3$, still the fit statistics (which are not even provided either in text or in figure) are very poor. So the conclusion drawn regarding cross over points are not robust. There is no clear crossover point for the BKK sites.

- Section 3.4: The scatter plots have very poor fit for Fig 6a and Fig 6c for the non-episode events. In addition to the slope and intercept, authors should also consider the goodness of fit before drawing any conclusion.

- Section 3.5.1 (Figure V of the supplement): Even in the best case, the $r^2$ is less than 0.3 in the best case. What is the significance of local source analysis based on such poor statistics? Why the frequency distribution of $SO_2$ (I assume it is frequency distribution as no information is provided either in figure caption or text) has wiggles in between.

- Lines 265-272: The statistics are too poor for the conclusion of $\sim$ 10 ppb enhancement in $O_3$. The spread in delta $O_3$ ranges from -66 to +96 ppb.

Finally, even if the manuscript is considered for publication, a major proof reading and improvement in Figure quality should be done.

---

## Author Response (AR2)

**Response to Reviewers Comments**

**Assessment of Gaseous Pollutants in Bangkok Metropolitan Region, Thailand**

Pornpan Uttamang, Viney P Aneja, Adel Hanna

**Ref:** acp-2017-1063

We wish to thank the reviewers for the careful and thoughtful review of our revised manuscript. All comments and suggestions now are incorporated in the manuscript. The main manuscript, supplement, figures and tables have been reviewed and modified following reviewers' comments. Furthermore, the manuscript has been reviewed by an English Editor.

**Reviewer#1**

**General Comment:**

I appreciate the major revision undertaken by the authors. They have improved the quality of analysis. Thanks for including a timeline of the measurements, inter-annual plots and seasonal variation. However, some of the important crucial concerns are still present. I had to face hard time to read the interactive discussion for my first review, where several special characters and formulae were not typeset properly. Some examples are on page C15 and Page C16. There have been several careless mistakes in the supplement. While the main text mentions values of $j_1$ in the range of 0.12-1.22 $min^{-1}$, corresponding values provided in the supplement table 1 are ~ 29 $min^{-1}$. The figure captions and legends are difficult to follow and sometimes even not explained properly. Examples are Main text figure 3, Main text figure 5, supplement Figure 5, An important concern I want to raise for the editor is related to the journal scope which is focused on studies with general implications for atmospheric science rather than investigations that are primarily of local or technical interest. How does this article fit in the scope of ACP considering the investigation of air pollution of a region presented in this study?

**Authors' response:** Thank you.

1) We have corrected and have also improved the table by including maximum, minimum, means and standard deviations of $j_1$ and $k_3$ based on observations at the three monitoring station types, and calculated $j_1$ based on modeling analysis. More details can be found in authors' response in comment 3.1.

Table II is changed from:

**Table II:** chemical rate coefficients during dry season at BKK sites, roadside and BKK suburb sites, 2010 to 2014

| Rate coefficient | Unit | BKK sites | Roadside sites | BKK suburb sites |
|---|---|---|---|---|
| $j_1$ | $min^{-1}$ | 29.7±0.7 | 29.7±1.0 | 29.8±0.7 |
|  | $s^{-1}$ | 0.004±0.002 | 0.007±0.0001 | 0.006±0.003 |
| $k_3$ | $ppm^{-1} min^{-1}$ | 0.47±0.2 | 0.64±0.3 | 0.55±0.3 |
|  | $cm^3 molecule^{-1} s^{-1}$ | $2.02e^{-14}$±$2.1e^{-16}$ | $2.03e^{-14}$±$1.2e^{-18}$ | $2.03e^{-14}$±$1.4e^{-16}$ |

Table is changed to:

**Table II:** Statistical analysis of the chemical rate coefficients ($j_1$ and $k_3$) based on an observational analysis during dry seasons at BKK sites, roadside and BKK suburb sites, 2010 to 2014.; and statistical analysis of $j_1$ based on a modeling analysis at the latitude and the longitude of 13.76 °N, 100.50 °E in a dry season, 2010.

| Sites | Rate coefficient | | | | | | | | | | | |
|---|---|---|---|---|---|---|---|---|---|---|---|---|
| | $j_1$ | | | | | | $k_3$ | | | | | |
| | min$^{-1}$ | | | s$^{-1}$ | | | ppm$^{-1}$ min$^{-1}$ | | | cm$^3$ molecule$^{-1}$ s$^{-1}$ | | |
| | Max | Min | Average | Max | Min | Average | Max | Min | Average | Max | Min | Average |
| **Based on observation\*** | | | | | | | | | | | | |
| *BKK* | 0.95 | 0.12 | 0.74±0.2 | 0.016 | 0.004 | 0.008±0.035 | 30.9 | 28.6 | 29.8±0.7 | 2.06e$^{-14}$ | 1.99e$^{-14}$ | 2.02e$^{-14}$±2.01e$^{-16}$ |
| *Roadside* | 0.90 | 0.36 | 0.64±0.3 | 0.015 | 0.011 | 0.013±0.002 | 30.6 | 28.3 | 29.7 | 2.03e$^{-14}$ | 2.03e$^{-14}$ | 2.03e$^{-14}$ |
| *BKK suburb* | 1.22 | 0.34 | 0.55±0.3 | 0.022 | 0.007 | 0.010±0.004 | 30.9 | 28.8 | 29.8±0.7 | 2.04e$^{-14}$ | 2.01e$^{-14}$ | 2.03e$^{-14}$±1.34e$^{-16}$ |
| **Based on modeling\*\*** | | | | | | | | | | | | |
| *13.7 °N, 100.5 °E* | | | | | | 0.021±0.002 | | | | | | |

2) As suggested by the reviewer, we have now improved Figure 3 caption in the manuscript.

Figure 3 caption is changed from:

**Fig 3:** Diurnal variations of gaseous species including $O_3$, NO, $NO_2$, CO and $SO_2$ at a) BKK site b) roadside sites and c) BKK suburb sites.

Figure 3 caption is changed to:

**Fig 3:** Diurnal variations of gaseous species. The plots provide the average concentrations of $O_3$, NO and $NO_2$ in ppb, the average concentrations of CO in ppm and the average concentrations of $SO_2$ in ppb at a) BKK site; b) roadside sites; and c) BKK suburb sites. Vertical bars provide ±1 standard deviations of the species concentrations.

3) We have improved Figure 5 and its caption in the manuscript. All the legends are now clear; and we have provided clarity on the crossover points.

Figure 5 is changed from:

[Figure]

**Fig. 5:** relationship, crossover point and concentration distribution of NO, NO$_2$ and O$_3$ at a) BKK sites b) roadside sites and c) BKK suburb sites.

Figure 5 is changed to:

[Figure]

**Fig. 5:** Relationships and crossover points of NO, NO₂ and O₃ at a) BKK sites b) roadside sites and c) BKK suburb sites; and concentration distributions of those species at d) BKK sites e) roadside sites and f) BKK suburb sites.

4) We have now removed Figure V in the supplement and replaced it by a table that provides the correlations between CO and $NO_x$; and $SO_2$ and $NO_x$ (Table III). We have also provided the correlation among the species at all the monitoring sites in this table. Table III provides comprehensive statistical information.

**Table III:** Correlation between CO and $NO_x$; and $SO_2$ and $NO_x$ at BKK sites, roadside sites and suburb sites, during 2010 to 2014; together with ±1 standard deviation.

| Station type | Station ID | | Correlation | |
|---|---|---|---|---|
| | | | CO and $NO_x$ | $SO_2$ and $NO_x$ |
| **BKK sites** | **3T** | | 0.76 | 0.34 |
| | **5T** | | 0.56 | 0.37 |
| | **10T** | | 0.76 | 0.36 |
| | **11T** | | 0.68 | 0.33 |
| | **12T** | | 0.61 | 0.26 |
| | **15T** | | 0.64 | 0.29 |
| | **61T** | | 0.85 | 0.28 |
| | | **Average** | 0.69±0.10 | 0.32±0.04 |
| **Roadside sites** | **52T** | | 0.73 | 0.49 |
| | **54T** | | 0.72 | 0.56 |
| | | **Average** | 0.72 | 0.53 |
| **Suburb sites** | **13T** | | 0.92 | 0.32 |
| | **14T** | | 0.64 | 0.11 |
| | **19T** | | 0.47 | 0.39 |
| | **20T** | | 0.55 | 0.21 |
| | **22T** | | 0.71 | 0.27 |
| | **27T\*** | | 0.77 | 0.53 |
| | | **Average** | 0.68±0.16 | 0.30±0.15 |

**Note**: *the correlations are calculated based observations during 2010 to 2013

**Some other concerns:**

1. Gaseous pollutants in the title is still too broad a domain for a study reporting only $O_3$, CO, $NO_x$ and $SO_2$.

**Authors' response:** Thank you. We have now modified the title from "Assessment of Gaseous Pollutants in Bangkok Metropolitan Region, Thailand" to "Assessment of Gaseous **Criteria** Pollutants in Bangkok Metropolitan Region, Thailand"

2. The details of calibrations are still not provided. Given the long measurement period reported in this study, it is very important to know how the instrument response drifted over time.

**Authors' response:** Thank you. According to a document of the Pollution Control Department, Thailand (PCD): term of reference (TOR) for air quality detectors and air quality monitoring stations in the notification of PCD, Number 17/2559, date November 17, 2016.

**Detector details:**

$SO_2$ detectors:
    range:  0-500 ppb to 0-20 ppm with auto ranging or better.
    lower detection limit: < 1 ppb
    precision: 0.5 ppb or < 1% of reading or better
    zero drift: < 1 ppb/24-hour

span drift: < 1% of reading/ 24-hour

$NO_x$ detectors:
      range:  0-500 ppb to 0-20 ppm with auto ranging or better.
      lower detection limit: < 0.5 ppb
      precision: 0.5 ppb or < 1% of reading or better
      zero drift: < 1 ppb/24-hour
      span drift: < 1% of full scale/ 24-hour

CO detectors:
      range:  0-50 ppm to 0-200 ppm with auto ranging or better.
      lower detection limit: < 0.05 ppm
      precision: < 1% of reading or better
      zero drift: < 0.1 ppm/24-hour
      span drift: < 1% of reading/ 24-hour

$O_3$ detectors:
      range:  0-500 ppb to 0-10 ppm with auto ranging or better.
      lower detection limit: < 0.6 ppb
      precision: 1% of reading or better
      zero drift: < 1 ppb/24-hour
      span drift: < 1% of reading/ 24-hour

Detector/ data loggers/ air inlets calibration/ maintenance:
      single point calibration for detectors: every 15 days
      multi-point calibration with 3 span levels (20 %, 40 % and 80 %): every 90 days
      mass flow adjustments: every 90 days
      molybdenum converter for $NO_2$ detectors: at least 4 times in 730 days
      zero air generators: at least 4 times in 730 days
      data loggers maintenance: every 15 days
      air inlets maintenance: every 15 days

**Acceptance data criteria:**

    1. Span drifts
        span drift: < ± 10 % of full scale for $NO_2$, $SO_2$, CO detectors
        span drift: < ± 7 % of full scale $O_3$ detectors
    2. Zero checks
        zero drift: < ± 5 ppb for $NO_2$, $SO_2$ and $O_3$ detectors
        zero drift: < ± 0.4 ppm for CO detectors

We have now included this information in section A, the supplement material.

3. Several major conclusions are drawn from poor correlation.

3.1 Section 3.3. I am not convinced by the PSS analysis performed by the authors in the revised manuscript. Apart from the method by Trebs et al. (as suggested in the first review), authors could have used NCAR TUV model for calculation of j1. Even in the polluted environment like in Delhi, deviation from PSS was observed at $NO_x$ values more than 10 ppb (Chate el al 2014). At such high $NO_x$ concentration, systematic deviation from PSS with Leighton ration less than 1 was observed.

Value of Leighton ratio =1 is a very rare finding in ambient environment. Hence, I again question the validity of conclusion drawn on this assumption. I again ask the authors to calculate j1 using TUV model or using solar radiation and check the Leighton ratio. In any case, given that j1 only depends on actinic flux, quantum yield and absorption cross-section, how would the authors explain a variation of more than an order of magnitude during the daytime hours of the same season (line 210 of the revised manuscript). Chate, D. M., et al. (2014), Deviations from the O3NONO2 photo-stationary state in Delhi, India, Atmospheric Environment, 96(0), 353-358, doi:http://dx.doi.org/10.1016/j.atmosenv.2014.07.054.

**Authors' response:** Thank you. As suggested by the reviewer, we have calculated the $j_1$ values using the NCAR Tropospheric Ultraviolet and Visible (TUV) Radiation model for 2010. We have used the missing information from scientific published values for the air quality monitoring stations. Those variables are

1. Overhead $O_3$ column in Dobson unit. The data is retrieved from National Aeronautics and Space Administration (NASA) website (https://ozoneaq.gsfc.nasa.gov/tools/ozonemap/) at the latitude and longitude of 13.76 ˚N and 100.50 ˚E (Bangkok, Thailand location).

2. Surface albedo. The data is retrieved from Janjai, S., Wanvong, W., and Laksanaboonsong, J.:The Determination of Surface Albedo of Thailand Using Satellite Data, The 2[nd] Joint International Conference on Sustainable Energy and Environment (SEE 2006), 21-23 November 2006, Bangkok, Thailand.

3. Cloud optical depth. The data is retrieved from NASA Earth Observations (NEO) (https://neo.sci.gsfc.nasa.gov/view.php?datasetId=MYDAL2_M_CLD_OT).

4. Aerosol optical depth and single scattering albedo (SSA). The data is retrieved from Janjai, S., Nunez, M., Masiri1, I., Wattan, R., Buntoung, S., Jantarach, T., and Promsen, W.: Aerosol Optical Properties at Four Sites in Thailand, Atmospheric and Climate Sciences, 2, 441-453, 2012.

The rate coefficients are calculated in 2010 for the dry season (January, February, March, April, May, October, November and December), during 10:00 LT to 16:00 LT, at the latitude and longitude of 13.7 ˚N and 100.5 ˚E. The $j_1$ values calculated from the NCAT TUV model are now shown in section F, supplement material in Table II.

**Table II:** Statistical analysis of the chemical rate coefficients ($j_1$ and $k_3$) based on an observational analysis during dry seasons at BKK sites, roadside and BKK suburb sites, 2010 to 2014.; and statistical analysis of $j_1$ based on a modeling analysis at the latitude and the longitude of 13.7 ˚N, 100.5 ˚E in a dry season, 2010.

| | Rate coefficient | | | | | | | | | | | |
|---|---|---|---|---|---|---|---|---|---|---|---|---|
| **Sites** | $j_1$ | | | | | | $k_3$ | | | | | |
| | min$^{-1}$ | | | s$^{-1}$ | | | ppm$^{-1}$ min$^{-1}$ | | | cm$^3$ molecule$^{-1}$ s$^{-1}$ | | |
| | **Max** | **Min** | **Average** | **Max** | **Min** | **Average** | **Max** | **Min** | **Average** | **Max** | **Min** | **Average** |
| **Based on observation*** | | | | | | | | | | | | |
| *BKK* | 0.95 | 0.12 | 0.74±0.2 | 0.016 | 0.004 | 0.008±0.035 | 30.9 | 28.6 | 29.8±0.7 | 2.06e$^{-14}$ | 1.99e$^{-14}$ | 2.02e$^{-14}$±2.01e$^{-16}$ |
| *Roadside* | 0.90 | 0.36 | 0.64±0.3 | 0.015 | 0.011 | 0.013±0.002 | 30.6 | 28.3 | 29.7 | 2.03e$^{-14}$ | 2.03e$^{-14}$ | 2.03e$^{-14}$ |
| *BKK suburb* | 1.22 | 0.34 | 0.55±0.3 | 0.022 | 0.007 | 0.010±0.004 | 30.9 | 28.8 | 29.8±0.7 | 2.04e$^{-14}$ | 2.01e$^{-14}$ | 2.03e$^{-14}$±1.34e$^{-16}$ |
| **Based on modeling**** | | | | | | | | | | | | |
| *13.7 ˚N, 100.5 ˚E* | | | | | | 0.021±0.002 | | | | | | |

The average $j_1$ value calculated from the NCAR TUV model is $0.021\pm0.0024$ s$^{-1}$, which is similar to our calculations based on observations in Table I of the manuscript ($j_1$ ranges from 0.008 to 0.013 s$^{-1}$).

The manuscript now includes the comparison of the $j_1$ result from the NCAR TUV model with our calculation. We are very encouraged by the similarity of the two results.

3.2 Cross over point and regime identification: First of all, legends are not provided in this figure 5. If I assume the purple points to be $O_3$, still the fit statistics (which are not even provided either in text or in figure) are very poor. So the conclusion drawn regarding cross over points are not robust. There is no clear crossover point for the BKK sites.

**Authors' response:** Thank you. We have modified Figure 5 by providing legends and clarity on the cross-over points.
Figure 5 is changed from:

[Figure]

Figure 5 is changed to:

[Figure]

**Fig. 5:** Relationships and crossover points of NO, NO₂ and O₃ at a) BKK sites b) roadside sites and c) BKK suburb sites; and data distributions of those species at d) BKK sites e) roadside sites and f) BKK suburb sites.

3.3 Section 3.4: The scatter plots have very poor fit for Fig 6a and Fig 6c for the non-episode events. In addition to the slope and intercept, authors should also consider the goodness of fit before drawing any conclusion.

**Authors' response:** Thank you. This is a very large (2010 to 2014) and robust air quality data set. Figure 6 provides the best linear regression lines during $O_3$ episodes and non-episodes condition and its relationship to the $O_3$ precursor $NO_x$. This has also been articulated by reviewer#2.

[Figure]

**Fig. 6:** Effects of local and regional contributions on $O_x$ during non-episode and episode days at a) BKK sites, b) roadside sites and c) BKK suburb sites.

3.4 Section 3.5.1 (Figure V of the supplement): Even in the best case, the r2 is less than 0.3 in the best case. What is the significance of local source analysis based on such poor statistics? Why the frequency distribution of SO2 (I assume it is frequency distribution has no information is provided either in figure caption or text) has wiggles in between.

**Authors' response:** We have now removed Figure V in the supplement and replaced it by a table that provides the correlations between CO and $NO_x$; and $SO_2$ and $NO_x$ (Table II). We have also provided the correlation among the species at all the monitoring sites in this table. Table II provides comprehensive statistical information.

**Table III:** Correlation between CO and $NO_x$; and $SO_2$ and $NO_x$ at BKK sites, roadside sites and suburb sites, during 2010 to 2014; together with ±1 standard deviation.

| Station type | Station ID | | Correlation | |
|---|---|---|---|---|
| | | | CO and $NO_x$ | $SO_2$ and $NO_x$ |
| BKK sites | 3T | | 0.76 | 0.34 |
| | 5T | | 0.56 | 0.37 |
| | 10T | | 0.76 | 0.36 |
| | 11T | | 0.68 | 0.33 |
| | 12T | | 0.61 | 0.26 |
| | 15T | | 0.64 | 0.29 |
| | 61T | | 0.85 | 0.28 |
| | | Average | 0.69±0.10 | 0.32±0.04 |
| Roadside sites | 52T | | 0.73 | 0.49 |
| | 54T | | 0.72 | 0.56 |
| | | Average | 0.72 | 0.53 |
| Suburb sites | 13T | | 0.92 | 0.32 |
| | 14T | | 0.64 | 0.11 |
| | 19T | | 0.47 | 0.39 |
| | 20T | | 0.55 | 0.21 |
| | 22T | | 0.71 | 0.27 |
| | 27T* | | 0.77 | 0.53 |
| | | Average | 0.68±0.16 | 0.30±0.15 |

**Note**: *the correlations are calculated based observations during 2010 to 2013

3.5 Lines 265-272: The statistics are too poor for the conclusion of ~10 ppb enhancement in $O_3$. The spread in delta $O_3$ ranges from -66 to +96 ppb.

**Authors' response:** Thank you. The delta $O_3$ analysis for Atlanta Metropolitan Region has been published (Lindsay and Chameides, 1988; Lindsay et al., 1989). This reference is provided in the manuscript. As discussed in the manuscript, ~10 ppb enhancement in $O_3$ for BMR is the average for the observation. These results are similar to Lindsay and Chameides, 1988 and Lindsay et al., 1989.

**Reviewer#2**

We thank the reviewer for the thoughtful reviews and comments. We are please that "I am writing to you that I accept all revisions to the comments and suggestions that I made to the manuscript". Moreover, we are pleased that the reviewer rates the manuscript as "Excellent" for the three categories including Scientific significance, Scientific quality and Presentation quality.